# Application of GRACE to the assessment of model-based estimates of monthly Greenland Ice Sheet mass balance (2003-2012)

N.-J. Schlegel[1,2], D. N. Wiese[2], E. Y. Larour[2], M. M. Watkins[2], J. E. Box[3], X. Fettweis[4], and M. R. van den Broeke[5]

[1]University of California, Los Angeles, Los Angeles, CA, USA
[2]Jet Propulsion Laboratory, California Institute of Technology, Pasadena, California, USA
[3]Geological Survey of Denmark and Greenland (GEUS), Øster Voldgade 10, 1350 København, Denmark
[4]University of Liège, Department of Geography, 4000 Liège, Belgium
[5]Institute for Marine and Atmospheric research Utrecht (IMAU), Utrecht University, Utrecht, The Netherlands

*Correspondence to:* Nicole-Jeanne Schlegel
schlegel@ucla.edu

**Abstract.**

Quantifying the Greenland Ice Sheet's future contribution to sea level rise is a challenging task that requires accurate estimates of ice sheet sensitivity to climate change. Forward ice sheet models are promising tools for estimating future ice sheet behavior, yet confidence is low because evaluation of historical simulations is challenging, due to the scarcity of continental-wide data for model evaluation. Recent advancements in processing of Gravity Recovery and Climate Experiment (GRACE) data using Bayesian-constrained mass concentration ("mascon") functions have led to improvements in spatial resolution and noise reduction of monthly global gravity fields. Specifically, the Jet Propulsion Laboratory's JPL RL05M GRACE mascon solution (GRACE_JPL) offers an opportunity for the assessment of model-based estimates of ice sheet mass balance (MB) at ~300 km spatial scales. Here, we quantify the differences between Greenland monthly observed MB (GRACE_JPL) and that estimated by state-of-the-art, high-resolution models, with respect to GRACE_JPL and model uncertainties. To simulate the years 2003-2012, we force the Ice Sheet System Model (ISSM) with anomalies from three different surface mass balance (SMB) products derived from regional climate models. Resulting MB is compared against GRACE_JPL within individual mascons. Overall, we find agreement in the Northeast and Southwest where MB is assumed to be primarily controlled by SMB. In the interior, we find a discrepancy in trend, which we presume to be related to millennial-scale dynamic thickening not considered by our model. In the Northwest, seasonal amplitudes agree, but modeled mass trends are muted relative to GRACE_JPL. Here, discrepancies are likely controlled by temporal variability in ice discharge, and other related processes not represented by our model simulations, i.e. hydrological processes and ice-ocean interaction. In the Southeast, GRACE_JPL exhibits larger seasonal amplitude than predicted by the models while simultaneously having more pronounced trends; thus, discrepancies are likely controlled by a combination of missing processes and errors in both the SMB products and ISSM. At the margins, we find evidence of consistent intra-annual variations in regional MB that deviate distinctively from the SMB annual cycle. Ultimately, these monthly-scale variations, likely associated with hydrology or ice-ocean interaction, contribute

to steeper negative mass trends observed by GRACE_JPL. Thus, models should consider such processes at relatively high (monthly-to-seasonal) temporal resolutions to achieve accurate estimates of Greenland MB.

## 1 Introduction

The Greenland Ice Sheet is a significant source of sea level rise, contributing approximately 0.75 mm/yr over the last decade (Shepherd et al., 2012; Luthcke et al., 2013; Velicogna et al., 2014), and its rate of contribution is expected to accelerate in the coming centuries (Pachauri et al., 2014; Church and White, 2006, 2011). The quantification of Greenland's future contribution to sea level rise is a challenging task, and uncertainty in such an estimate is high. The largest source of this uncertainty is estimation of future contribution of ice sheet surface mass balance (SMB), or the sum of atmospheric processes: snow accumulation, surface runoff (including melt-water retention and refreezing), and evaporation (Vernon et al., 2013; Enderlin et al., 2014). An additional source of this uncertainty is the estimation of how much ice the ice sheet will discharge into the ocean. This requires an accurate understanding of ice flow sensitivity to future changes in surface mass balance and of the physical processes responsible for driving rapid changes in ice dynamics, specifically those related to basal hydrology and ice-ocean interactions (e.g., Hanna et al., 2008; Holland et al., 2008; Schoof, 2010; Walter et al., 2012; Bell et al., 2014; Moon et al., 2014). Often, current observable trends in ice sheet MB are extrapolated (Shepherd and Wingham, 2007; Velicogna, 2009; Rignot et al., 2011) in order to estimate future changes to sea level; however, such projections are not grounded in a physical understanding of the ice sheet. Conversely, computational tools such as numerical ice sheet models, that have been validated against historical data, offer a well-informed physically based method for such future projections (e.g., Huybrechts, 2002; Price et al., 2011; Perego et al., 2012; Gillet-Chaulet et al., 2012; Greve and Herzfeld, 2013; Aðalgeirsdóttir et al., 2014).

Forward model simulations that include numerical ice sheet models are the most promising tools for estimating Greenland's future contribution to sea level. However, model-based estimates of Greenland MB are associated with large uncertainties that are difficult to quantify. For instance, simulation results are dependent on spinup procedure, treatment of boundary conditions/model forcing, implementation of sliding laws, spatial resolution, and choice of ice flow equations (e.g., Nowicki et al., 2013; Goelzer et al., 2013; Greve and Herzfeld, 2013; Aschwanden et al., 2013; Yan et al., 2013; Aðalgeirsdóttir et al., 2014), to name a few. Thus, in order to assess the uncertainties associated with a particular ice sheet model simulation, it is necessary to specifically quantify uncertainties associated with the representation of ice dynamic processes - i.e., basal lubrication due to surface runoff reaching the bed (Schoof, 2010; Bartholomew et al., 2012; Tedesco et al., 2012; Joughin et al., 2013), warming of ice due to runoff refreeze (Fausto et al., 2009; Phillips et al., 2010, 2013), ice/ocean interaction including grounding line retreat (Holland et al., 2008; Rignot et al., 2010; Chauche et al., 2014), changes to flow resistance at the calving front (Nick et al., 2010; Walter et al., 2012), as well as limitations in model spinup and errors in surface mass balance.

In particular, it is difficult to quantify the contribution from ice dynamic processes, because the physical mechanisms associated with these processes are highly non-linear and not well understood. As a result, it has been challenging for continental ice sheet model simulations to accurately capture the complex variability of Greenland MB observed over the last decade. On a regional scale, however, there has been steady progress towards improving the physical representation of specific dynamic

processes. For instance, Nick et al. (2010) implemented a physically-based calving model that captured the seasonal cycle of ice front advancement and retreat, similar to those observed in four of Greenland's major outlet glaciers (Nick et al., 2013), and there has been significant advancement in the glacier-scale modeling of surface runoff drainage and enhancement of basal sliding (e.g., Hewitt, 2013). While these models have been implemented and validated regionally, they have not been generally adapted into continental-scale ice sheet models. Instead, on the continental scale, recent efforts have used simplified parameterizations to model outlet response to ocean warming and/or enhance basal lubrication due to runoff, and this strategy has proven successful in reproducing present-day observed trends in Greenland total MB (e.g., Price et al., 2011; Fürst et al., 2015). Unfortunately, quantification of the uncertainties associated with highly parameterized projections is challenging, especially considering the fundamentally non-linear nature of the temporally-varying processes being represented.

In terms of ice sheet model spinup, a number of studies have been conducted to determine the sensitivity of model results to the model setup and spinup methods (e.g., Rogozhina et al., 2011; Gillet-Chaulet et al., 2012; Aschwanden et al., 2013; Goelzer et al., 2013; Greve and Herzfeld, 2013; Yan et al., 2013; Saito et al., 2016). Overall, such studies conclude that results are indeed sensitive to the spinup procedures. The various methods used to spinup ice sheet models are widely diverse, but they are typically based in either (1) the use of inversion techniques to infer spatial patterns of largely unknown parameters (i.e. basal drag); or (2) a paleoclimate spinup over thousands of years. Both of these methods have advantages and disadvantages, and each has been modified in various ways to produce results that better match observations. For instance, the first method results in a surface velocity field and surface topography that is similar to present, but spurious errors arise from the use of mis-matched observational datasets and forcing products. As a result, this method is often followed by a model relaxation procedure (e.g., Gillet-Chaulet et al., 2012; Schlegel et al., 2013), which helps to remove these errors but also changes the observationally-based model state. The second method has the advantage that the simulated present-day ice sheet has memory of past glacial cycles in its thermal and mechanical state, so ice dynamics resulting from past climatic changes are captured. The disadvantage of this procedure is that, often, the simulated present day ice sheet velocities and surface topography vary significantly from those observed. Recently, new methods have been implemented to address this mismatch. Particularly, the use of a flux-correction method during the few thousand years of simulation prior to present, has led to improved agreement between observations and the simulated present day state of the ice sheet (e.g., Price et al., 2011; Aschwanden et al., 2016). While many improvement continue to be made in this field, quantification of uncertainties associated with choice of spinup procedure and model setup remains a challenge. In fact, the proper quantification of uncertainties in a particular method likely requires analyses that are computationally expensive, including thorough sensitivity analysis and formal propagation of error within the model code itself (Aschwanden et al., 2013).

In comparison to model spinup and ice dynamic processes, uncertainties associated with Greenland's SMB are relatively well understood. Indeed, over the last decade, notable progress has been made in understanding the temporal and spatial variability of Greenland's SMB (and its components), through dynamical downscaling of available observations. For example, Regional Climate Models (RCMs) like the Regional Atmospheric Climate Model version 2.3 (RACMO) (van Meijgaard et al., 2008) and the Modèle Atmosphérique Régional (MAR) (Gallée and Schayes, 1994) run complex surface snow models and are capable of resolving SMB over Greenland at considerably high spatial resolutions (i.e. 5-25 km) (e.g., Fettweis et al.,

2005; van Meijgaard et al., 2008). Output from these models offer insight into SMB variability, as well as the errors associated with each. On regional scales, various RCMs have been shown to have good agreement. However, these products show less agreement on local scales, particularly in terms of the relative magnitudes of SMB components, spatial patterns, or seasonal amplitudes (Rae et al., 2012; Vernon et al., 2013). Such variations in SMB forcing have been shown to be sources of uncer-

tainties in forward ice flow simulations (Schlegel et al., 2013). This is particularly the case considering that variations in SMB - including increased melt and subsequent drainage of runoff - may alter ice flow and contribute to changes in ice discharge. In addition, physically based models of surface processes are difficult to evaluate due to the scarcity of continental-wide data for SMB model validation (Lucas-Picher et al., 2012). As a consequence, while on broad regional scales SMB errors are relatively well documented, local partitioning of Greenland's MB between SMB and discharge - and associated uncertainties

- remains difficult to quantify accurately (Howat et al., 2007; van den Broeke et al., 2009; Pritchard et al., 2009; Kjær et al., 2012; Yan et al., 2013; Khan et al., 2014).

Another challenge is the lack of observational data for ice sheet model evaluation. In response to the need for such data, government agencies have deployed a number of instruments for the purpose of monitoring the MB of the polar ice sheets through satellite altimetry, interferometry, and gravimetry (Pritchard et al., 2009; Zwally et al., 2011; Rignot et al., 2011; Shepherd et al.,

2012; Velicogna and Wahr, 2013). For more than a decade, the joint U.S.-German Gravity Recovery and Climate Experiment (GRACE) has continuously acquired time-variable measurements of the Earth's gravity field and has provided unprecedented surveillance of the MB of the Polar ice sheets (e.g., Wu et al., 2002; Luthcke et al., 2006; Baur et al., 2009; Velicogna, 2009; Wu et al., 2009; Chen et al., 2011; Schrama and Wouters, 2011; Jacob et al., 2012; Sasgen et al., 2012; Velicogna and Wahr, 2013; Schrama et al., 2014). GRACE data are typically processed by estimating gravity field variations using unconstrained

spherical harmonic basis functions. These estimates ultimately suffer from a highly correlated error structure in the form of longitudinal stripes, and a variety of methods have been developed to remove these artifacts. Recent advancements in GRACE data processing have provided a Bayesian-framework for removing this correlated error structure (ultimately resulting in improved spatial resolution and noise reduction) using mascon basis functions rather than spherical harmonics (Luthcke et al., 2013; Watkins et al., 2015). Such solutions now offer an opportunity to improve upon the current assessments of model-based

estimates of Greenland MB.

Here, we take advantage of a high-resolution ($\sim 300$ km) monthly mascon solution for the purpose of mass balance comparison with independent historical model-based estimates of Greenland Ice Sheet mass balance evolution. A mascon-by-mascon comparison is made between the model-based mass balance estimates and the JPL RL05M GRACE mascon solution provided by the Jet Propulsion Laboratory (GRACE_JPL) (Wiese et al., 2015), and uncertainties are quantified for each. The model-

based estimates of Greenland MB are derived from ice sheet model forward simulations forced with SMB anomalies from three different RCM-based estimates of historical SMB. Here, we use the Ice Sheet System Model (ISSM) (Larour et al., 2012), a state-of-the-art finite element ice sheet model that is run on an anisotropic mesh at high (1 km) spatial resolution within Greenland's fast-flowing outlet glaciers. To best capture observed surface velocities, we use inversion techniques to infer for basal drag followed by model relaxation to reduce model drift. In order to minimize uncertainty in the model simulations, we model

only the spatial and temporal dynamic ice flow response to recent changes in historic SMB. Therefore, model results presented

here do not include physical representations for 1) rapid changes in ice dynamics driven by temporally varying processes - such as ice-ocean interaction and basal hydrology - nor 2) background dynamic ice thickness changes related to millenium-scale changes in climate. Consequently, model MB estimates are expected to disagree with GRACE_JPL in locations where these processes are important. In fact, we deliberately do not rely upon simplified parameterizations of rapidly changing ice dynamics nor upon on a paleo-climate spinup, as doing so would introduce additional model-based uncertainties that would be highly challenging to quantify. Instead, we assume modeled MB uncertainty is sourced in the SMB forcing itself, which we define to be the 1-sigma spread between the three different model simulations. Using this approach, we identify regions where the modeled MB estimates disagree with GRACE_JPL in both time and space, outside of the assessed uncertainties. Subsequently, we are able to statistically quantify the magnitude of this disagreement and hypothesize about which factors (i.e., errors in SMB, assumptions in ISSM spinup, or physical processes not in ISSM) may be responsible. This study is being conducted in the spirit of ultimately improving model estimates of Greenland MB (including SMB and ice dynamics) to enable reduction of uncertainty in future projections of sea level. While several studies have conducted basin-scale comparisons of GRACE against only the SMB component of Greenland MB (i.e., Sasgen et al., 2012; Velicogna et al., 2014), this study constitutes the first direct comparison of GRACE data with ice sheet model output at regional ($\sim$ 300 km) spatial scales.

This study is organized as follows: in the first section we describe the GRACE_JPL mascon solution. In the second, we discuss the models and describe the model application to the Greenland Ice Sheet, our spinup methodology, model inputs, and estimates of SMB. In the third section, we discuss the quantification of uncertainty in both the GRACE_JPL solution and the model output. In the fourth section, we present results, focusing on spatial and temporal comparisons between the model estimates of Greenland MB and GRACE_JPL mascon solutions. Finally, we discuss the application of GRACE_JPL mascon solutions to the quantification of regional ice discharge and to the evaluation of model-based estimates of historical ice sheet mass balance. With consideration to the calculated uncertainties, we hypothesize about regional partitioning between SMB and discharge and how the relative contribution of these mass balance components may vary seasonally.

## 2  JPL RL05M GRACE mascon solution

This study utilizes a new GRACE gravity solution, JPL RL05M Version 2 (publicly available at www.grace.jpl.nasa.gov), which solves for monthly gravity anomalies in terms of equal-area 3-degree surface spherical cap mass concentration ("mascon") functions (Watkins et al., 2015; Wiese et al., 2015), for which it takes 4551 to cover the surface of the Earth. This mascon solution is fundamentally different than other mascon solutions (Luthcke et al., 2013), both in the type of basis function used, as well as the choice of regularization to remove correlated error. The JPL RL05M solution is unique in the sense that it applies statistical information on expected mass variability derived from geophysical models and altimetry data to condition the solution and remove correlated error. This solution has been shown to have slightly better spatial resolution than spherical harmonic solutions, and in particular has shown significant improvement in recovering ocean mass variations, including ocean currents (Landerer et al., 2015), which are small in amplitude and typically difficult to detect.

Over the Greenland Ice Sheet, the solution is relatively unconstrained, and not guided by any model of surface mass varia­tions, since these are poorly understood from a physics-based modeling perspective. As such, the model output assessed here and the GRACE_JPL data are completely independent of each other. The placement of the mascons is seen in Fig. 1. Note that this placement was arbitrary in the derivation of the JPL RL05M solution and was not optimized for any specific application of the GRACE data (i.e., recovering Greenland mass variations). Note also that the estimate of mass in each mascon can be considered relatively independent of the other mascons, as we do not apply any apriori spatial correlation between mascons (this choice is appropriate for high latitudes due to dense ground track coverage), and the formal posteriori covariance matrix indicates small correlations between adjacent mascons. This solution has been shown to agree with previously published re­sults (within formal error bars) regarding the total rate of mass loss from the Greenland Ice Sheet (Watkins et al., 2015). In this analysis we examine mass changes over the ice sheet with the native resolution of the gravity solution (i.e. individual mascons), constituting the highest spatial resolution analysis of the Greenland Ice Sheet from GRACE data thus far.

The GRACE data have certain known limitations, and as such, we apply standard post-processing procedures to correct for these. The $C_{20}$ coefficient (defining the oblateness of the Earth), which is poorly observed by GRACE, has been substituted with an estimate derived from Satellite Laser Ranging (SLR) data (Cheng and Tapley, 2004). GRACE does not observe movement of the center of mass of the Earth, since the satellites orbit this point at all times; as such, we use an estimate of geocenter motion from Swenson et al. (2008). The position of the Earth's mean pole has been corrected using the recommendation of Wahr et al. (2015). Known jumps in the background atmosphere dealiasing products (occurring in 2006 and 2010) have been corrected (Fagiolini et al., 2015). Finally, the solid Earth glacial isostatic adjustment (GIA) signal has been removed from the GRACE data using the model provided by A et al. (2013), in an effort to isolate only surface mass variations for a direct comparison against the ice sheet model. Each of these post-processing corrections is consistent with the publicly available data described in Watkins et al. (2015), with the exception of the correction to the Earth's mean pole (Wahr et al., 2015) and the correction to the background atmosphere dealiasing products (Fagiolini et al., 2015). These two corrections have not been applied to the publicly available data.

One post-processing algorithm that is unique to the JPL mascon solution concerns the treatment of leakage errors. Many of the Greenland mascons lie on both land and ocean regions (Fig. 1), and as such, the solution for these mascons will contain the average mass of both the land and ocean. We apply a Coastline Resolution Improvement (CRI) filter to the data (Watkins et al., 2015) to separate the land and ocean portions of each of these mascons. As such, in all analyses presented here, the ocean mass from each mascon has been removed, and we are analyzing only the land component of each mascon. For further details on the JPL RL05M mascon solution and the CRI filter, the reader is referred to Watkins et al. (2015). From hereafter, we refer to the JPL RL05M GRACE mascon solution as GRACE_JPL.

## 3 Model Descriptions

### 3.1 Ice sheet model

The Ice Sheet System Model is a thermo-mechanical finite-element ice flow model. It relies upon the conservation laws of momentum, mass, and energy, combined with constitutive material laws and boundary conditions. The implementation of these laws and treatment of model boundary conditions are described by Larour et al. (2012). In this study, we simulate the Greenland Ice Sheet with a two-layer thin-film approximation (L1L2) (Schoof and Hindmarsh, 2010; Hindmarsh, 2004), implemented within ISSM. The L1L2 formulation is based on the Stokes equations, includes effects of longitudinal stresses, considers the contribution of vertical gradients to vertical shear, and assumes that bridging effects are negligible.

### 3.2 Initialization and Relaxation

The strategy for ISSM Greenland continental initialization, relaxation, and spinup is described in detail by Schlegel et al. (2013, 2015). For this study, the anisotropic mesh is composed of 91,490 triangular elements, refined using observed surface elevation (Scambos and Haran, 2002) and surface velocity (Rignot and Mouginot, 2012) fields (Fig. S1). Mesh resolution is set to a minimum of 1 km in steep areas with high velocity gradients and to a maximum of 15 km at the ice divides. We initialize the bedrock geometry with 150 m gridded BedMachine bedrock (Morlighem et al., 2014a) and ice surface data (Howat et al., 2014). Three dimensional ice temperature and ice viscosity are derived from a steady-state higher-order (Blatter, 1995; Pattyn, 2003) three-dimensional solution of the thermal regime, using observed velocities, surface temperatures, and geothermal heat flux (Larour et al., 2012; Seroussi et al., 2013). Surface temperatures are from Ettema et al. (2009) and geothermal heat flux estimates are from Shapiro and Ritzwoller (2004). We determine the spatially-varying basal drag coefficient using inverse methods (MacAyeal, 1993), following Morlighem et al. (2010), in order to best match the modeled ice surface velocities with InSAR surface velocities (Rignot and Mouginot, 2012). We hold ice viscosity and basal drag constant during the forward simulation.

For model relaxation, we consider that ice sheet total mass balance is comprised of two major components, SMB and discharge (D), such that $MB = SMB - D$. For the ISSM simulations presented here, basal hydrology is not simulated; therefore, we consider the local basal mass balance to be equal to zero everywhere and assume that the value does not change over time. As a result, a fully relaxed ice sheet model, with MB near zero, would be in a "steady"-state such that the ice outflux from the model margins (D) is nearly equal to SMB.

The goal of the relaxation step is to achieve an ice sheet that is in virtual steady-state with regards to ice thickness and velocity, through relaxation to a reference SMB climatology, after Schlegel et al. (2013, 2015). This reference SMB climatology serves as the base SMB forcing for all of our historic simulations; therefore, ideally it would 1) represent a historical time period when total Greenland MB was believed to be relatively stable (i.e. close to zero) and 2) be defined for all SMB forcing products such that anomalies can be calculated against a consistent time period (see Sect. 3.3 for a description of these products and the associated timeframes for which they are available). Previous studies have shown that the ice sheet had a mass balance near zero (i.e. it was close to steady-state) in the 1970s-1980s (e.g., Rignot et al., 2008). The period 1979-1988 satisfies

both of the above criteria, and as such is chosen to serve as our reference climatology. Note that even though we choose this particular time period as reference, additional experiments reveal that our results are not sensitive to the relaxation climatology chosen (e.g., Figs. S2B, S3B, Appendix Sect. 4, and additional experiments not shown), in agreement with studies conducted by Fettweis et al. (2013).

Thus, to accomplish model relaxation, we force the ISSM Greenland with 1979-1988 average SMB estimates from Box (2013) (hereafter referred to as $\overline{SMB}$, Appendix Sect. 1), for a total of 56,000 years. The SMB product from Box (2013) is chosen because it is defined for the longest period of time, starting in 1840. SMB is interpolated onto the ISSM Greenland mesh and is imposed through a one-way coupling scheme (Schlegel et al., 2013). During relaxation, we find that the major outlet glaciers slow, and total volume of the ice sheet is reduced from its present-day initialization value by 2.6% (for more

details, see Appendix Sect. 4, Fig. S4 and Fig. S5). This resulting relaxed model state (ice thickness, bedrock elevation, and ice velocities), represents our assumed state of the ice sheet in 1840, and serves as the initial state for the historic ISSM spinup simulation described in the following section (Sect. 3.3).

Note that in this study, it is our approach to use state-of-the-art assimilation techniques to best capture the Greenland Ice Sheet's present-day state, specifically surface topography and surface velocities at the beginning of our simulation. This ap-

proach offers the advantage of a well-captured present-day surface velocity field. However, it is important to acknowledge that there are key limitations, including lack of validation due to general uncertainty about the state of the ice sheet in 1840 and mismatch between the observational datasets (i.e. surface velocity map derived from measurements acquired in 2008-2009 ) that have been assimilated into the fields of ice viscosity and basal drag during initialization. In addition, because we do not spinup the model through past climate and instead relax the ice model towards steady-state, our results do not reflect present-

day changes to the ice sheet that may be occurring in response to climatological conditions prior to 1840 (e.g., Colgan et al., 2015; MacGregor et al., 2016). Instead, for this study, our model-based mass balance estimates consider any remaining drift in the relaxed ice sheet model (represented by a control run, illustrated in Fig. 2), a high-resolution estimate of the SMB forcing over that period, and an ice model calculation of the ice dynamic response to the historical (1840-2012) SMB forcing.

### 3.3    Models of Historical SMB

After relaxation, we spinup the forward ice sheet model for 173 years, from 1840-2012, using reconstructed monthly SMB after Box (2013). This product, hereafter referred to as BOX, is described in more detail in Appendix Sect. 1. The 173-year historic run constitutes our base simulation. We plot the total yearly BOX SMB forcing for this run in gray in Fig. 2. On the same figure, in red, we plot the monthly evolution of total ice sheet mass during spinup. From 1840-1900, the model maintains a mass balance near zero. After 1900, the overall trend in ice sheet total mass balance is dominated by mass loss

until 1970, when accumulation over the ice sheet increases. During the following decades - which include the period used as climatological reference period for $\overline{SMB}$ - through the end of the 1990's, we find that the simulated ice sheet re-achieves a near stable condition, growing only slightly from 1970 to 2000.

The state of the ice sheet (dictated by ice thickness, bedrock elevation, and ice velocities), at the end of year 1978, is the initial condition for two additional historic simulations. For these simulations, we restart from the 1978 model state and force

ISSM Greenland with the SMB anomalies (with respect to the specific product's 1979-1988 mean) from two different regional, coupled surface-atmosphere models: a) MAR 3.5.2 [1979-2014] (hereafter referred to as MAR), run at a 10 km resolution, downscaled to 5 km (Fettweis et al., 2005, 2011) and b) RACMO 2.3 [1979-2014] (hereafter referred to as RACMO), run at an average 11 km resolution (van Meijgaard et al., 2008; Ettema et al., 2009). The total SMB forcing for each RCM product is

equal to $\overline{SMB}$ plus the monthly SMB anomalies derived for that particular product beginning in 1979. It is important to note that both of these RCMs are forced at their lateral boundaries by the European Centre for Medium-Range Weather Forecasts (ECMWF) Interim (ERA-I) reanalysis, which begins in 1979 (Uppala et al., 2005). The SMB forcing is applied to ISSM as a monthly forcing of ice-equivalent thickness change, thus snow compaction and firn densification are captured only by each RCM's surface snow model and in that model's specific determination of SMB. In this study, we will focus on comparison of

historical simulation results from 2003-2012, which is the time overlap between the GRACE_JPL solution and the three SMB forcing products considered here.

## 4    GRACE period mass estimates

The purpose of this study is to compare ISSM forward simulations, forced with three different high-resolution RCM-derived SMB products, against the monthly GRACE_JPL product, in order to highlight the regions where modeled ice sheet mass

differs from GRACE outside of the assessed uncertainties. The resolution at which the comparison is made is limited by the spatiotemporal resolution of the GRACE data; therefore all comparisons are made on monthly timescales, from 2003-2012, at the spatial resolution of individual mascons ($\sim$110,000 km$^2$).

In this analysis, we consider 2003-2012 SMB anomalies (with respect to $\overline{SMB}$, hereafter, referred to as SMB_GrIS BOX, SMB_GrIS MAR, and SMB_GrIS RACMO), ISSM Greenland Ice Sheet ice thickness changes, and the temporal evolution

of mass beyond the ice sheet margin (hereafter referred to as periphery). To determine the modeled mass balance within the Greenland ice sheet boundary, we assume an ice density of 917 kg/m$^3$ and assess ISSM-modeled mass changes within all mascons that contain portions of the Greenland land mass (Fig. 1). For the periphery, we assess areas of bare rock/tundra and the glaciers/ice caps that are not physically attached to the ice sheet. A high resolution (1/120 degree) mask, distinguishing the ice sheet, peripheral ice, and land (Gardner et al., 2013) is relied upon to create the original ISSM domain outline of the ice sheet.

We then use this mask to categorize all land within the Greenland mascons (Fig. 1). Also, note that because GRACE does not capture mass change over floating ice, we remove the mass signal from areas classified as ice shelf (Morlighem et al., 2014a) for this analysis. The outline of each mascon is projected into the ISSM Greenland coordinate projection (polar stereographic projection with standard parallel at 71°N and a central meridian of 39°W). Within the ice sheet boundary, mass changes are considered on individual elements of the ISSM mesh and outside of the ice sheet boundary, mass changes are considered on

individual elements of a 10 km triangular mesh. To assess mass change within each mascon, elements within the projected mascon boundaries are summed, and elements bisected by mascon boundaries contribute to this sum proportionally (by area) to the mascons that fall within their individual outlines. This procedure is mass conserving on the continental-scale; however,

it introduces small leakage errors along the mascon boundaries that are insignificant compared to the uncertainties considered in this study (which we describe in detail in Sect. 5 and Appendix Sect. 3).

Cumulative mass change within each Greenland mascon is determined monthly from 2003-2012. First, over the ice sheet area, we sum the SMB anomalies for each of the forcing products: SMB_GrIS BOX, SMB_GrIS MAR, and SMB_GrIS RACMO, over time. This resultant mass signal for each product represents the anomalous SMB forcing for the ISSM historical simulations. Next, we sum mass changes simulated by ISSM Greenland for the BOX, MAR, and RACMO historic simulations (hereafter, referred to as ISSM_GrIS BOX, ISSM_GrIS MAR, and ISSM_GrIS RACMO). This mass signal represents the ISSM model estimate of ice sheet total mass balance through time and is comprised of the anomalous SMB forcing and the dynamic response to SMB changes since the year 1840.

Finally, we assess the monthly mass change over the peripheral areas, which includes bare rock/tundra and glaciers and ice caps, as these signals are captured by GRACE_JPL. Peripheral mass changes have previously been shown to be significant, on the order of -40 Gt/yr for the period of 2003-2009 (Gardner et al., 2013). Because each SMB forcing product represents accumulation and melt differently over the bare rock/tundra, our analysis methods vary depending on the variables available from each product. Peripheral glaciers and ice caps are assumed to not evolve dynamically; therefore, we assume that SMB is the only component that drives the cumulative mass trend in those regions. Hereafter, we refer to the individual ISSM simulation results plus periphery estimates as ISSM_GrIS+P BOX, ISSM_GrIS+P MAR, and ISSM_GrIS+P RACMO. For details on how mass balance is calculated for each SMB product on the periphery, see Appendix 2.

## 5   Quantification of errors and uncertainty

Uncertainty in both the GRACE_JPL surface mass estimates and in the SMB-forced ice sheet model estimates of Greenland MB are considered in this study. Details on our assessment of these values are provided below.

### 5.1   Uncertainty in GRACE_JPL surface mass estimates

Error is assessed in GRACE_JPL using the diagonal elements of the formal posteriori covariance matrix from the gravity field inversion. The covariance matrix indicates that adjacent mascons have small correlations ($\sim$0.2) with each other. As such, each mascon is assumed to be uncorrelated with neighboring mascons. A mascon-by-mascon comparison to ICESat altimetry data (Csatho et al., 2014) validates this assumption, showing excellent agreement (Fig. S6). Additionally, leakage errors are considered by assuming a 50% error in the ability of the CRI filter to perfectly separate land/ocean mass within mascons that span coastlines. GIA model uncertainty is taken to be the 1-sigma spread of an ensemble of four GIA models, providing an uncertainty over the Greenland Ice Sheet of $\pm$15 Gt/yr, in good agreement with what is reported in Velicogna and Wahr (2013). In our analysis, GIA model uncertainty is shown as one that increases linearly with time when interpreting GRACE observations. For further details on the derivation of uncertainty in GRACE_JPL surface mass estimates, see Appendix Sect. 3.

## 5.2 Model Uncertainty

We take uncertainty in the modeled estimate of mass balance to be the 1-sigma spread between ISSM_GrIS+P BOX, ISSM_GrIS+P MAR, and ISSM_GrIS+P RACMO. As such, the ranges presented capture uncertainty rooted solely in the SMB models and in the ISSM simulations. In terms of the RCM uncertainties, Vernon et al. (2013) found that the disagreement between various RCMs is generally larger than the combined errors of the individual models, and they thus concluded that the errors reported by individual models are likely underestimated. These finding support our treatment of SMB model errors. This approach allows us to explicitly identify regions for which GRACE_JPL and ice sheet model output diverge outside of formal uncertainties, and attribute these differences to likely error in ISSM (which could be due to limitations in spinup, lack of a basal hydrology model, unmodeled ocean-ice interactions, errors in bedrock, errors in the basal drag coefficient, or resolution limitations with the mesh size). Note that since all SMB products are based on output from RCMs that are forced at the boundaries with the ERA-I reanalysis, there could be common mode errors in the SMB products that are not considered here.

## 6 Results

### 6.1 Greenland Cumulative Mass

For an overall comparison between GRACE_JPL and the ISSM Greenland simulation results, we plot the total cumulative Greenland mass over the 10-year study period in Fig. 3A and the total interior cumulative mass in Fig. 3B. These plots include the mean total SMB anomaly of RCM-derived forcing over the ice sheet (SMB_GrIS), the mean simulation results of the ISSM Greenland historical runs over the ice sheet (ISSM_GrIS), and the mean ISSM_GrIS simulation results over the ice sheet plus the calculated mass change over the Greenland periphery (ISSM_GrIS+P). All timeseries are plotted as cumulative mass through time and are offset to begin at zero at the start of 2003. Plots showing the timeseries for the three individual runs plus periphery are also provided for reference (Fig. S7). Note that in terms of cumulative mass, ISSM_GRIS+P MAR has the largest negative trend, ISSM_GrIS+P RACMO has the smallest negative trend, and the ISSM_GRIS+P BOX timeseries falls between ISSM_GRIS+P MAR and ISSM_GrIS+P RACMO. Additionally, with exception to 2012, the ISSM_GRIS+P BOX timeseries is very similar to that of the model mean. We quantify the linear 10-year trends for each individual RCM and their associated means and summarize them in Table 1. It is immediately clear that the model estimates (ISSM_GrIS+P) of the trend in cumulative total Greenland ice sheet mass are less negative than captured by GRACE_JPL, and account for only 64% of the total GRACE signal. The seasonal variability, on the other hand, appears to be well captured by the ISSM_GrIS+P estimates. Note that the reported GRACE_JPL trend of -284 Gt/yr includes Mascon 33 which includes a portion of Ellesmere Island, so it is not a true estimate of mass change solely over Greenland.

### 6.2 Regional Trends and Amplitudes

Within the interior (Fig. 1), we find that the total signal for GRACE_JPL is positive throughout the study period (Fig. 3B), while the models suggest that mass increases until 2006 and remains neutral for the second half of the simulation. For the

ten-year period, the total interior discrepancy between GRACE_JPL and the model estimate is $9 \pm 4$ Gt/yr (most pronounced in Mascons 58 and 88 - Fig. 4C), which would be equivalent to an average (unmodeled) dynamic background thickening of approximately $2 \pm 1$ cm/yr ice equivalent within the area defined by the interior mascons (Fig. 1). Overall, the comparison suggests that GRACE_JPL does capture dynamic thickening in the Greenland interior, as the differences in trend between

GRACE_JPL and the models are slightly outside the uncertainty estimates for these products. We also note that these interior trends are small relative to trends in the marginal mascons, and as a consequence the interannual variability of the GRACE_JPL signal is a significant feature in the total mass balance time series (Fig. 3B).

In Fig. 4, we plot the difference in trend spatially, per mascon. The trend is obtained by simultaneously fitting a linear trend along with sinusoids with frequencies of once and twice per year to each timeseries of mass. We find that the majority of the

discrepancy occurs in specific regions: Mascon 167 (Kangerdlugssuaq), Mascon 266 (Southeast glaciers), and Mascons 86/87 (Northwest glaciers). Some Southwest mascons also contribute to this discrepancy, though to a lesser extent. For instance, in Mascon 165 (Jakobshavn Isbræ) the models result in a smaller negative trend by about 15 Gt/yr, and in Mascon 212 the models result in a larger negative trend by about 10 Gt/yr.

In Fig. 5 we plot the mean annual amplitudes for each mascon. The annual amplitude is calculated by first removing a

13-month running mean from each mascon timeseries, and then simultaneously fitting a sinusoid with a frequency of once and twice per year to each timeseries of mass. Overall, the annual amplitudes are well captured by the model results, suggesting that the seasonal variability of SMB and its spatial distribution are most likely well represented by the three forcing products. This is especially the case for the mascons that disagree the most in trend (i.e. 86, 87, 167, and 214), suggesting that errors in the SMB models are not dominantly responsible for the differences between modeled mass trends and GRACE_JPL. More likely, these

differences are related to background dynamics (not considered in our steady-state ice sheet model spinup) and to increases in marginal ice discharge, driven by temporally varying processes not modeled here (including the effects of hydrology and ice-ocean interaction).

### 6.3   Contribution from SMB-driven Ice Dynamics

Notable in Fig. 3A is the difference in trend between ISSM_GrIS and the SMB_GrIS. Indeed, the ISSM_GrIS trend is less

steep than that of the SMB forcing anomalies, by $14 \pm 6$ Gt/yr (Table 1). This difference represents the total mass balance contribution from SMB-driven dynamics, as calculated by ISSM. In Fig. 6, we plot the regional distribution of the SMB-driven dynamics, for trend and amplitude. Spatially, the modeled dynamic response contributes to the trend predominantly in the south (especially in the Southeast) and the Northeast. Strikingly, SMB-driven dynamics contribute the most to the mass balance in the marginal mascons: both positively to the trend (Fig. 6A) and negatively to the annual amplitude (Fig. 6B).

Further analysis of the ISSM_GrIS simulations reveals that these results depend strongly on the amount of marginal runoff in the SMB forcing. Larger runoff, and consequently more negative SMB forcing in the margins, directly leads to thinning in the lower elevations of the ice sheet and ultimately results in an overall decrease in total ice discharge into the oceans (Goelzer et al., 2013; Enderlin et al., 2014). In addition, we find that as runoff increases through time (Fig. 2) the margins thin (Fig. S8B), flatten (Fig. S9B), and slow down (Fig. S8B). This results in dynamic thickening (ice thinning at a lesser rate than

that predicted by the SMB forcing), especially in the Southeast and in the large outlet glaciers in the north (Fig. S9A, see Appendix Sect. 4). We find that especially in these areas, flattening in the ablation zone decreases the driving stresses (Fig. S9C) and therefore, the marginal velocities (Fig. S8B). The consequence of these changes is an ultimate decrease in marginal mass flux, due to 1) a local decrease in ice thicknesses and 2) a local decrease in ice velocities. The resultant feedback, between

SMB forcing, marginal thinning, and ice discharge, dominates the ice sheet model response to SMB forcing during the study period. Consequently, our simulations indicate that the simulation with the more negative SMB (i.e. ISSM_GrIS MAR) loses less mass as ice discharge and more as runoff; similarly, the simulation with a less negative SMB (i.e. ISSM_GrIS RACMO) loses more mass as ice discharge and less as runoff (Table 1 and Fig. S7). As a result, the uncertainty range for all three simulations (ISSM_GrIS) is less than the uncertainty range predicted by the SMB models themselves (SMB_GrIS) (Fig. 3A

and Table 1).

While SMB-driven dynamics predicted by ISSM are most prominent along the margins, we also find that in the southern interior, a decrease in SMB is responsible for minor thinning (Fig. S8B) and resultant dynamic thickening (i.e. Mascons 166 and 124, Fig. 6A and Fig. S9A). In addition, simulation results indicate that SMB-driven dynamics affect ice velocities in the interior areas just upstream of the ablation zone. In these areas, marginal thinning steepens surface slopes (Fig. S9B), which

leads to larger driving stresses (Fig. S9C) and increases in local velocities (Fig. S8A). Overall, however, it is clear that the interior SMB-driven dynamics play a minor role in dictating total mass balance trends of the ISSM simulations. Indeed, within the interior region we find a close similarity between the cumulative mass change predicted by ISSM_GrIS and SMB_GrIS (Fig. 3B and 6).

### 6.4 Contribution from Periphery

Another significant component of the mass signal is the contribution from peripheral glaciers. In Fig. 7, we plot the spatial contribution of Greenland's periphery on trend and amplitude. We find that inclusion of the periphery contributes negatively to the trend (Fig. 7A), particularly in the Southwest, in Mascon 33 (i.e. Ellesmere Island) and in Mascon 167, but it contributes positively to the amplitude of the annual signal (Fig. 7B). Increased mass gain in the winter is driven largely by seasonal snow load on tundra, while summer melt of peripheral land ice dominates the signal and contributes to the overall negative trend.

We estimate that the peripheral glaciers are responsible for a total trend of $-37\pm25$ Gt/yr (Table 1), which agrees well with Gardner et al. (2013). However, it is important to note that this estimate is associated with large uncertainty (Fig. 3A - difference between red and blue shading). In fact, while the inclusion of the periphery glaciers allows us to account for a part of the discrepancy between GRACE_JPL and ISSM_GrIS, doing so also increases the estimated uncertainty of the model results (Fig. 3A). Comparison between the ISSM_GrIS+P RACMO, ISSM_GrIS+P MAR, and ISSM_GrIS+P BOX trends reveals

that indeed, there are substantial differences in trend between periphery estimates (Fig. S10 and Table 1). The discrepancy between the models is particularly large in the southern mascons. In the Southeast, where slopes and gradients in SMB are large along the ice sheet margins, there is inconsistency between the RCMs, even in the sign of the trend. Analysis of the periphery in RACMO reveals slightly positive trends in the south, particularly in the Southeast. Likely, this is partially due to the lower resolution of the RACMO product (which is not downscaled to a higher resolution in post-processing, like the

other two RCM products). The lower resolution leads to difficulty in resolving the ice margin near the coast. Comparison of mass trends in GRACE_JPL with annual altimetry estimates (which do not include periphery) from 2003-2009, offers an observational estimate of peripheral mass trend during the first portion of the GRACE record (Fig. S6). Results suggest that peripheral estimates of MB trend from MAR have the best overall agreement with those observed in the south, while BOX has

the best overall agreement in the north.

## 6.5   Seasonal Variability

Full timeseries and mean seasonal cycle of mass change estimated by GRACE_JPL and by ISSM_GrIS+P are compared in individual non-interior mascons in Figs. 8 - 11. The mascons are organized geographically, by ice sheet region (i.e. Northeast, Southeast, Southwest, and Northwest). In some cases, mascons contain small fractions of the ice sheet margin. In these cases,

mascons are combined with a neighboring mascon. We include timeseries for all individual mascons in the Supplement, Figs. S11 - S15.

    In the majority of the mascons, ISSM_GrIS+P captures the seasonal cycle observed by GRACE_JPL. The largest discrepancies between ISSM_GrIS+P and GRACE_JPL occur during the summer. During this time, we also find that there is the largest disagreement between the ISSM_GrIS+P runs. These results suggest, in agreement with Velicogna et al. (2014), that

uncertainty in estimates of runoff within the SMB products are largely responsible for driving diverging uncertainty through time (most notably in the Southwest, Fig. 8). Consistent error in RCM estimates of runoff may also be partially responsible for the trend differences between ISSM_GrIS+P and GRACE_JPL, particularly in the mascons that agree well during the winter months but differ during the summer. Overall, it is difficult to pinpoint a consistent bias that is associated with a particular region or SMB model. Specifically, when comparing these products at the spatial scale of a mascon ($\sim$300 km), discrimination

of the sources of uncertainty becomes more complicated.

    In the Northeast, for instance, GRACE_JPL and ISSM_GrIS+P agree well in overall trend for mascons 59, and 89+90, while the annual cycle in 89+90 is consistent with GRACE_JPL estimates (Fig. 9). However, the annual cycle for Mascon 59 in ISSM_GrIS+P is more exaggerated than GRACE_JPL, with greater accumulation during the winter months and greater mass loss during the summer months. For Mascon 35 and 125+126, we find that GRACE_JPL has a more negative trend

than ISSM_GrIS+P. In Mascon 35, this is due to an ISSM_GrIS+P underestimation of mass loss in the summer relative to GRACE_JPL, while in 125+126, it is due to an ISSM_GrIS+P overestimation of mass gain during the winter relative to GRACE_JPL (Fig. 9). In the Southwest, such a discrepancy in trend for Mascon 165 is due to a combination of the two (Fig. 8), with the spread between ISSM_GrIS+P and GRACE_JPL increasing non-linearly through time (Fig. S12). Mascons 212 and 265 have more negative trends in ISSM_GrIS+P than GRACE_JPL, but agree very well in the seasonal cycle. This area

is well covered by observations (including the K-transect), and often RCMs are evaluated in this area. Here, we find that the differences are due to a higher estimate of runoff during the summer months (Fig. 8), predominantly causing a divergence between ISSM_GrIS+P and GRACE_JPL from 2003-2009 (Fig. S12). Finally, for Mascon 324, we find that the ISSM_GrIS+P estimates have a large spread in trend, largely due to uncertainty in periphery estimates (Fig. S10). The periphery estimate for Mascon 324 also contributes to an exaggeration of the annual amplitude for this mascon (Fig. 7), resulting in a perceived

overestimate of mass gain in the winter and mass loss in the summer (Fig. 8). While errors in SMB forcing in these regions likely play the dominant role in differences between GRACE_JPL and ISSM_GrIS+P, there is evidence that missing model processes may also play a role. Specifically, during the winter, GRACE_JPL captures month-to-month variability that is beyond the spread of the three ISSM_GrIS+P runs (e.g., Mascons 165 and 324). In comparison, the ISSM_GrIS+P results are smooth and do not exhibit the same type of variability.

Such variability is also present in the Southeast (Fig. 10) and Northwest (Fig. 11) sectors, where the majority of mascons show a significant discrepancy in trend between GRACE_JPL and ISSM_GrIS+P. Mascons 213 and 33+34 are clear exceptions, and match well in both trend and annual amplitude. However, note that for Mascon 213, ISSM_GrIS+P and GRACE_JPL agree prior to 2010 and then continuously diverge through 2012 (Fig. S14). These results agree with observations of velocity for the region, in particular the acceleration of Ikativaq region in 2009 and of Helheim Glacier in 2010 (Joughin et al., 2008; Moon et al., 2012; Khan et al., 2014). The rest of the mascons in the Southeast and the Northwest, where changes in ice discharge are believed to play a large role in recent mass changes, have GRACE_JPL signals that show large negative trends. These negative trends are consistently underestimated by the ISSM_GrIS+P runs, even in the locations where the seasonal signal appears to be well captured (e.g., Fig. 10, Mascon 214 and Fig. 11, Mascon 56). In the Northwest, in particular, it is clear that in this region, it is not just the summer season that is responsible for the difference between GRACE_JPL and ISSM_GrIS+P. In Mascons 56, 86+87, and 123, we find a distinct difference between GRACE_JPL and ISSM_GrIS+P during the entire year (Fig. 11). In these cases during the winter, GRACE_JPL indicates that mascon regions continue to lose mass, while the SMB forcing (represented by the ISSM_GrIS+P runs) remains positive. In fact, for Mascons 56 and 86+87, the summer melt appears to be well represented by the models. For Mascon 123, we find that summer runoff is overestimated; yet the negative trend is simultaneously underestimated due to an overestimate of mass gain during the rest of the year.

## 7 Discussion

Based on our analysis of error and uncertainty, we assume that the majority of the difference in trend between ISSM_GrIS+P and GRACE_JPL (-101 $\pm$ 35 Gt/yr; Table 1) can be attributed to processes not included in the ice sheet model. This assumption would be consistent with recent studies (Moon et al., 2014) which report observed seasonal accelerations in local ice flow of magnitudes far larger (by a factor of 10) than the changes in ice velocity modeled by ISSM_GrIS over the ten-year simulation period (Fig. S8A). In some cases, we find evidence that errors in SMB, especially within the periphery, may also significantly contribute to these discrepancies. Below, we discuss the differences between ISSM_GrIS+P and GRACE_JPL for each region of the Greenland Ice Sheet. In addition, under the assumption that these differences represent dynamic mass changes not simulated by the models, we hypothesize about what this comparison may reveal about the temporal variability of dynamic mass change on a regional scale.

### 7.1 Northwest

Overall, results from this comparison suggest that largest discrepancies in mass trend between the model and GRACE_JPL are in the Northwest sector of Greenland. Here, such discrepancies are likely due to consistent ocean forcing, hydrology-driven events, errors in modeling the bedrock, or error in the ice model spinup. Mean annual plots of GRACE-measured mass change (e.g. Mascons 86/87, 123, and 56) reveal that the Northwest loses mass throughout the entire year, even during winter months (Fig. 11). Comparison between the mean annual cycles and GRACE_JPL indicate that it is an increase in ice discharge - not captured by the model - that dominates the mass trends here. Indeed, it is in this region where we find SMB plays less of a role in determining mass balance, particularly in areas where modeled mass and GRACE_JPL disagree outside of estimated uncertainty. Since SMB is positive during the winter, and the SMB products have strong agreement in this region during the fall, winter, and spring, increased ice discharge is most likely responsible for the strong discrepancies between GRACE_JPL and ISSM_GrIS+P during non-summer months. During these months, GRACE_JPL exhibits mass loss inconsistent with SMB, which suggests that the total mass in the Northwest is strongly out of balance (Reeh et al., 2001; MacGregor et al., 2016). This finding is supported by the model behavior during relaxation to steady-state. During relaxation, we find that many glaciers in the Northwest slow down in order to be in balance with the SMB forcing (Fig. S4B). These results suggest that our assumption of historical steady-state is likely invalid for the Northwest region of Greenland.

### 7.2 Southeast

In the Southeast, it is more difficult to pinpoint a particular factor that drives the differences between GRACE_JPL and the model estimates of mass change. Mean seasonal plots (Fig. 10) of the mascons in this area reveal that the GRACE_JPL signal exhibits larger seasonal variations than estimated by the models. This suggests that discrepancies may be controlled by errors in modeled SMB, including errors in mass contribution from the periphery (i.e., trend from glaciers and annual signal from load on bare rock and tundra). The topography in the Southeast is steep, mountainous, and generally plagued by the largest uncertainties in modeled snowfall estimates, yet we find that the SMB products represented here tend to agree well with GRACE_JPL during the majority of the year. The largest discrepancies with GRACE_JPL occur during the summer months, which also happens to be when the discrepancies between the SMB forcing products are the largest. Such results suggest that SMB errors may contribute to model uncertainty in the Southeast, and RCM estimates of runoff for both the ice sheet and periphery glaciers may not be accurately captured. This may particularly be the case in Mascon 266 (Fig. 10), where the steep terrain creates a very narrow ablation zone that is difficult to capture at the resolution of the SMB forcing. In contrast with the other Southeast mascons, Mascon 266 exhibits poor agreement in annual amplitude. In this mascon, we find that a consistent annual discrepancy between GRACE_JPL and the model estimates of mass loss occurs almost exclusively during the summer months (Fig. 10), suggesting that a seasonal phenomenon may be responsible for mass loss in this region. It is important to note that according to observational evidence most glaciers within this mascon are characterized by a summer slowdown, not an acceleration (Moon et al., 2014). As a result, we find that seasonal discrepancies in the Southeast are at least partially

rooted in errors in SMB forcing, specifically errors in runoff, including those associated with modeling melt-water retention and refreeze.

In this region, there is also evidence that, in agreement with recent publications, (e.g., Csatho et al., 2014; Khan et al., 2014; Moon et al., 2012, 2014; Velicogna et al., 2014), temporally varying processes (not captured by the ice sheet model) play a role by altering ice discharge. For instance, mass balance within the Helheim and Ikativaq region (Mascon 213) is well captured by the model overall, but it is clear that GRACE_JPL and the models differ in trend between 2005 and 2006 and then again in 2010 (Fig. S14). This discrepancy is consistent with observations of high velocities in Helheim in 2005, followed by a slowdown, and then acceleration in Ikativaq in 2009 and Helheim in 2010 (Joughin et al., 2008; Moon et al., 2012; Khan et al., 2014; Csatho et al., 2014). Similarly, a well-documented shift in ice discharge is captured by GRACE_JPL at Kangerdlugssuaq Glacier (Mascon 167) in 2005 (Fig. S14). Observational evidence suggests that such changes in sensitive tidewater glaciers are strongly coupled to calving events and the position of the glacier terminus, especially during periods of rapid advancement during the spring and early summer (Joughin et al., 2008). This is consistent with the behavior of the mean GRACE_JPL seasonal signal in most of the Southeast mascons (i.e. 167/168, 213, and 214), which appear to have a much noisier signal during the spring than is simulated by the models (Fig. 10), including single months of high mass loss. These results suggest that the GRACE_JPL solution is capable of capturing monthly-scale changes in ice discharge within large outlet glaciers, and therefore it may be possible to quantify dynamic mass loss by removing the ISSM_GrIS+P from the GRACE_JPL signal. However, it is clear that with regards to the seasonal cycle, where model results fall within the GRACE_JPL range of uncertainty, we cannot confidently distinguish between errors in SMB and high-frequency (monthly-scale) changes in ice discharge. This is the case in many of the mascons, particularly in the Southeast, with the exception of Mascon 266 where (as discussed above) we can confidently conclude that SMB is a significant contributor to the disagreement between trends in GRACE_JPL and ISSM_GrIS+P. Overall, in the Southeast, discrepancies are likely caused by a combination of errors including lack of ocean forcing, poor bedrock, inadequate mesh representation of the smaller and steeper glaciers in ISSM, as well as uncertainty in SMB forcing due to the steep terrain and narrow ablation zone.

## 7.3 Northeast

In the Northeast, we find overall good agreement between the models and GRACE_JPL in both amplitude and trend (Figs. 4, 5, 8, and 9). The Northeast Greenland Ice Stream (Mascon 59) is well captured in trend, though we find that the annual amplitude of the GRACE_JPL signal is highly muted, particularly during the summer. Such a discrepancy could be caused by common mode errors in the SMB forcing, but the match in trend suggests that unmodeled hydrological processes may be responsible for this discrepancy (e.g. storage and delayed release of runoff) (Willis et al., 2015). Reconciling these results with observations of ice elevation (i.e. altimetry measurements), when available at a monthly to seasonal temporal resolution, could shed light on the key processes responsible for continued mass loss in this area.

## 7.4 Southwest

In the SMB-dominated Southwest region, our results also capture signals that may be related to temporal changes in ice discharge, despite the fact that most of the glaciers are land-terminating and the position of the glacier termini are not affected by ice-ocean interaction (Khan et al., 2015). Specifically, we find that in this region, the relationship between SMB and mass change is not consistent through time. For instance, the model and GRACE_JPL disagree for Mascons 212 and 265 between 2005 and 2010, but then agree well for the remainder of the simulation (Fig. S12). In contrast, Mascon 165 (i.e. Jakobshavn Isbræ) has good agreement between the model and GRACE_JPL at the beginning of the simulation, but the signals begin to disagree around 2008 (Fig. S12). In fact, the mass loss in Jakobshavn Isbræ appears to be accelerating through time. These results are consistent with published observations of minor speedups in velocity for Jakobshavn Isbræ beginning in 2008, as well as a general velocity decrease in the Southwest between 2005 and 2010 (Moon et al., 2012; Tedstone et al., 2015) (corresponding to Mascons 212 and 265). The SMB in this area is well validated (i.e. K-transect), and annual amplitudes agree well. Therefore, it is likely that temporal variability in ice discharge, driven by processes not modeled here, contributes to the disagreement in trend between GRACE_JPL and the model estimates of mass loss. The monthly-scale variations in regional mass loss evidenced in the GRACE_JPL seasonal cycle (Fig. 8) is most likely driven by changes in ice discharge within the few, but active, marine-terminating glaciers in the region, including the effects of hydrology and ice-ocean interaction (calving events/position of glacier terminus) (Holland et al., 2008). It is clear that over the course of just a decade, consistent ice flow response to these types of climate-driven forcing can ultimately perturb regional mass trends, even in the regions where mass loss appears to be dominated by SMB.

## 7.5 Interior

Though the majority of the GRACE_JPL interior mascons exhibit possible background dynamic thickening, it is difficult to explicitly quantify the effects of millennial-scale forcing within all interior mascons, as trends are not consistent throughout the 10-year study period, and the GRACE_JPL signals in the interior are strongly convolved with large inter-annual variability. For instance, in Southern Greenland, observed thickening is often attributed to the downward displacement of less viscous ice from the last glacial period with more viscous Holocene ice (Reeh, 1985; Huybrechts, 1994; Colgan et al., 2015). Due to the placement of the GRACE_JPL mascons, only one interior mascon (Mascon 166), is located within this region. While we do detect a positive difference in trend between the GRACE_JPL and the models in this mascon (Fig. 4 and Fig. S11), we cannot confidently conclude that a dynamic thickening is responsible for the discrepancy. Indeed, we would expect millennial-scale dynamics to contribute a relatively constant perturbation in trend over the study period, but we find this to be the case only within the Northeast interior (Fig. S11; Mascons 58 and 88). Here, the background dynamic thickening signal is likely a millennial-scale response resulting from ice deceleration, recently attributed to a modern-day decrease in accumulation in comparison to the average Holocene accumulation rates (MacGregor et al., 2016). We estimate that within the Northeast (Mascons 58 and 88) this thickening is occuring at a rate of about 2.5 cm/yr, which is consistent with other observationally-based estimates (Krabill et al., 2000; Paterson and Reeh, 2001).

In general, for the interior we find periodic disagreement between GRACE_JPL and models, outside of the assessed uncertainty (Fig. 3b). One possible explanation (besides dynamic processes not captured in ISSM) is that the RCMs - which are commonly forced by ERA-I and agree well in the interior - are not capturing stochastic accumulation events that occur during roughly the same time every year. However, evidence suggests that this is not the case; in particular, analysis reveals that

the MAR3.5.2 product forced with NCEP/NCAR Reanalysis 1 (Kalnay et al., 1996) exhibits similar temporal variability (not shown) and annual amplitude (Fig. S2A) to the ERA-I SMB products considered here. A more likely explanation is that the discrepancy is caused by a combination of unmodeled dynamic thickening, and noise in a locally small signal in GRACE_JPL coupled with modest leakage errors from neighboring coastal mascons with much stronger signals (not considered in our GRACE uncertainty analysis). If so, these results indicate that mass signals in the high altitude interior of Greenland are suf-

ficiently small enough to push the limits of GRACE utility for model evaluation, both temporally and spatially. The use of GRACE in this area is additionally complicated by a GIA correction that is significant in comparison to the magnitude of the GRACE signal. We expect that advances in GRACE mascon processing, GIA modeling, and progressions in RCM estimates of SMB (including improved validation in the interior using satellite data and data from a growing network of in-situ stations, and the diversification of RCM forcing products) may, in the near future, help clarify these discrepancies.

**7.6    Model Assessment**

It is important to acknowledge that upon relaxing the model using a historical period of neutrality in ice sheet total mass balance, the spin-up procedure adopted for this study assumes that the Greenland Ice Sheet was in near steady-state during the recent past. More specifically, we relax the model to a steady-state condition, using a mean climate forcing from the 1979-1988 period - a period in which the rate of ice sheet mass loss was negligible compared to the mass loss captured by GRACE

during the last decade. We adopt this procedure in order to remove spurious transients from the model that may manifest due to mismatched input including: bedrock and ice surface elevation maps, surface ice velocities, and SMB. After relaxation, ISSM_GrIS discharge is nearly equal to the mean SMB forcing, and resultant perturbations to ice thickness or velocity that occur in the forward model are solely in response to anomalies in the transient SMB forcing starting in 1840.

We acknowledge that these assumptions may result in differences in modeled and observed ice thicknesses (Fig. S5), and

in turn may cause the model to exhibit second-order deviations in ice velocities. This is especially the case considering that - even though the SMB products have been validated against observations - these observations are sparse, and all SMB products are associated with systematic errors that may impact spinup and propagate into the simulation. However, it is clear that in the absence of a long-term model spin-up (on the order of thousands of years), the assumption of steady-state is adequate for short term (annual-to-decadal scale) simulations. Indeed, we find that our results are fairly insensitive to the SMB product chosen as

forcing during relaxation (Figs. S2B and S3B). These results suggest that - on the temporal and spatial scales considered here - background trends in mass balance (that occur in response to paleo-climate forcing) may play a minor role in dictating present-day evolution of Greenland MB when compared to SMB anomalies and seasonal-to-annual scale variations in ice velocities (Csatho et al., 2014). In the marginal mascons, with the exception of the Northwest (Mascon 123), significant background trends (which would manifest as continued mass loss during the months of accumulation in the seasonal cycle, i.e. Figs. 8-11)

are not detectable outside of our assessed uncertainty. In addition, since the regions that are currently in the strongest imbalance are also affected by seasonal- to annual-scale variability in ice discharge, we cannot quantify the magnitude of the background trends in dynamics in these areas. In the interior, where we find that the SMB models agree well, the comparison presented here offers an opportunity to quantify background dynamic trends, despite the complexities in the variability of the GRACE_JPL signal discussed above. In particular, we observe that the interior has gained mass throughout study period, in agreement with other observations (Fig. 3B) (Krabill et al., 2000; Paterson and Reeh, 2001; Csatho et al., 2014; MacGregor et al., 2016).

Though it is clear that SMB accounts for the majority of Greenland mass balance, our results indicate that consistent intra-annual variations, not explained by SMB, can accumulate over time and contribute significant regional trends in mass balance. These variations are likely driven by the evolution of the hydrological system and ice-ocean interactions, which are believed to be responsible for monthly-scale perturbations in ice velocity in major outlet glaciers in Greenland (e.g., Csatho et al., 2014; Khan et al., 2014; Moon et al., 2012, 2014). Continued advancement in physically-based model representations of these processes promise to improve ice sheet model skill for decadal-scale simulations (Nick et al., 2010; Bartholomew et al., 2012; Yoshimori and Abe-Ouchi, 2012; Carr et al., 2013; Joughin et al., 2013; Schild and Hamilton, 2013). Such model improvements are difficult on a continental scale, because these processes are not universally well understood. In addition, they are associated with large uncertainties. However, in order for the glaciological community to take full advantage of the array of new observational products that are available (and will be made available) for model evaluation in the near future, it is essential that simulations consider how such temporally evolving processes affect the variability and overall trend in total Greenland MB. The future success of Greenland ice sheet model simulations, including hindcasts as well as future projections of ice sheet mass balance and sea level change, will require high confidence in SMB forcing and the incorporation of accurate representations of key processes that vary on intra-annual to seasonal timescales.

## 8  Conclusions

In a mascon-by-mascon comparison of model estimates of Greenland Ice Sheet mass with the GRACE_JPL mascon solution, we investigate the differences between average trends and seasonal amplitudes with respect to uncertainties in each. Model estimates are based on the mean output of three ISSM_GrIS simulations from 2003-2012, each forced with anomalies from a different RCM-based SMB product. Overall, the largest discrepancies between GRACE_JPL and the model-based estimates of mass balance are located in the Northwest and Southeast. In the Northwest, though we find that the seasonal amplitudes agree well between the two products, it is clear that the models vastly underestimate the regional mass trends captured by GRACE_JPL. This result suggests that changes in ice discharge, not captured by ISSM_GrIS, are largely responsible for the considerable discrepancy; and, that the glaciers in the Northwest coast of Greenland are strongly out of balance. In the Southeast, large uncertainty ranges prevent us from differentiating which factors are most responsible for differences in trend, but results suggest that it is likely a combination of processes that alter ice discharge at a relatively high (monthly) temporal frequency (not represented by our ice sheet model) and errors in SMB forcing (dominated by discrepancies in summer surface runoff and errors along the periphery). Inaccuracies in this area are rooted in the coarse spatial resolutions of the ISSM_GrIS

and the RCMs, as the regional terrain is steep, complex, and difficult for models to resolve. In the high-altitude interior of Greenland, the mass signal is dominated by snow accumulation. Here, we find evidence of background dynamic thickening, particularly in the Northeast, albeit the utility of using GRACE data for model validation in this region remains challenging due to the level of noise present in GRACE relative to the small signal size. In the other marginal regions of the ice sheet (i.e. the Southwest and Northeast), we find strong agreement in both amplitude and trend, suggesting (in agreement with recent publications), that mass balance is dominated by SMB in these areas. By and large, we find that SMB is a significant source of mass variability over the majority of the ice sheet. Future improvements in RCM resolution, snowpack models for tundra regions, and simulation of climate over the peripheral glaciers and ice caps will be essential for future comparisons and validation against seasonal-scale mascon-style GRACE products.

Overall, throughout the simulation period, we find ISSM_GrIS responds to the SMB forcing (dominated by increases in surface runoff) with marginal thinning. This thinning is accompanied by increases in interior velocity, dampening of the annual total mass balance signal, and overall reduction of ice discharge. While over longer periods the ice sheet response to changes in SMB may contribute more significantly to ice sheet total MB, over the observational period analyzed in this study we find that such responses are minor in comparison to the direct contribution from the SMB forcing itself. In many cases, we find that errors in SMB forcing may be directly responsible for differences between the models and GRACE_JPL, especially in the periphery, however temporally-varying processes missing from the ice sheet model - including the effects of supra- and en- glacial hydrology, ice-ocean interactions, and calving events - are also known to affect ice discharge on intra-annual timescales. Therefore we consider these processes to be strong candidates for those that may be responsible for the high-frequency discrepancies exposed in this study. Future progress in observing these processes (including increased spatial and temporal resolution) and future improvements in the physical modeling of their effects on ice sheet flow, will be necessary to confidently partition Greenland MB into its key attributes. Such advancements promise to improve the skill of physically-based ice sheet models, as accurate estimates of Greenland MB may require consideration of processes that occur on high (monthly-to-seasonal) temporal resolutions.

## 9 Appendix

### 1 Description of the BOX reconstruction

BOX (Box, 2013) is based on calibration of observational data to regional climate model (RCM) output, in this case RACMO2.3 (van Meijgaard et al., 2008; Ettema et al., 2009; van den Broeke et al., 2009; van Angelen et al., 2011). The calibration for temperature (T) and SMB components is based on a 53-year overlap period (1960-2012). Note that the overlap period for the calibration of snow accumulation rate is shorter, since ice core data availability drops after 1999. Calibration is made using linear regression coefficients for 5 km grid cells that match the average of the reconstruction to RACMO2.3. The RACMO2.3 output are resampled and reprojected from the native 0.1 deg ($\sim$10 km) grid to a 5 km grid better resolving areas where sharp gradients occur, especially near the ice margin where mass fluxes are largest.

To create the BOX SMB forcing used here, several refinements are made to the Box (2013) T and SMB reconstruction. Multiple station records now contribute to the near surface air temperature for each given year, month and grid cell in the domain, while in Box (2013) data from the single highest correlating station yielded the reconstructed value. The estimation of values is made for a domain that includes land, sea, and ice, which is an expansion to the Box (2013) product that estimates
T over ice only. A physically-based meltwater retention scheme (Pfeffer et al., 1990, 1991) replaces the simpler approach used by Box (2013). The RACMO2.3 output have a higher native resolution of 11 km as compared to the 24 km Polar MM5 output used by Box (2013) for air temperatures. In addition, the revised SMB product ends two years later, in year 2012. The annual accumulation rates from ice cores are dispersed into a monthly temporal resolution by weighting the monthly fraction of the annual total for each grid cell in the domain evaluated using 1960-2012 RACMO2.3 output.

**2   Methods for defining peripheral SMB**

For this study, we define peripheral ice as isolated permanent ice that exists outside of the ISSM Greenland domain (see Fig. 1). A land-ice-ocean mask accompanies all the SMB products considered here. The masks differ for each product; therefore we must interpret them independently, with reference to the specific mask defined for a particular product. The 5-11 km resolution of the products, in many cases, do not properly represent the aerial extent or the topographical features of the peripheral ice. In
order to better capture the SMB within these complex areas and to more easily compare their mass balance estimates, we use the 150 m gridded GIMP Digital Elevation Model (DEM) (Howat et al., 2014; Morlighem et al., 2014b) to, separately for each mascon, determine a hypsometric curve for the areas masked by Gardner et al. (2013) as peripheral ice. The curve is binned at every 150 m of surface elevation. For every month, and for each mascon, we plot the SMB of each product separately as a function of elevation, and fit a curve (Gardner et al., 2013). Only SMB values over peripheral ice are considered in these
curves. Mascons with similar climates are combined in order to refine the fit. The resulting curve is used to determine the mean SMB value within each elevation bin. Finally, the SMB value within each bin is multiplied by the area of each elevation band, and the results for all elevation bands are summed as the total SMB mass contribution for a particular month.

For determining snow load outside of areas of permanent ice, we define peripheral tundra as area masked as land on Greenland or Ellesmere Island, within our Greenland mascons (see Fig. 1). Once again, because the product masks differ, we must
consider each mask independently. On all grid points that contain only fractional areas of tundra, the snow load is scaled to the percentage of the grid point covered by only land.

**3   Calculations of uncertainty in GRACE_JPL surface mass estimates**

To derive the GRACE_JPL error in each mascon, we use the formal posteriori covariance matrix from the gravity field inversion. Typically, the formal covariance matrix is regarded to provide an optimistic estimate of errors, as it is uninformed
of certain error sources that affect the GRACE_JPL mass estimates, such as temporal aliasing errors. We find that this is the case for ocean and land-hydrology regions of the world for which apriori information is derived from geophysical models: the posteriori covariance matrix is too optimistic and must be scaled up to accurately reflect uncertainty. However, for ice-covered regions, such as Greenland, the apriori information is derived from a bootstrapping methodology from which the magnitude

of the K-band range-rate data residuals dictate the spatial variations in the apriori covariance matrix, and the magnitude of these terms is purposefully left large, to be conservative. As such, we find the resulting posteriori covariance matrix to give an adequate estimate of uncertainty in each mascon. This hypothesis was tested by using spatial variance information from the MAR SMB model to derive an apriori covariance matrix that was used to constrain the GRACE_JPL solution, and analyzing how this impacted the results. Differences in this MAR-constrained solution vs. the relatively unconstrained solution presented here are captured by the formal errors. Furthermore, the posteriori covariance matrix shows adjacent mascons to have small correlations ($\sim$0.2) with each other. As such, we assume all mascons to be uncorrelated with their neighbors. A comparison to ICESat altimetry data (Csatho et al., 2014) validates this assumption (Fig. S6).

Leakage errors are considered explicitly by evaluating the expected accuracy of the CRI filter used to separate land and ocean mass components of mascons that lie on coastlines. Simulation results show the CRI filter is effective in reducing leakage errors by greater than 50% globally. Thus, we assume that the estimate of ocean mass for each land/ocean mascon has an error of 50%, and this error is added in a root sum of squares (RSS) to the formal covariance for each mascon.

Finally, since we are ultimately interested in surface mass variations (and as such remove solid Earth Glacial Isostatic Adjustment (GIA) signals using a model), the uncertainty in the GIA model must be considered. We take the 1-sigma spread of the ensemble mean of four GIA models. The four models used include the ICE-6G_C (VM5a) model (Peltier et al., 2015), a model by A et al. (2013) which uses ICE-5G loading history and a VM2 viscosity profile, a model using ICE-5G loading history and a Paulson viscosity profile (Paulson et al., 2007), and a model by Simpson et al. (2009) which uses the Huy1 (Huybrechts, 2002) ice load history and an independently derived viscosity profile. Using this approach, we derive a GIA uncertainty for the entire Greenland Ice Sheet of $\pm$15 Gt/yr, which matches closely with what is reported in Velicogna and Wahr (2013). The derived uncertainty in GIA is added to the RSS of the formal covariance and leakage error discussed above to arrive at an estimate of uncertainty in surface mass variations for each mascon. Note that in all figures, we show the GRACE_JPL surface mass uncertainty to be increasing linearly with time. This is directly due to uncertainty in the GIA model. Typical GRACE_JPL uncertainties are not presented in this fashion; however, since we are trying to compare surface mass variations directly, and identify regions that diverge outside of uncertainties, we present the uncertainty in GRACE-derived surface mass as one that grows linearly with time.

## 4 Details of ISSM spinup and forward simulation

For this study, we compare model-based monthly mass balance estimates of the Greenland Ice Sheet with the GRACE_JPL observational timeseries, on a mascon-by-mascon basis. The model estimates consist of ISSM Greenland simulations forced with anomalies from RCM-based SMB products. To spin up ISSM Greenland, we have used numerical techniques (e.g. assimilation of observations) to capture key features of the present-day ice sheet, including topography and surface velocities; however such a procedure and associated assumptions do have limitations (Sect. 3.2). To aid in assessment of our relaxation procedure, we plot the difference in velocity (Fig. S4B) and in ice thicknesses (Fig. S5B) between ISSM Greenland, and observed values (Rignot and Mouginot, 2012; Morlighem et al., 2014b), after relaxation. With respect to model velocities, the root-mean-square deviation between the ISSM relaxation surface velocities and observed surface velocities is 90 m/yr for the

continental ice sheet and 79 m/yr for the area of grounded ice considered in this study. Spatially, our comparison reveals that the modeled velocities are generally slower than those observed along the margins, especially in large outlet glaciers, including Jakobshavn Isbræ, Petermann Glacier, and the outlets of the Northeast Greenland Ice Stream (Fig. S4B). The smaller fast-flowing outlet glaciers on the Northwest coast also have velocities lower than observed, while in the Southeast, marginal ice speeds are greater than observed. With respect to ice thickness, we find that to reach a near steady-state, the model thins in the Northern interior and thickens in the Southern interior.

To illustrate how the ice sheet model responds to the historical SMB forcing, we include plots of the change in mean yearly ice velocity from 2003 to 2012 (Fig. S8A) and the change in ice thicknesses from 2003 to 2012 (Fig. S8B). With respect to ice velocities, the Southeast glaciers which were generally faster than observed after relaxation, continuously slow down over the ten years of simulation (Fig. S8A). We find that in general throughout the study period, modeled ice velocities along the margins are slowing, while interior ice velocities are accelerating. Accompanying these velocity changes are general thinning along the margins and minor thinning in the interior (Fig. S8B). Overall, the model ice thickness changes are dominated by marginal thinning (Fig. S8B) and are driven strongly by a decrease in SMB during the GRACE period (Fig. 2). For instance, we plot the difference between the ISSM ice thickness changes during the study period and the SMB contribution to ice thickness changes in Fig. S9A. We find that along the margins, and especially along the Southeast coast, where velocities slow down throughout the study period (Fig. S8A), the model contributes to ice thickening. Just upstream from the margins, where velocities increase during the study period, the model contributes to thinning (Fig. S9A).

*Acknowledgements.* This work was performed at the California Institute of Technology's Jet Propulsion Laboratory under a contract with the National Aeronautics and Space Administration's Cryosphere Program. The contribution from J. E. Box was supported by Geocenter Denmark. The authors would like to acknowledge the data provided by the National Snow and Ice Data Center DAAC, University of Colorado, Boulder, CO, Operation IceBridge, as well as CReSIS data generated from NSF grant ANT-0424589 and NASA grant NNX10AT68G (Gogineni, 2012). This work was made possible through model development of the ISSM team, including invaluable guidance in model setup by Dr. Helene Seroussi and incorporation of the most recent BedMachine bedmap of Greenland provided by Dr. Mathieu Morlighem. The authors would also like to thank Dr. Alex Gardner for his invaluable contributions - including discussion and advice pertaining to the periphery; GRACE_JPL team members, in particular Dr. Carmen Boening and Dr. Isabella Velicogna, for their support and advice with respect to interpretation of the GRACE solution; and Dr. Beata Csatho for sharing results of altimetrically-derived trends over the Greenland Ice Sheet. Finally, the authors would like to extend gratitude towards three anonymous referees for their helpful comments and discussions pertaining to this manuscript.

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

**Figures**

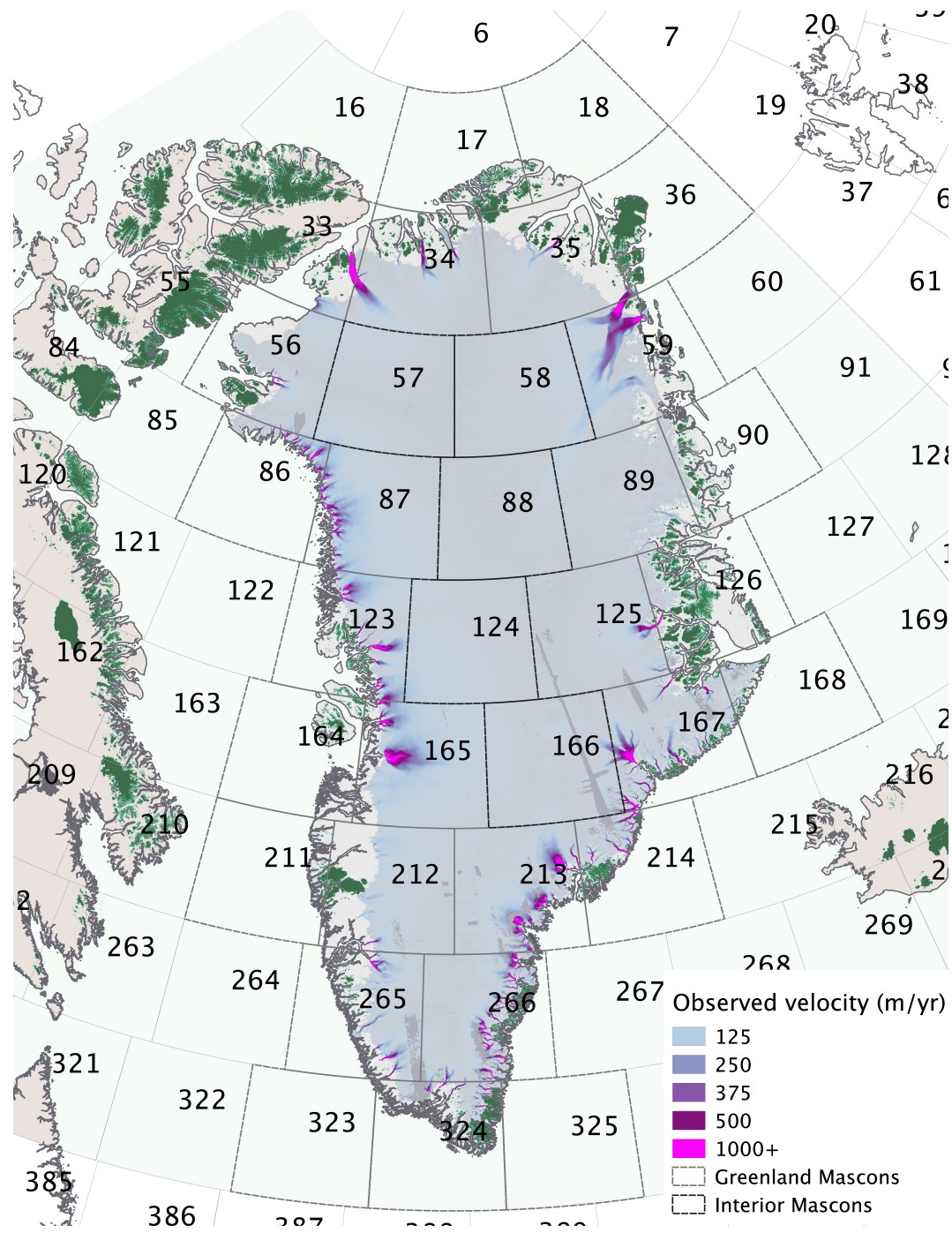

**Figure 1.** Greenland Ice Sheet (gray) with an overlay of observed surface velocities (Rignot and Mouginot, 2012), peripheral tundra (beige), and peripheral permanent ice (green) (Gardner et al., 2013). GRACE_JPL mascons are outlined in light gray and numbered for reference. Marginal (Interior) mascons that contribute to total Greenland Ice Sheet mass balance are outlined in dark gray (black).

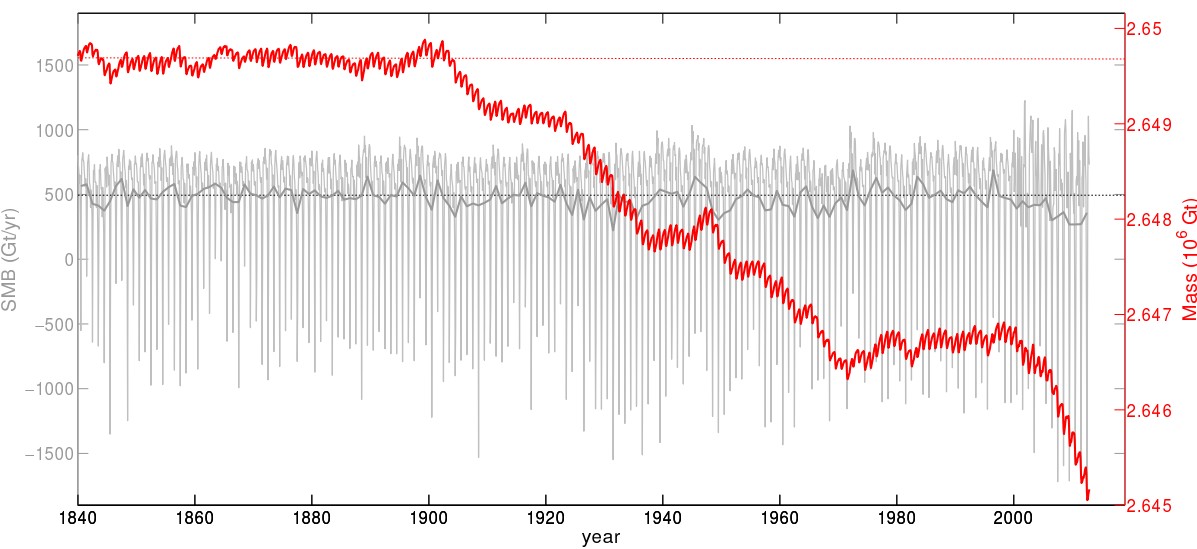

**Figure 2.** Timeseries of total ice sheet mass resulting from the Greenland ice sheet model historical spinup (solid red), compared to a control run (dashed red) of constant SMB climatological forcing, $\overline{SMB}$ (dashed black). For the historical simulation, ISSM is forced with monthly surface mass balance anomalies from 1840-2012 (Box, 2013). Monthly ice sheet total SMB forcing is plotted in light gray and yearly total SMB is presented in dark gray.

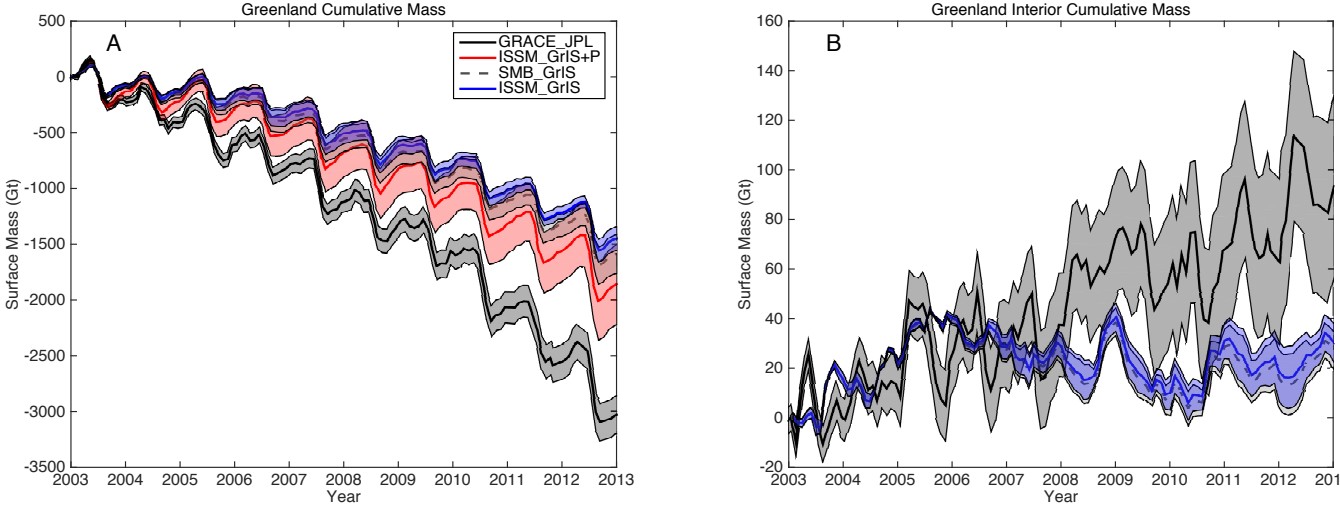

**Figure 3.** Cumulative mass from 2003-2012 for (A) all of Greenland and (B) the Greenland Interior, comparing observations from GRACE (GRACE_JPL) with model output: mean of the model simulations of the Greenland Ice Sheet (ISSM_GrIS), ISSM_GrIS with mass from the periphery (ISSM_GrIS+P), and the mean of the SMB anomalies over the Greenland Ice Sheet (SMB_GrIS). For all timeseries, 1-sigma uncertainties (see Sect. 5) are displayed. Note the differences in scale between the two figure panels.

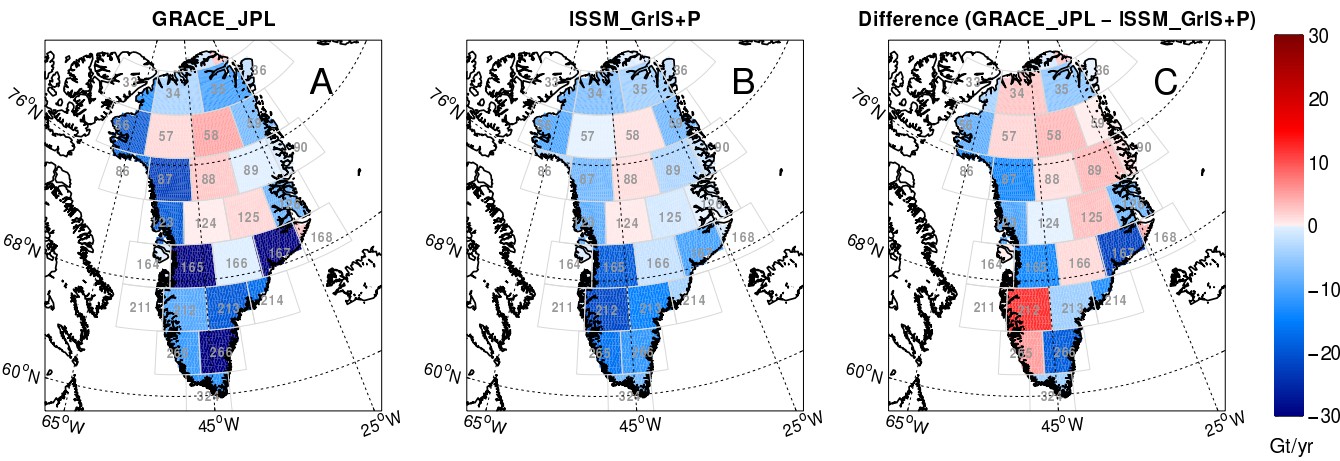

**Figure 4.** Spatial representation of trend in surface mass over Greenland from (A) GRACE_JPL, (B) ISSM_GrIS+P, and (C) the difference between GRACE_JPL and ISSM_GrIS+P.

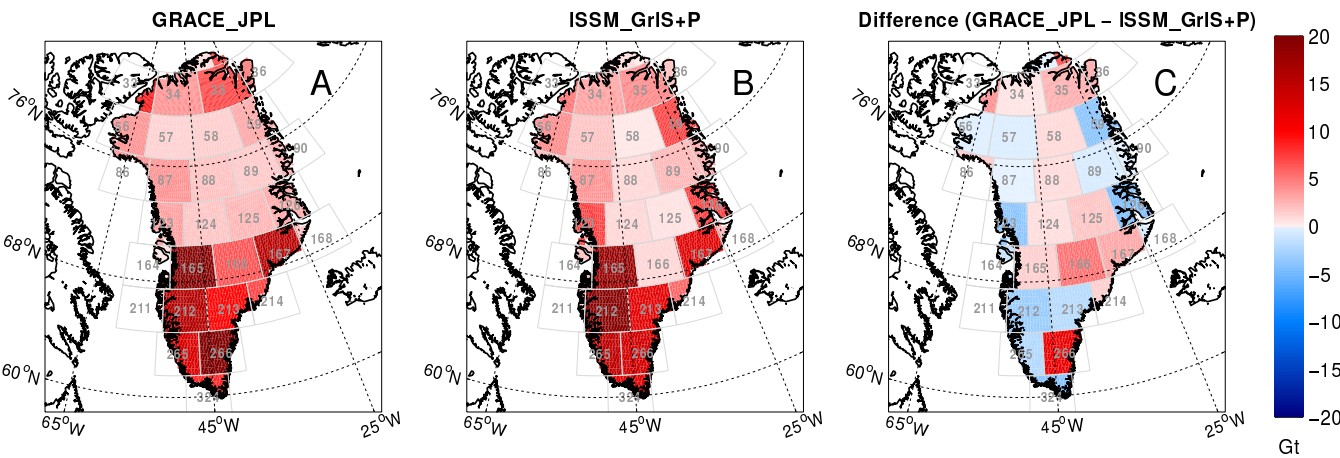

**Figure 5.** Spatial representation of annual amplitude of surface mass over Greenland from (A) GRACE_JPL, (B) ISSM_GrIS+P, and (C) the difference between GRACE_JPL and ISSM_GrIS+P.

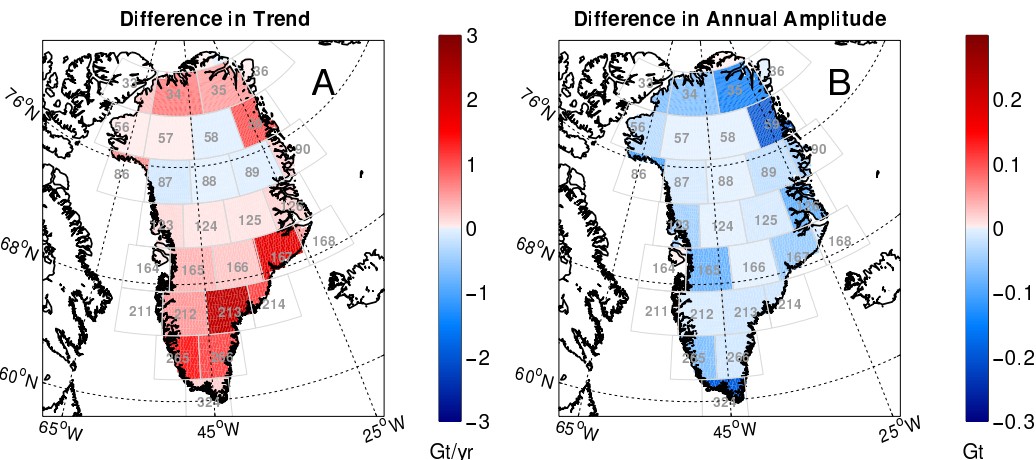

**Figure 6.** Difference in (A) trend and (B) annual amplitude of (ISSM_GrIS - SMB_GrIS) showing the contribution from SMB-driven dynamics, as calculated by ISSM, to both the trend and annual amplitude. Here, we define SMB-driven dynamics as the difference between the ISSM-simulated mass balance and the mass balance predicted by SMB anomalies alone. A spatial representation of the difference between ISSM_GrIS and SMB_GrIS ice thickness change over the study period (plotted on the ISSM mesh) is presented in Fig. S9.

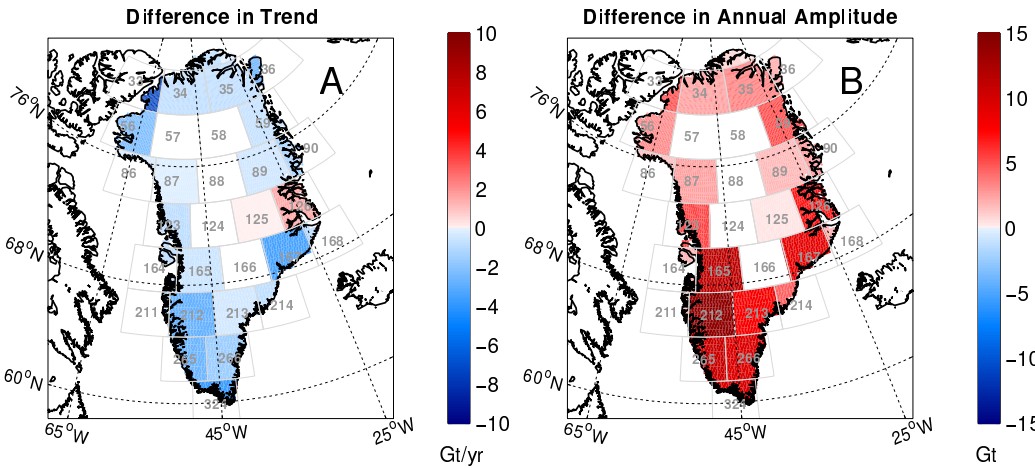

**Figure 7.** Difference in (A) trend and (B) annual amplitude of (ISSM_GrIS+P - ISSM_GrIS) showing that including estimates of mass from the periphery both increases the magnitude of the annual amplitude and the negative trend of surface mass.

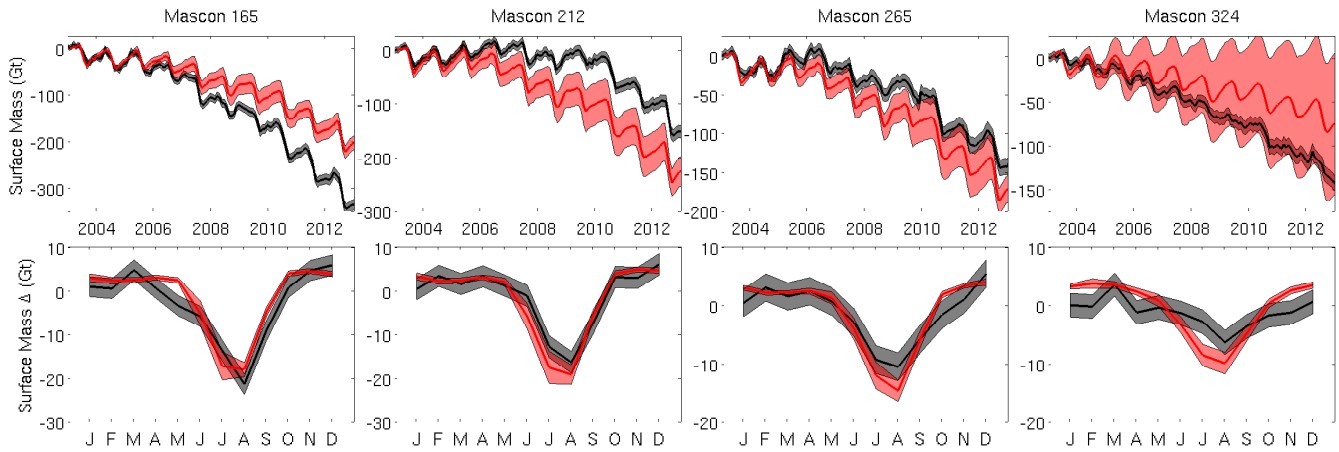

**Figure 8.** Southwest Greenland (top row) cumulative surface mass and (bottom row) a climatology of surface mass change comparing GRACE_JPL (black) and ISSM_GrIS+P (red) with 1-sigma uncertainties displayed. (See Sect. 5 for details on calculation of errors and uncertainties.)

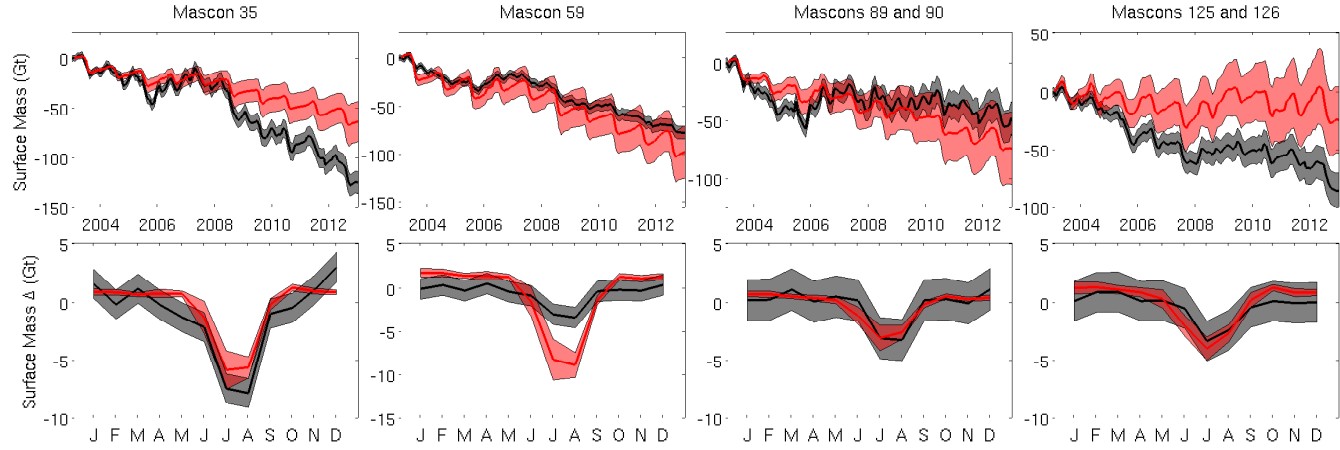

**Figure 9.** Same as Fig. 8, but for Northeast Greenland.

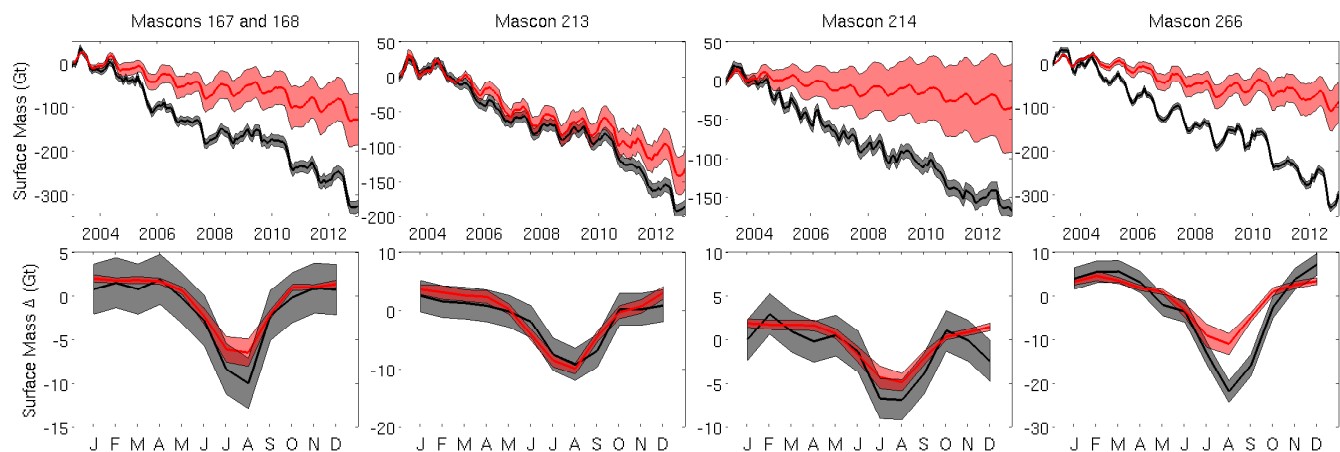

**Figure 10.** Same as Fig. 8, but for Southeast Greenland.

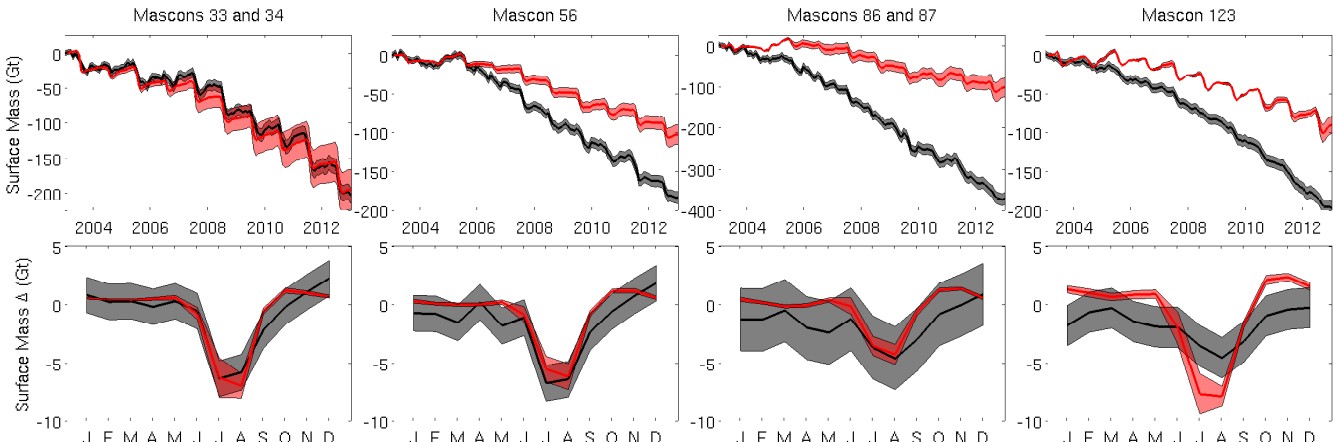

**Figure 11.** Same as Fig. 8, but for Northwest Greenland.

| Product | Cumulative Mass Trend (Gt/yr) | | | |
|:---:|:---:|:---:|:---:|:---:|
| **GRACE_JPL** | | | | - 284±19 |
| | *RACMO2.3* | *MAR3.5.2* | *BOX* | Mean |
| **SMB_GrIS** | -150 | -171 | -158 | -160±11 |
| **ISSM_GrIS** | -140 | -150 | -149 | -146±5 |
| **ISSM_GrIS+P** | -155 | -213 | -180 | -183±29 |
| **ISSM contribution to trend** | 10 | 21 | 10 | 14±6 |
| **Periphery contribution to trend** | -14 | -63 | -32 | -37±25 |

**Table 1.** Cumulative mass trends (Gt/yr) for anomalies of the individual RCM SMB products (SMB_GrIS), ISSM forced with each individual RCM SMB product (ISSM_GrIS), and ISSM_GrIS plus cumulative mass estimates for the ice sheet periphery (ISSM_GrIS+P) are presented in the left three columns. Presented in the right column are the mean cumulative mass trend and trend uncertainty for each timeseries (see also Fig. 3A). Also reported for each of these columns are the total ISSM contribution to the ISSM_GrIS+P trend (the SMB-driven dynamics predicted by ISSM, calculated as the difference between SMB_GrIS and ISSM_GrIS) and the total Periphery contribution to the ISSM_GrIS+P trend (the sum of SMB calculated over peripheral permanent ice defined in Fig. 1). Along with the trends, 1-sigma uncertainties (see Sect. 5) are displayed.