# Peer review of "Application of GRACE to the assessment of model-based estimates of monthly Greenland Ice Sheet mass balance (2003-2012)"

_The Cryosphere, 2015_

## Referee Comment (RC1) · Anonymous Referee #1 · 19 Feb 2016

Introduction:

Schlegel and co-authors present a methodological paper that deals with the important question how an ice flow model (here of the Greenland ice sheet) could be validated with observational data. Measured changes in the gravitational field as recovered from the GRACE satellite mission are utilised to estimate mass trends on a relatively small regional scale (300 km) and are compared with mass changes derived from a combination of three SMB models with an ice flow model (ISSM). The ice flow model is initialised to a steady state with the average SMB from the Box model for the period 1979-1988, by first inverting for observed velocities and than relaxing the geometry for 30 kyr. The analysis focusses on the period 2003-2012 for which GRACE observations are available.

[Figure]

Main comments:

The paper is of good presentational quality and overall well written. I see a couple of problems with the proposed methodology and the drawn conclusions, but I believe major revisions addressing these concerns can make the manuscript an interesting contribution to The Cryosphere.

While the technical efforts that go into this work in terms of spinning up the model and performing the analysis are in themselves impressive and state of the art, I have my doubts whether the presented methodology would actually succeed in validating the ice *flow* model, as suggested by the title. The given approach is more likely to validate the output from the SMB models (which are taken as pre-existing products) rather than the ice flow model itself. Notably, a large part of the discussion is dealing with the SMB results. Validation of the ice flow results proper seem only possible where the SMB can be assumed to be sufficiently adequate for residual arguments. Even then, the given analysis (in terms of validating ice flow) is mainly limited to explaining remaining mismatch to observations with some missing processes not included in the model. The authors largely follow the argument that the SMB may be trusted, where they can explain the seasonal cycle well, However, some of the missing processes can be expected to have a seasonal cycle as well, which renders the attribution problem underdetermined. I believe the title of the paper and other passages claiming validation of "ice flow" or "ice dynamics" should be modified to reflect that limitation. It should be clearly distinguished what the contribution of ice dynamics is in the modelled trends to make clear what can be expected to be validated with the given observations.

It is regrettable and maybe symptomatic that the main plot that shows the effect of the ice flow model in the presented analysis is displaced to the appendix (Figure S3 A). It is important to realise, that the dynamic thickness change presented here is what needs to be validated (if the aim of the paper would really be validating the ice flow model!). The dominant signal including all seasonal variations are governed by the prescribed SMB forcing (Fig 2D).

The Greenland ice sheet responds on multiple time scales (seasonal to millennial) to changes in its SMB, and on the long time scales also to ice temperature and bedrock changes. This implies that changes observed today can have their origin in recent changes in SMB as well as processes set in motion hundreds to thousands of years ago. The present study by construction (steady state initialisation) only accounts for the effect of the anomalous SMB history of the last 173 years. Anything outside of this range is omitted in the model, but will still be imprinted on the observed mass changes to some extent. This is notably the case for dynamic thickening of the interior (Reeh, 1985; Huybrechts, 1994; and recently brought up again by Colgan et al 2015), which should be discussed in in the paper as a limitation of the steady state initialisation approach.

Since biases originating from the initialisation should be excluded from the analysis, what is the remaining background trend of the model after initialisation? It is important to show with an adequate control experiment that the model response is dominated by the anomalous SMB forcing and not by background model drift due to the initialisation. This should be verified with a model run forced with zero SMB anomalies over the same time span as the forward experiments (173 years).

There appears to be an inconsistency for the initialisation, because SMB_bar (1979-1988, assumed to be in equilibrium) is combined with observed velocities at 2012 (which already show some acceleration in them). While probably of minor importance for the results, this should be clearly stated. Also mention in the text (P9,l20-24) that the spin-up procedure implies that modelled mass changes over the period 2003-2012 are governed by SMB changes over that period itself, ice dynamic changes forced by SMB changes 1840-2012 and a background trend that you estimate from a control run (see point before).

Is the mass conservation approach from Morlighem at al. performed with the same SMB_bar as in the present approach. If not, I doubt that it can be called mass conserving at all. Please clarify.

A number of questions for general consistency between modelled and observed quantities. How are ice thickness changes converted to mass? What density is assumed? How do you deal with the firn layer? Did you account for the map projection error when converting between lat-lon and projected coordinates?

I disagree with the conclusion (p17, l33; p18, l27) that seasonal variations in ice flow are important features of an ice flow simulation in terms of sea-level contributions. Furthermore, I don't see any reason why an ice sheet model that does not exhibit any sub-annual variations could not be validated by GRACE data. Alternatively, you may want to discuss the risk of overfitting when including processes with a large amount of (tuned) unknown parameters to better match (seasonal) observations.

Other comments:

The Results and Discussion sections are a bit difficult to navigate, due to the lack of any subdivision. It should help to group the results and discussion into different themes or regions and introduce subsections. One could e.g. distinguish between results for mass changes in the centre from the more complex marginal regions and discuss them separately.

Confusingly, the term mascon is used throughout the manuscript in two different interpretations. While it is introduced as a short form for 'mass concentration', it is later used to refer to the regional subdivision of areas in which mass changes are measured, modelled and compared. 'mascon' seems to me like a technical slang term in the second interpretation and should be replaced by something meaningful (maybe simply 'region').

The terms BOX, MAR and RACMO are used to describe the SMB products and the ice sheet runs they are based on. Better to be distinguished.

P1, l13. Replace "is primarily controlled" by "is assumed to be primarily controlled"

P1, l18-19. What are "transient dynamics"?, rephrase.

P2, l2. Something not right with the reference here.

P2, l2. It would be good to specify the current estimate for the rate of the GrIS sea-level contribution here as a reference value.

P2, l4. More important for the future sea-level contribution from Greenland are changes in the SMB, not ice flow. Clarify in the text.

P2, l7. Replace "ice flow models" by "ice sheet models" or otherwise make clear that SMB has to be included. An ice flow model in itself is not an alternative to the extrapolation methods because it misses the most important mass change component (SMB). Please also apply for the rest of the document.

P2, l8. Please give some references for these models here or refer to past initiatives (searise and ice2sea).

P2, l9. Should say here why this alternative is most promising: because the models are physically based.

P2, l10-15. The given interpretation of the current state of ice sheet modelling is a bit simplistic and should be extended. There are recent examples of models that do capture the observed trends: Fürst et al. (2015) for Greenland and Ritz et al. (2015) for Antarctica.

P2, l21. Please be more specific what you mean by "ice flow dynamics".

P4, l5. What does "inversion" refer to here?. Clarify.

P4, l16. Incorrect reference A et al.

P5, l31. Where does the number of years 25 come from?

P5, l33-. I suppose SMB anomalies are calculated against the mean SMB (1979-1988) of the same product and then added to the mean reference SMB_bar of the BOX model. This should be mentioned.

P6, l8. "to highlight the regions where the modeled ice sheet *mass trend* differs from GRACE", or similar.

P6, l8-9. Topography is not a surface feature of the ice sheet.

P6, l9-12. This description pertaining to init and relaxation may be better placed in Section 3.2.

P6, l11. Mention here that basal melting is ignored and why.

P6, l12. Include "ideally" before "in a steady state" and "nearly" before "equal". This conditions is never strictly fulfilled in any ice sheet model I know of. Also add here that this is the assumed initial state for the year 1840.

P6, l16. Add "errors in GRACE-JPL" to the list of possible explanations for the mismatch. The background trend after initialisation (see point on control experiment) could be compounded in "limitations of our model spinup", but may need extra mention if significant.

P6, l25. Add "over time" after "RACMO" to avoid confusion.

P6, l26. Add "anomalous" before "SMB forcing".

P6, l26. Maybe "Next, we sum mass changes simulated by ISSM for the BOX ..."

P6, l27. Maybe "This mass signal represents the ISSM model estimate of ice sheet mass balance through time and is comprised of the anomalous SMB forcing at the time and the dynamic response to SMB changes since the year 1840." If a background trend from the control experiment is not negligible and not removed beforehand, it should be mentioned here as an additional contribution.

P7, l7. Replace "directly" by "is the only component that"

P8, l26. Not clear what you mean by "Regional climate model SMB products are considered to be more mature than ice dynamic models on decadal time scales". I certainly

don't see the causality between this statement and the next. Please clarify.

P9, l1. Too much information combined in this sentence makes it confusing. Revise and consider splitting in two. Also, topography is not a surface feature of the ice sheet.

P9, l5. I have not understood why velocity changes over this period are an important quantity to look at and what role they play in the interpretation. Maybe you could add a sentence to motivate that.

P9, l7. There are no outlet glaciers in the interior of the ice sheet. Please correct this sentence.

P9, l15-18. I find it confusing to discuss panel C and especially F here, in relation to panels B, D and E. The model-observed thickness (F) must be largely the results of the relaxation and (assuming small model drift) changes relatively little over the spinup. It would be much clearer to discuss a version of C and F, with modelled thickness and velocity after relaxation as the steady state of the ice sheet. Any changes afterwards can then be attributed to the historical SMB forcing and the dynamic response to that.

P9, l24. Replace "are fixed" by "are corrected"

P10, l4. What are "annuals" and "semiannuals"? Maybe "sinusoidals with an annual/semiannual cycle"?

P10, l5-6. "suggesting that the seasonal variability of SMB and its spatial distribution are well represented by the three forcing products"

P10, l23-24. "suggesting that the effect is related to melt". Could you explain? Also see comment (P13, l11-14.)

P10, l24. Should refer to (Fig. 2B) instead of (Fig. 2C) here.

P10, l25. Insert "the" before same.

P10, l28. I hope the model is conserving ice (in the sense of mass conservation).

Anyway, please reformulate.

P10, l29. "reduce the spread" is a technical interpretation. My guess is that this is not true for the relative spread. But even if there is a non-linearity in the dynamic effect, that should be the interpretation, not the pure numbers.

P11, l3. Replace "in" by "is" before "driven"

P12, l14. "be a factor" or "play a role"

P13, l10. Clarify where these numbers come from. The model could distinguish between SMB and dynamics, but the model does not agree with the GRACE data.

P13, l11-14. Since your modelling approach excludes seasonal effects from basal lubrication by melt water and ocean forcing of outlet glaciers, it is on first view somewhat surprising that your dynamic response shows any significant seasonal signal at all. Given that reduced ice discharge due to marginal thinning is the declared responsible mechanism, it seems important to mention that this is a 'passive' dynamical effect and direct consequence of the SMB forcing. In other words, the dynamics in themselves have no seasonal signature other than the one imprinted directly by the SMB change.

P13, l15-17. See my comment P10, l29. about reduced spread above. I strongly hope "that the model state ... play[s] a role in dictating the results of the three different simulations", since otherwise there would be no need to run an ice sheet model at all. However, I don't see any causal relation between the apparent change in spread and this statement.

P13, l17. As stated above, I believe the seasonal aspect of the dynamic dampening of mass loss is not really relevant. I find it generates confusion in the distinction between the two processes.

P13, l20. Not clear what "acceleration in ice dynamics is not a trivial component" means. Please reformulate.

P13, l22. While I agree generally that dynamic changes likely represent "a minor source of uncertainty" compared to uncertainty in SMB, I have quite some difficulty to see how that can be derived from the presented comparison. Please clarify.

P13, l23. Some of these marginal processes that are excluded from the modelling could certainly compensate for errors in SMB and/or included dynamics, especially since they can be assumed to have a seasonal signature by themselves. I therefore find the attribution of model error and uncertainty very much complicated, if not rendered practically impossible. This should be discussed and be reflected in the degree of certainty in the statements. E.g. replace "are responsible" by "may be assumed to be responsible" and similar.

P14, l1. "Seasonal snow cover on tundra, bare rock, ..."

P14, l3. Remove "results suggest that"

P14, l13. Please reformulate "not enough melt in relaxation SMB".

P15, l15. Replace "it is possible to quantify", by "it may be possible to quantify"

P16, l31. Maybe "both temporally and spatially"?

P16, l35. Include a discussion on interior dynamic thickening here (see comment above).

P17, l28. Remove "the" before "not well understood"

P18, l4. Move "from 2003-2012" to just after "simulations" to avoid confusion over which period SMB products are applied (namely also before 2003).

P18, l10. Insert "is" after "it"

P18, l25. I don't think you can make such firm statements about processes that are not modelled, not studied and not analysed.

Figures:

Fig 1 The colors in the legend do not match with the ones on the figure. Probably because of the gray overlay.

Fig 2 The model mesh is hardly visible at the size of panel A. An inset for one prominent region could maybe help to visualise the grid.

For my eyes panels D and E are indistinguishable. I would therefore suggest to show the difference from S3 panel A here rather than practically showing the same figure twice. Clearly, the dynamic thickness change is one of the most important variables when considering the dynamic changes and should not be hidden in the appendix. It represents the added benefit and justification for performing your analysis with an expensive ice sheet model.

Please consider using a non-linear scale for the panels B-F. It should for example become better visible that there is a positive SMB anomaly in the centre in D and E.

Why are velocities in panel C given for December 2012? Is that the reference date for the observations? If not, maybe an annual average would be more appropriate.

Fig 3 If the grey curve gives the SMB forcing for the model, shouldn't it also show seasonal variations? Please clarify and correct if necessary.

Fig 4 Maybe some of the interior mass gain could be explained by "millennial-scale ice-sheet thickening is an anticipated result of the downward advection through the ice sheet of the transition between relatively 'soft' Wisconsin ice and relatively 'hard' Holocene ice." (Colgan et al., 2015).

References

Colgan, W., Box, J. E., Andersen, M. L., Fettweis, X., Csatho, B., Fausto, R. S., Van As, D. and Wahr, J.: Greenland high-elevation mass balance: inference and implication of reference period (1961–90) imbalance, Annals of Glaciology, 56(70), 105–117, doi:10.3189/2015AoG70A967, 2015.

Fürst, J. J., Goelzer, H. and Huybrechts, P.: Ice-dynamic projections of the Greenland ice sheet in response to atmospheric and oceanic warming, The Cryosphere, 9(3), 1039–1062, doi:10.5194/tc-9-1039-2015, 2015.

Ritz, C., Edwards, T. L., Durand, G., Payne, A. J., Peyaud, V. and Hindmarsh, R. C. A.: Potential sea-level rise from Antarctic ice-sheet instability constrained by observations, Nature, doi:10.1038/nature16147, 2015.

Shannon, S. R. and Payne, A. J.: Enhanced basal lubrication and the contribution of the Greenland ice sheet to future sea-level rise. PNAS, 2013.

---

## Referee Comment (RC2) · Anonymous Referee #2 · 6 May 2016

I am not sure what the paper intends to achieve. As stated, the idea of the paper is to evaluate (validate?) the behavior of ISSM by comparing results from an initialisation experiment to observed mass changes as derived from GRACE. However, I understand the purpose of the paper more as trying to explain the current evolution of the Greenland ice sheet and as an attempt to decompose the observed ice-sheet imbalance into its possible contributions, and attribute the residual from subtracting the modeled trend from the observed trend to (mainly) missing ice physics at the margin. If so, I believe there are serious problems with the way the experiments have been set up.

First, the ice-sheet model is run to steady state with the 1979-1988 average SMB from the Box SMB model, and this state is then taken as initial condition for 1846. Implicit in this approach is that the ice sheet was also in steady state in 1846 and that the ice flow and ice thickness field for both periods was in equilibrium with the 1979-1988

SMB for both periods (and identical). Even though it is rather well established that Greenland ice sheet volume was overall not changing during the 1979-1988 period, this does not mean that the local mass balance was necessarily zero all over the ice sheet (which is in fact highly unlikely), i.e. thinning in certain regions could well have been compensated by thickening in other regions. The current evolution of the Greenland ice sheet is the result of the superimposition of many different signals on a multitude of time scales. Very long time scales of 10**3-10**4 years are connected to viscosity changes in the basal layers from ice temperature and ice property changes and to ice-dynamic adjustments to geometry changes that likely extend back as far as the Last Glacial Maximum. The approach taken by the authors ignores all these longer-term effects as a contribution to current mass changes of the Greenland ice sheet.

Second, I am somewhat surprised by the choice of the ice sheet model. For their study, the authors opted to use the 2D SSA version of ISSM, which ignores vertical shearing. That is fine for modeling ice streams in Antarctica with high basal sliding, and may apply in Greenland in outlet glaciers close to the coast, but much of the Greenland ice sheet is frozen to bedrock with a flow regime that is better approximated by the SIA. It is furthermore not clear whether ice temperature is evolving together with the ice flow (I guess not, it seems to be prescribed) which a priori excludes a temperature change contribution to the current ice evolution (as are e.g. changes in ice hardness related to the downward advection of the LGM/Holocene boundary, amongst possible other processes).

Third, the paper does not convince me that the difference between modeled and observed trends can be attributed well on a regional scale as the uncertainties in input and observed fields are too large (especially SMB, but also bedrock elevation, in addition to errors in the GRACE field that is moreover spatially poorly resolved) and the simplifications in the model setup and initialization procedure are too important to reach solid conclusions.

Apart from these reservations, partly confirmed by the authors when differentiating

between different regions, I found the results section hard to swallow as it is much too long, not well organized, and lacks synthesis.

I don't think the problems with the paper as it stands now (focus and length of the paper, problems with the model setup and the initialization, not considering pre-1846 contributions to ice evolution, messy conclusions, ...) can be fixed with only a major revision.

A few other comments

Abstract, p.1, lines 17, 19: 'transient' processes, 'transient' dynamics: what is meant with 'transient' in this context?

p. 2, line 2: 'Sto': the referencing is somewhat sloppy. Presumably, 'Sto' is Stocker et al., 2014. This kind of referencing occurs in many other places in the manuscript. More generally, when referring to the IPCC work, it is recommended to refer to the individual chapters.

p. 4, line 16: who are 'A. et al.'?

p. 4, line 6: what is meant with 'surface mass variations'? Do the authors perhaps mean surface elevation changes? If so, are these expressed in ice equivalents, i.e. in mass changes? Or do the authors mean 'ice mass' variations as opposed to the GIA contribution to GRACE?

p. 5: why is only a 9-year period chosen for averaging SMB. Isn't that a bit short considering the inherent variability of climate conditions over the ice sheet?

p. 6, line 9: 'ISSM Greenland observed velocities': do you perhaps mean 'modelled velocities'?

...

p. 29, Fig. 2: the figures are too small to distinguish the patterns. The blue-red colour scale does not allow to differentiate much.

p. 31, Figure 5: the colour legend seems to be for the difference plot (panel C) only, but not for panels A and B. 'Mascons' with the same colour do not always appear green in panel C. Otherwise there is a problem with the color scale.

---

## Author Comment (AC1) · 4 Jun 2016

We would like to thank the referees for giving their time to raise important discussion points and to assess the quality of our manuscript. We believe many of the suggestions were helpful in improving the presentation of our results and the manuscript overall. Below, we address the review comments and suggested modifications. For those comments that required updates to text, we enclose a paper draft with corresponding changes in red. The main modifications can be summarized as follows:

1. The tone of the manuscript has been changed to focus on the comparison of ice sheet model estimates of Greenland mass balance to GRACE, rather than focusing on the ice flow portion of the ice sheet model.

2. All figures/analysis have been updated with new GRACE data (version 2 of the

[Figure]

JPL mascon product), as well as new ISSM runs that implement the L1L2 formulation which includes effects of longitudinal stresses and considers the contribution of vertical gradients to vertical shear. We note that updates to both GRACE and ISSM did not change the conclusions presented.

3. Much of the technical discussion and several figures has been moved to the appendix in an effort to shorten the manuscript. Additionally, we have added many subheadings to increase the readability of the manuscript.

4. We have included discussion on dynamic thickening in the interior of the ice sheet.

**1 Anonymous Referee 1**

Introduction:

Schlegel and co-authors present a methodological paper that deals with the important question how an ice flow model (here of the Greenland ice sheet) could be validated with observational data. Measured changes in the gravitational field as recovered from the GRACE satellite mission are utilised to estimate mass trends on a relatively small regional scale (300 km) and are compared with mass changes derived from a combination of three SMB models with an ice flow model (ISSM). The ice flow model is initialised to a steady state with the average SMB from the Box model for the period 1979-1988, by first inverting for observed velocities and than relaxing the geometry for 30 kyr. The analysis focusses on the period 2003-2012 for which GRACE observations are available.

Main comments:

The paper is of good presentational quality and overall well written. I see a couple of problems with the proposed methodology and the drawn conclusions, but I believe major revisions addressing these concerns can make the manuscript an interesting contribution to The Cryosphere.

While the technical efforts that go into this work in terms of spinning up the model and performing the analysis are in themselves impressive and state of the art, I have my doubts whether the presented methodology would actually succeed in validating the ice *flow* model, as suggested by the title. The given approach is more likely to validate the output from the SMB models (which are taken as pre-existing products) rather than the ice flow model itself. Notably, a large part of the discussion is dealing with the SMB results. Validation of the ice flow results proper seem only possible where the SMB can be assumed to be sufficiently adequate for residual arguments. Even then, the given analysis (in terms of validating ice flow) is mainly limited to explaining remaining mismatch to observations with some missing processes not included in the model. The authors largely follow the argument that the SMB may be trusted, where they can explain the seasonal cycle well, However, some of the missing processes can be expected to have a seasonal cycle as well, which renders the attribution problem underdetermined. I believe the title of the paper and other passages claiming validation of "ice flow" or "ice dynamics" should be modified to reflect that limitation. It should be clearly distinguished what the contribution of ice dynamics is in the modelled trends to make clear what can be expected to be validated with the given observations.

These are valid points, and stem from our attempt to cover many topics in the same manuscript. We have attempted to narrow our focus in the manuscript, to compare GRACE-JPL against state-of-the-art model estimates (which include SMB and ISSM). As suggested, references to "ice flow" have been changed to "ice sheet" throughout the manuscript, and we have changed all references to "ice dynamics" to reflect more

precise mechanisms (i.e., ice discharge or paleo-driven background dynamics).

The goal of this study was never to use GRACE to validate our model, as many processes are admittedly missing from the model itself and are clearly at play in determining Greenland MB. We chose to use the terminology "evaluate" in order to reflect this, but the tone of the manuscript did not always complement these intentions. To remedy this, we have changed the title of the paper and have expanded the introduction to include a discussion of SMB models, which are the most important element of the modeled mass balance.

We hope that the title, tone, and focus of the manuscript now better reflects these goals.

It is regrettable and maybe symptomatic that the main plot that shows the effect of the ice flow model in the presented analysis is displaced to the appendix (Figure S3 A). It is important to realise, that the dynamic thickness change presented here is what needs to be validated (if the aim of the paper would really be validating the ice flow model!). The dominant signal including all seasonal variations are governed by the prescribed SMB forcing (Fig 2D).

The dynamic thickness change, on a mascon-by-mascon basis, is shown in Fig. 7A. The high resolution spatial patterns of the ice sheet model response is included in the supplement for reference, in order to better explain the mascon-scale spatial patterns presented in Fig. 7A. Because our goal here is not to validate the ice flow model (as originally suggested by our title), we have decided to focus on assessment of the model-based estimate of MB, and have decided to keep this figure in the supplement, for reference.

The Greenland ice sheet responds on multiple time scales (seasonal to millennial) to changes in its SMB, and on the long time scales also to ice temperature and bedrock

changes. This implies that changes observed today can have their origin in recent changes in SMB as well as processes set in motion hundreds to thousands of years ago. The present study by construction (steady state initialisation) only accounts for the effect of the anomalous SMB history of the last 173 years. Anything outside of this range is omitted in the model, but will still be imprinted on the observed mass changes to some extent. This is notably the case for dynamic thickening of the interior (Reeh, 1985; Huybrechts, 1994; and recently brought up again by Colgan et al 2015), which should be discussed in in the paper as a limitation of the steady state initialisation approach.

This is a valid point, and you bring up key limitations of our spinup process that we agree should be highlighted more in the text of the manuscript. In order to do so, we have added a paragraph in the Initialization and Relaxation section of the manuscript to discuss caveats. In addition, as suggested, we have extended our discussion of the interior thickening to include references to past studies that describe the southern interior thickening driven by the downward propagation of more rigid Holocene ice (*Reeh*, 1985) as well as the observed thickening in the Northeast that has recently been attributed to recent decrease in accumulation in comparison to the average Holocene accumulation rates (*MacGregor et al.*, 2016). We have taken advantage of our steady-state spinup to estimate dynamic mass gain in the interior of the ice sheet (about 9 Gt/yr), which is focused in the Northeast. Unfortunately, the resolution and placement of the mascons in the South do not allow us to quantify "Reeh" thickening that may be occurring there (though we note that minor thickening signals in mascon 166 may be related to this phenomenon).

Since biases originating from the initialisation should be excluded from the analysis, what is the remaining background trend of the model after initialisation? It is important to show with an adequate control experiment that the model response is dominated by the anomalous SMB forcing and not by background model drift due to the initialisation.

This should be verified with a model run forced with zero SMB anomalies over the same time span as the forward experiments (173 years).

We have added a dotted line to Fig. 3 that represents a control experiment, forced with $\overline{SMB}$ during the 173 years. This line illustrates a change in ice sheet mass over that period, which is an order of magnitude less than the intra-annual variability over the historical period.

There appears to be an inconsistency for the initialisation, because SMB_bar (1979-1988, assumed to be in equilibrium) is combined with observed velocities at 2012 (which already show some acceleration in them). While probably of minor importance for the results, this should be clearly stated. Also mention in the text (P9,l20-24) that the spin-up procedure implies that modelled mass changes over the period 2003-2012 are governed by SMB changes over that period itself, ice dynamic changes forced by SMB changes 1840-2012 and a background trend that you estimate from a control run (see point before).

Such inconsistencies are, unfortunately, the downside to assimilating the best observations currently available into ISSM. They are, in fact, the very reason why the model drifts upon spinup - and why we have decided to relax the model to a virtual steady-state before forcing it with historical SMB. After relaxation, the resulting velocities, though similar to present day velocities, have changed to be in a relative steady-state with $\overline{SMB}$. Since we have little to no information about how the basal conditions of the ice sheet have changed over the last 173 years, we believe our best assumption is to hold the assimilated ice sheet properties (i.e. ice viscosity and basal drag) constant. At the end of the Initialization and Relaxation section, we have added a final paragraph that acknowledges the key limitations of our approach. We hope this sufficiently covers the referee's concerns expressed here.

Is the mass conservation approach from Morlighem at al. performed with the same SMB_bar as in the present approach. If not, I doubt that it can be called mass conserving at all. Please clarify.

The reference to mass-conserving has been replaced with "BedMachine bedrock", which is a more accurate description of the product. We are not using the same $\overline{SMB}$ as Dr. Morlighem, which, in part (along with a number of other data mismatches noted earlier), is why we have decided to relax to a virtual steady-state before running our historical simulation.

A number of questions for general consistency between modelled and observed quantities. How are ice thickness changes converted to mass? What density is assumed?

We have added some additional sentences in the GRACE period mass estimates section of the manuscript that now indicate that we assume the density of ice to be 917 kg/m$^3$, and describe our process for translating ISSM ice thickness changes to mass within mascons: "Within the ice sheet boundary, mass changes are considered on individual elements of the ISSM mesh and outside of the ice sheet boundary, mass changes are considered on individual elements of a 10 km triangular mesh. To assess mass change within each mascon, elements within the projected mascon boundaries are summed, and elements bisected by mascon boundaries contribute to this sum proportionally (by area) to the mascons that fall within their individual outlines. This procedure is mass conserving on the continental-scale, but please note that it introduces small leakage errors along the mascon boundaries that are insignificant compared to the uncertainties considered in this study."

How do you deal with the firn layer? Did you account for the map projection error when converting between lat-lon and projected coordinates?

At the end of the Models of Historical SMB section, we now explain that any processes related to firn densification are modeled by each individual RCM surface model, and beyond that calculation, ISSM assumes that the SMB provided by the RCM is ice. Any map projection errors are assessed in translation from ISSM thickness changes to mass change in each mascon. This explanation (as stated above) has been placed in the GRACE period mass estimates section and describes our effort to conserve total ice sheet mass change and to minimize leakage between mascons during this calculation.

I disagree with the conclusion (p17, l33; p18, l27) that seasonal variations in ice flow are important features of an ice flow simulation in terms of sea-level contributions. Furthermore, I don't see any reason why an ice sheet model that does not exhibit any sub-annual variations could not be validated by GRACE data. Alternatively, you may want to discuss the risk of overfitting when including processes with a large amount of (tuned) unknown parameters to better match (seasonal) observations.

We have reworded the end of the abstract, discussion, and conclusion sections to say that continued improvements in physically-based modeling of the processes most likely to be responsible for intra-annual variability will likely improve the skill of ice sheet models, particularly in terms of decadal-scale modeling. In addition, we stress that such models which consider hydrological processes will have the opportunity to take full advantage of observations that are available on a monthly-to-seasonal timescales.

Other comments:
The Results and Discussion sections are a bit difficult to navigate, due to the lack of any subdivision. It should help to group the results and discussion into different themes or regions and introduce subsections. One could e.g. distinguish between results for

mass changes in the centre from the more complex marginal regions and discuss them separately.

We have organized the Results section into categories, which follows the logic of our plots from a continental-scale view of the comparison down to a higher spatial resolution (regional) and higher temporal resolution (seasonal). We believe the new subsections will make it easier for the reader to navigate. We have also reorganized the Discussion section, which is now categorized by region of the ice sheet.

Confusingly, the term mascon is used throughout the manuscript in two different interpretations. While it is introduced as a short form for 'mass concentration', it is later used to refer to the regional subdivision of areas in which mass changes are measured, modelled and compared. 'mascon' seems to me like a technical slang term in the second interpretation and should be replaced by something meaningful (maybe simply 'region').

We understand the reviewer's concern; however, we feel that the terminology presented here is self-consistent. As mentioned, 'mascon' is short for 'mass concentration'. The unique aspect of using mascons as basis functions when processing GRACE data is that they do explicitly define regions within a known latitude/longitude domain (unlike other basis functions such as spherical harmonic coefficients which are global by nature): so, each mascon explicitly defines a region. The word "region" and "mascon" are somewhat synonymous. We hesitate to use a word such as "region" when discussing the results, simply because the placement of the mascons is rather arbitrary and their boundaries do not necessarily delineate specific geographic regions of interest (i.e. drainage basins). Therefore, we prefer to keep the terminology as is.

The terms BOX, MAR and RACMO are used to describe the SMB products and the ice sheet runs they are based on. Better to be distinguished.

We have made a distinction in the text between BOX, MAR, RACMO and the resulting ISSM runs, which we now refer to as ISSM-GrIS BOX, ISSM-GrIS MAR, and ISSM-GrIS RACMO.

P1, l13. Replace "is primarily controlled" by "is assumed to be primarily controlled"

This line has been updated as suggested, in the abstract.

P1, l18-19. What are "transient dynamics"?, rephrase.

Transient dynamics has been removed from the abstract and the last three sentences of the abstract have been reworked as a result.

P2, l2. Something not right with the reference here.

Thank you for pointing this out. Something did not render correctly by TCD, and we will make sure the reference is correct when the corrected manuscript version is uploaded.

P2, l2. It would be good to specify the current estimate for the rate of the GrIS sea-level contribution here as a reference value.

This has been added to the text. Thank you for the suggestion.

P2, l4. More important for the future sea-level contribution from Greenland are changes in the SMB, not ice flow. Clarify in the text.

We have added a sentence here to specify that the largest source of uncertainty in future sea level rise is SMB, and secondly ice sheet discharge into the ocean.

P2, l7. Replace "ice flow models" by "ice sheet models" or otherwise make clear that SMB has to be included. An ice flow model in itself is not an alternative to the extrapolation methods because it misses the most important mass change component (SMB). Please also apply for the rest of the document.

Throughout the text, we have replaced the reference to "ice flow models" with "ice sheet models", where we refer to total mass balance of the ice sheet (or SMB+discharge), as suggested by the referee.

P2, l8. Please give some references for these models here or refer to past initiatives (searise and ice2sea).

Here, we have added reference to both searise and ice2sea as sources that showcase a collection of ice sheet models and various sea level-based experiments.

P2, l9. Should say here why this alternative is most promising: because the models are physically based.

We have added a reference to the fact that these models are physically based.

P2, l10-15. The given interpretation of the current state of ice sheet modelling is a bit simplistic and should be extended. There are recent examples of models that do capture the observed trends: Fürst et al. (2015) for Greenland and Ritz et al. (2015) for Antarctica.

The introduction of the manuscript has been reworked, and now includes a discussion of ice sheet models (including an acknowledgement to efforts that have been successful in matching current Greenland mass balance trends), the most current SMB

products derived from Regional Climate Models, and an explanation of GRACE. We believe this summary, which now focuses on all the products we are assessing in this study, is more comprehensive.

P2, l21. Please be more specific what you mean by "ice flow dynamics".

"Ice flow dynamics" has been updated to read, "ice discharge into the ocean". We believe this statement is more accurate.

P4, l5. What does "inversion" refer to here?. Clarify.

We have removed the text referring to the "inversion" as it was unnecessary to make our point.

P4, l16. Incorrect reference A et al.

Actually, this reference is correct. Mr. A had the very unfortunate life circumstance of having a last name that was uniquely given by the first letter of the alphabet! We share in your sentiments that this is quite unfortunate for Mr. A.

P5, l31. Where does the number of years 25 come from?

This sentence has now been updated to read "from 1970 to 2000".

P5, l33. I suppose SMB anomalies are calculated against the mean SMB (1979-1988) of the same product and then added to the mean reference SMB_bar of the BOX model. This should be mentioned.

Yes, this is correct. We have updated the paragraph to read "The total SMB forcing for each RCM product is equal to $\overline{SMB}$ plus the monthly SMB anomalies derived for that particular product beginning in 1979."

P6, l8. "to highlight the regions where the modeled ice sheet \*mass trend\* differs from GRACE", or similar.

We have updated this sentence to state that our goal is to compare ISSM simulations that are "forced with three different high-resolution RCM-derived SMB products, against the monthly GRACE-JPL product, in order to highlight the regions where modeled ice sheet mass trend and annual amplitude differ from GRACE".

P6, l8-9. Topography is not a surface feature of the ice sheet.

We have changed "surface features" in this statement to read "surface properties".

P6, l9-12. This description pertaining to init and relaxation may be better placed in Section 3.2.

Thank you for this suggestion. A portion of this description has been moved to Section 3.2, and the second part, describing forward model SMB forcing has been moved to the beginning of Section 3.3.

P6, l11. Mention here that basal melting is ignored and why.

The formula for MB (now in Sect. 3.2), has been updated to include basal mass balance, and we explain that this term is ignored because ISSM does not simulate basal hydrology.

P6, l12. Include "ideally" before "in a steady state" and "nearly" before "equal". This conditions is never strictly fulfilled in any ice sheet model I know of. Also add here that this is the assumed initial state for the year 1840.

We have updated this statement to reflect that our simulation is in a "virtual" steady-state and have added "nearly" before "equal" to reflect that mass balance is near zero after relaxation. In addition, in Section 3.2, we have added a sentence to clarify that the relaxed ice sheet is taken as an assumed state of the ice sheet in 1840.

P6, l16. Add "errors in GRACE-JPL" to the list of possible explanations for the mis-match. The background trend after initialisation (see point on control experiment) could be compounded in "limitations of our model spinup", but may need extra mention if significant.

We have added GRACE-JPL to the list and have mentioned that we have attempted to quantify errors in SMB and GRACE where possible. We do not feel it is necessary to expand upon the limitations in model spinup, as we have added a paragraph in Section 3.2 listing the limitation and assumptions with the steady-state spinup procedure.

P6, l25. Add "over time" after "RACMO" to avoid confusion.

We have added "over time" to the end of this sentence.

P6, l26. Add "anomalous" before "SMB forcing".

We have added "anomalous" here, as suggested.

P6, l26. Maybe "Next, we sum mass changes simulated by ISSM for the BOX ..."

Thank you for this suggestion. The change has been incorporated.

P6, l27. Maybe "This mass signal represents the ISSM model estimate of ice sheet mass balance through time and is comprised of the anomalous SMB forcing at the time and the dynamic response to SMB changes since the year 1840." If a background trend from the control experiment is not negligible and not removed beforehand, it should be mentioned here as an additional contribution.

The sentence in question has been updated to read, "This mass signal represents the ISSM model estimate of ice sheet mass balance through time and is comprised of the anomalous SMB forcing and the dynamic response to SMB changes since the year 1840."

P7, l7. Replace "directly" by "is the only component that"

The manuscript has been updated as suggested.

P8, l26. Not clear what you mean by "Regional climate model SMB products are considered to be more mature than ice dynamic models on decadal time scales". I certainly don't see the causality between this statement and the next. Please clarify.

We agree. This sentence has been removed for clarity.

P9, l1. Too much information combined in this sentence makes it confusing. Revise and consider splitting in two. Also, topography is not a surface feature of the ice sheet.

We have split this sentence into three and have reworked these sentences to better motivate why we are interested in evaluating the model spin-up procedure.

P9, l5. I have not understood why velocity changes over this period are an important quantity to look at and what role they play in the interpretation. Maybe you could add a sentence to motivate that.

It true that velocity changes have only been included as a reference for showing how ISSM models the change in ice flow over the 10 year period of our study. Therefore, we have moved this figure to the supplement and have removed extended discussion pertaining to this figure from the main text of the manuscript

P9, l7. There are no outlet glaciers in the interior of the ice sheet. Please correct this sentence.

Thank you for pointing this out. We have changed "interior" to now read "along the margins".

P9, l15-18. I find it confusing to discuss panel C and especially F here, in relation to panels B, D and E. The model-observed thickness (F) must be largely the results of the relaxation and (assuming small model drift) changes relatively little over the spinup. It would be much clearer to discuss a version of C and F, with modelled thickness and velocity after relaxation as the steady state of the ice sheet. Any changes afterwards can then be attributed to the historical SMB forcing and the dynamic response to that.

As suggested, we have updated the difference plots for ISSM velocity and thickness against observations, so that they now show the difference between the relaxed ISSM "steady-state" and observational datasets. We agree with the referee that this is a

much clearly comparison, especially since most of the differences are due to the model relaxation. These figures are now located in the supplement.

P9, l24. Replace "are fixed" by "are corrected"

We have updated this sentence to read "are offset".

P10, l4. What are "annuals" and "semiannuals"? Maybe "sinusoidals with an annual/ semiannual cycle"?

We have reworded this sentence to say we estimate "sinusoids with frequencies of once and twice per year".

P10, l5-6. "suggesting that the seasonal variability of SMB and its spatial distribution are well represented by the three forcing products"

Thank you for this suggestion. The manuscript has been updated as suggested here.

P10, l23-24. "suggesting that the effect is related to melt". Could you explain? Also see comment (P13, l11-14.)

We have added more text in this paragraph, to explain why muted annual amplitudes are likely related to runoff-induced thinning during the summer months. Since we are focusing on assessing annual amplitudes and seasonal cycles, we believe it is important to point out how the ice sheet model affects results on monthly to seasonal timescales. Therefore, we have left this description in the results section. However, as suggested for P13, l11-14, we have removed the extended discussion of this point, as we agree that it is confusing to the reader.

P10, l24. Should refer to (Fig. 2B) instead of (Fig. 2C) here.

The reference has been updated to refer to 2B instead of 2C.

P10, l25. Insert "the" before same.

Thank you; this has been updated in the manuscript.

P10, l28. I hope the model is conserving ice (in the sense of mass conservation). Anyway, please reformulate.

This sentence has been reworked, and the manuscript has been updated to read, "Overall, this behavior increases the modeled MB".

P10, l29. "reduce the spread" is a technical interpretation. My guess is that this is not true for the relative spread. But even if there is a non-linearity in the dynamic effect, that should be the interpretation, not the pure numbers.

These lines have been updated to state that the behavior decreases MB "similarly for all three simulations, ultimately resulting in a better agreement between ISSM-GrIS BOX, ISSM-GrIS MAR, and ISSM-GrIS RACMO".

P11, l3. Replace "in" by "is" before "driven"

This typo has been corrected.

P12, l14. "be a factor" or "play a role"

"Factor" has been changed to "role" as suggested.

P13, l10. Clarify where these numbers come from. The model could distinguish between SMB and dynamics, but the model does not agree with the GRACE data.

We have modified this text and have moved it to Section 6.1. There, we state the percentage of mass loss that the model captures relative to GRACE. In addition, in order to address this comment, we have made an effort to reference a figure or the table when placing a number in the text.

P13, l11-14. Since your modelling approach excludes seasonal effects from basal lubrication by melt water and ocean forcing of outlet glaciers, it is on first view somewhat surprising that your dynamic response shows any significant seasonal signal at all. Given that reduced ice discharge due to marginal thinning is the declared responsible mechanism, it seems important to mention that this is a 'passive' dynamical effect and direct consequence of the SMB forcing. In other words, the dynamics in themselves have no seasonal signature other than the one imprinted directly by the SMB change.

This discussion has been moved to Section 6.3 and has been updated. The statements, now in the Results section, have been updated to point out that the decrease in discharge is a direct consequence of the SMB-driven thinning of the margins, and that the model simulates this change in discharge because ice thickness changes through time.

P13, l15-17. See my comment P10, l29. about reduced spread above. I strongly hope "that the model state ... play[s] a role in dictating the results of the three different simulations", since otherwise there would be no need to run an ice sheet model at all.

However, I don't see any causal relation between the apparent change in spread and this statement.

This is a good point, and we have updated the manuscript accordingly. The statement about the initial model state has been removed from these lines, and the statement about model spread (now in the Results, Section 6.3.) has been updated to read that the marginal thinning serves to reduce "the MB of all the simulations in a similar way, resulting in better agreement between ISSM-GrIS BOX, ISSM-GrIS MAR, and ISSM-GrIS RACMO than between the SMB products themselves."

P13, l17. As stated above, I believe the seasonal aspect of the dynamic dampening of mass loss is not really relevant. I find it generates confusion in the distinction between the two processes.

We have removed this sentence and consolidated this discussion to the results, Section 6.3. In the discussion Model Assessment section, we also make the point that the marginal thinning results in an overall decrease in mass trend for the mascons along the margins.

P13, l20. Not clear what "acceleration in ice dynamics is not a trivial component" means. Please reformulate.

This sentence has been removed from the discussion, and we now indicate that the changes to ice discharge modeled by ISSM are "minor in comparison to the direct contribution from the SMB forcing itself," in the Model Assessment section.

P13, l22. While I agree generally that dynamic changes likely represent "a minor source of uncertainty" compared to uncertainty in SMB, I have quite some difficulty to see how that can be derived from the presented comparison. Please clarify.

Along with the previous sentence, this sentence has been reworded to clarify that over the short time period analyzed for this study, that the ice responses captured by the model are minor compared to the changes in the SMB forcing itself.

P13, l23. Some of these marginal processes that are excluded from the modelling could certainly compensate for errors in SMB and/or included dynamics, especially since they can be assumed to have a seasonal signature by themselves. I therefore find the attribution of model error and uncertainty very much complicated, if not rendered practically impossible. This should be discussed and be reflected in the degree of certainty in the statements. E.g. replace "are responsible" by "may be assumed to be responsible" and similar.

We have updated the verbiage in this passage and throughout the paper, to not include discussion of uncertainty, as it confuses the issue. Our goal here is to assess the importance of including an ice sheet model in the calculation of "ice dynamics" typical for GRACE-based studies (i.e. subtract SMB from GRACE to get a value for ice dynamics). We have tried to make this clearer in the paragraph noted here, by eliminating the uncertainty terminology and adding the wording suggested by the referee.

P14, l1. "Seasonal snow cover on tundra, bare rock, ..."

This part of the discussion has been removed from the manuscript.

P14, l3. Remove "results suggest that"

These words have been removed from the manuscript.

P14, l13. Please reformulate "not enough melt in relaxation SMB".

This has been changed to "general underestimation of surface melt runoff".

P15, l15. Replace "it is possible to quantify", by "it may be possible to quantify"

The suggested change has been made on P15, l20.

P16, l31. Maybe "both temporally and spatially"?

"Temporal" has been changed to "temporally" as suggested.

P16, l35. Include a discussion on interior dynamic thickening here (see comment above).

As suggested, we have included a discussion of dynamic thickening of the interior. Such a discussion strengthens this manuscript, and we are thankful for the suggestion.

P17, l28. Remove "the" before "not well understood"

This typo has been updated in the manuscript.

P18, l4. Move "from 2003-2012" to just after "simulations" to avoid confusion over which period SMB products are applied (namely also before 2003).

As suggested, we have moved "from 2003-2012" to the earlier part of the sentence, after "simulations".

P18, l10. Insert "is" after "it"

The manuscript has been updated as suggested.

P18, l25. I don't think you can make such firm statements about processes that are not modelled, not studied and not analysed.

We have reworked the end of this concluding paragraph to state that (in reference to the high-frequency variations that appear in the GRACE-JPL signal), that hydrological and ocean-driven processes are strong candidates for those processes that could account for such a signal.

Figures:
Fig 1 The colors in the legend do not match with the ones on the figure. Probably because of the gray overlay.

Yes, this is a problem with the overlay. The figure has been updated to take the overlay into consideration and the colors should match now. Thank you for bringing this up to us.

Fig 2 The model mesh is hardly visible at the size of panel A. An inset for one prominent region could maybe help to visualise the grid. For my eyes panels D and E are indistinguishable. I would therefore suggest to show the difference from S3 panel A here rather than practically showing the same figure twice. Clearly, the dynamic thickness change is one of the most important variables when considering the dynamic changes and should not be hidden in the appendix. It represents the added benefit and justification for performing your analysis with an expensive ice sheet model. Please consider using a non-linear scale for the panels B-F. It should for example become better visible that there is a positive SMB anomaly in the centre in D and E. Why are velocities in

panel C given for December 2012? Is that the reference date for the observations? If not, maybe an annual average would be more appropriate.

All the panels for Fig. 2 have been moved to the supplement, because, as discussed above, these figures are for reference and do not pertain directly to the mascon-by-mascon comparison we are addressing here. In the supplement, the figures are now much larger, 2 panels per page, so the colors and spatial patterns are more visible for the reader. As discussed above, a mascon version of the dynamic thickness change plot is already in the manuscript (Fig. 6). The dynamic plot in the supplement is now enlarged, so that it shows more detail. For the velocities and thickness comparisons against observations, we now use the relaxed model as the reference. For the velocity change from 2003-2012, we now use annual averages of velocities for these years.

Fig 3 If the grey curve gives the SMB forcing for the model, shouldn't it also show seasonal variations? Please clarify and correct if necessary.

We have included a light gray line in this plot, which represents the seasonal SMB forcing. That you for pointing this out.

Fig 4 Maybe some of the interior mass gain could be explained by "millennial-scale ice-sheet thickening is an anticipated result of the downward advection through the ice sheet of the transition between relatively 'soft' Wisconsin ice and relatively 'hard' Holocene ice." (Colgan et al., 2015).

Again, thank you for this suggestion. Unfortunately, the largest "Reeh" thickening occurs in the Southern ice sheet, and most of our observed thickening is taking place in the Northeast. (The spatial resolution in the south is not refined enough to separate

the interior from the margins, so any background trend is indistinguishable in magnitude over the mascon.) We have noted this process as a possible explanation for the thickening, however, and have also discussed other theories for background thickening in the Northeast.

**2   Anonymous Referee 2**

I am not sure what the paper intends to achieve. As stated, the idea of the paper is to evaluate (validate?) the behavior of ISSM by comparing results from an initialisation experiment to observed mass changes as derived from GRACE. However, I understand the purpose of the paper more as trying to explain the current evolution of the Greenland ice sheet and as an attempt to decompose the observed ice-sheet imbalance into its possible contributions, and attribute the residual from subtracting the modeled trend from the observed trend to (mainly) missing ice physics at the margin. If so, I believe there are serious problems with the way the experiments have been set up.

As suggested by referee 1, we have reworked the manuscript, including the title, tone, and focus to address the confusion about validation of an ice sheet model that is missing physical processes that play a significant role in dictating Greenland MB. We agree that, instead, our goal is to use GRACE to assess the dynamics that are not captured by current model-based estimates of Greenland MB (comprised of SMB and ice sheet model), and we hope that the new version of the manuscript makes this goal clearer.

First, the ice-sheet model is run to steady state with the 1979-1988 average SMB from the Box SMB model, and this state is then taken as the initial condition for 1846. Implicit in this approach is that the ice sheet was also in steady state in 1846 and that the ice flow and ice thickness field for both periods was in equilibrium with the 1979-1988 SMB for both periods (and identical). Even though it is rather well established that Greenland ice sheet volume was overall not changing during the 1979-1988 period, this does not mean that the local mass balance was necessarily zero all over the ice sheet (which is in fact highly unlikely), i.e. thinning in certain regions could well have been compensated by thickening in other regions.

This is a very good point, and part of the assessment of this study is to determine where our assumption might be invalid, through comparison with observations. Overall, we find that in order to study mass balance variations over such a short timescale, the results are not strongly affected by our spinup assumptions. For instance, in the supplement, we include a number of variations in model spinup, using different MAR products for the reference climatology (for example MAR2, MAR 3.5.2, and MAR 3.5.2 with NCEP boundary conditions), and each product has different spatial patterns in the spinup climatology. We find (as mentioned in the manuscript) that our results are not strongly sensitive to the initial spatial patterns of our spinup climatology. Additional experiments have also shown that our results for the short period 2003-2012, would not change if a climatology between 1840 and 1900 were used for the spinup. In fact, to the first order, the ice sheet model responds almost exclusively to the anomalies in SMB, and these responses are small compared to the SMB internal variability. We expect any second-order responses due to our choice of steady-state climatology to be even smaller, and would not change the conclusions presented here.

The current evolution of the Greenland ice sheet is the result of the superimposition of many different signals on a multitude of time scales. Very long time scales of $10^{**}3$-$10^{**}4$ years are connected to viscosity changes in the basal layers from ice temperature and ice property changes and to icedynamic adjustments to geometry changes that likely extend back as far as the Last Glacial Maximum. The approach taken by the authors ignores all these longer-term effects as a contribution to current mass changes of the Greenland ice sheet.

The spinup procedure does ignore these background effects, and this is a limitation that we discuss in the initialization section of the manuscript. However, because we have chosen a steady-state: spinup procedure, we are confident that our ice sheet model is responding to only changes in SMB. In terms of dynamic background trends in the ice sheet, this is an advantage, as pointed out by referee 1. As suggested, by removing

the model results from interior, we are able to isolate signals of interior thickening. We believe the addition of this component to our results and discussion strengthens the manuscript.

Second, I am somewhat surprised by the choice of the ice sheet model. For their study, the authors opted to use the 2D SSA version of ISSM, which ignores vertical shearing. That is fine for modeling ice streams in Antarctica with high basal sliding, and may apply in Greenland in outlet glaciers close to the coast, but much of the Greenland ice sheet is frozen to bedrock with a flow regime that is better approximated by the SIA. It is furthermore not clear whether ice temperature is evolving together with the ice flow (I guess not, it seems to be prescribed) which a priori excludes a temperature change contribution to the current ice evolution (as are e.g. changes in ice hardness related to the downward advection of the LGM/Holocene boundary, amongst possible other processes).

For the newly revised version of the manuscript, we have updated and reprocessed all of our results to use the L1L2 version of ISSM (which considers vertical shear to the same extent that SIA does, but also considers longitudinal stresses), in order to address your concerns. This update does not change any of our main discussion or conclusions.

Over the short time period we investigate here, changes to ice temperature are not expected to cause any significant changes to our results (*Seroussi et al.*, 2012). Because of this, we believe that we are justified in holding the ice temperature (ice viscosity) constant throughout the simulation. As discussed above, since dynamic thickening due to past climate forcing of the ice sheet is not included in our simulation, we expect it to be captured in the difference between GRACE-JPL and the model simulations.

Third, the paper does not convince me that the difference between modeled and observed trends can be attributed well on a regional scale as the uncertainties in input

and observed fields are too large (especially SMB, but also bedrock elevation, in addition to errors in the GRACE field that is moreover spatially poorly resolved) and the simplifications in the model setup and initialization procedure are too important to reach solid conclusions.

The strategy of our manuscript was to quantify the uncertainties in GRACE-JPL and the uncertainties in SMB, because they indeed do have significant uncertainties at a 300 km spatial scale/monthly temporal scale (i.e., the spatiotemporal scales of interest for our study). It is of utmost importance that these errors are rigorously defined. Upon comparison, it is clear that there are regions, and during specific times of the year, where observations and the model results differ outside the bounds of the calculated uncertainty, which allows us to then probe into attribution of this signal. We believe that the methods presented here are among the most rigorous that have been applied for this type of comparison, and that our conclusions are justified based on our calculated uncertainties.

Apart from these reservations, partly confirmed by the authors when differentiating between different regions, I found the results section hard to swallow as it is much too long, not well organized, and lacks synthesis.

Based on suggestions from referee 1, we have removed many of the technical details from the main text of the manuscript, have organized the results and the discussion into sections, and have reworked the introduction to focus on the three key components of this comparison (SMB models, ice sheet model, and GRACE). We believe that these changes address the concerns mentioned here.

I don't think the problems with the paper as it stands now (focus and length of the paper, problems with the model setup and the initialization, not considering pre-1846

contributions to ice evolution, messy conclusions, ...) can be fixed with only a major revision.

During manuscript revision, we have reduced the technical details in the main text of the paper and have tried to better focus on our goal of assessing regional mass variability in Greenland (both observed and modeled). We do not believe that our model setup has any notable problems, but we do admit that there are limitations as a result of our assumptions. These limitations are discussed in detail in the manuscript, and we believe the steady-state spinup offers an opportunity to quantify ice sheet changes in response to pre-1840 climate forcing of the paleo ice sheet, particularly in the interior of the ice sheet. Such background trends should manifest as trends in GRACE, and not in temporal variability. By separating trend and annual amplitudes, our methods should expose significant shifts in trend that are related to dynamic background trends. For example, we discuss that the Northwest is out of balance, as GRACE-JPL exhibits mass loss even during the winter months.

A few other comments Abstract, p.1, lines 17, 19: 'transient' processes, 'transient' dynamics: what is meant with 'transient' in this context?

The term transient has been changed to "temporally evolving", which we believe is a more clear description of the processes we are referring to here.

p. 2, line 2: 'Sto': the referencing is somewhat sloppy. Presumably, 'Sto' is Stocker et al., 2014. This kind of referencing occurs in many other places in the manuscript. More generally, when referring to the IPCC work, it is recommended to refer to the individual chapters.

As we have responded to referee 1, we do not know why this reference did not render correctly during the TCD upload process. This has been updated, and the reference has been updated to refer to the specific chapter of interest (*Church and White*, 2011).

p. 4, line 16: who are A. et al.?

As noted to referee 1, this reference is actually correct. Mr. A had the very unfortunate life circumstance of having a last name that was uniquely given by the first letter of the alphabet! We share in your sentiments that this is quite unfortunate for Mr. A.

p. 4, line 6: what is meant with 'surface mass variations'? Do the authors perhaps mean surface elevation changes? If so, are these expressed in ice equivalents, i.e. in mass changes? Or do the authors mean 'ice mass' variations as opposed to the GIA contribution to GRACE?

We believe the referee is referring to a phrase that is located on p. 4, line 16. GRACE observes total mass variations: these include mass variations on the surface of the earth (water/ice), above the surface of the Earth (atmosphere), and below the surface of the Earth (glacial isostatic adjustment). By 'surface mass variations', we mean exactly that: mass variations on the surface of the Earth. These include not only ice mass variations, as you suggest, but also mass variations due to snow and water. By removing glacial isostatic adjustment from the GRACE signal, we remove the solid Earth mass variations, and are left with only "surface mass variations". We prefer to keep the verbiage as is.

p. 5: why is only a 9-year period chosen for averaging SMB. Isn't that a bit short considering the inherent variability of climate conditions over the ice sheet?

We have reworded Section 3.2 to be more explicit about why this 10-year period was choses as our reference period. Most importantly, it is the overlap period of all 3 SMB models, that also happens to exists during the 1971-1988 steady-state period noted by *Rignot et al.* (2008). This is bound by the beginning of the ERA-I reanalysis period. In order to avoid inconsistencies between continuity strategies for the SMB products, we decided not to use any products forced with ERA-40. We were satisfied that the ice sheet remained in a near steady-state during the 1840-1900 ISSM simulation, suggesting that decade is reasonably similar in climatology to the first 60 years of simulation. Overall, we feel the choice is justified and that the mean climatology chosen is reasonably representative of the mean SMB for the beginning of the historical spin up period.

p. 6, line 9: 'ISSM Greenland observed velocities': do you perhaps mean 'modelled velocities'?

We believe the referee is referring to a phrase that is located on p. 9, line 6. If so, thank you for pointing this out. This was an important mistype that his not been fixed as suggested.

p. 29, Fig. 2: the figures are too small to distinguish the patterns. The blue-red colour scale does not allow to differentiate much.

All the plots in this figure have been enlarged (and are now located in the supplement), so that we only present 2 panels per page. We believe that these figures, now being larger, are easier to distinguish in terms of the color bar and spatial patterns.

p. 31, Figure 5: the colour legend seems to be for the difference plot (panel C) only, but not for panels A and B. 'Mascons' with the same colour do not always appear green in panel C. Otherwise there is a problem with the color scale.

[Figure]

We doublechecked Figure 5 and found consistency in A, B, and C. Colors are consistent between the three plots. Figure 5 is now updated, as we have updated all our results to include a new version of GRACE and the L1L2 version of ISSM results. We have ensured that the colors are also consistent in this new version of the manuscript.

**Supplement:**

[revised manuscript text omitted]

**Supplement Figures**

Model Mesh

[Figure]

**Figure S1.** The ISSM Greenland mesh for (A) the entire ice sheet and (B) the northwest margin.

[Figure]

**Figure S2.** (A) Modeled surface velocities (m/yr), from the relaxed ISSM Greenland, on a log scale and (B) the departure of modeled surface velocities from observations (Rignot and Mouginot, 2012).

**Modeled Ice Thickness (m)     Model−Obs Thickness (m)**

[Figure]

**Figure S3.** (A) Modeled ice thickness (m), from the relaxed ISSM Greenland and (B) the departure of modeled thicknesses from observationally-based data (Morlighem et al., 2014b).

[Figure]

**Figure S4.** Spatial representation of trend in surface mass from 2003-2009 as estimated from (A) GRACE-JPL and (B) ICESat altimetry (Csatho et al., 2014), and (C) the difference: GRACE-JPL - ICESat.

[Figure]

**Figure S5.** Cumulative mass from 2003-2012 for (A) all of Greenland and (B) the Greenland Interior, comparing observations from GRACE (GRACE-JPL), with model outputs: ISSM over the Greenland Ice Sheet (ISSM-GrIS), SMB anomalies over the Greenland Ice Sheet (SMB-GrIS), ISSM-GrIS with mass from the periphery (ISSM-GrIS+P), and ISSM-GrIS+P for each individual SMB forcing (ISSM-GrIS+P MAR, ISSM-GrIS+P RACMO, ISSM-GrIS+P BOX).

[Figure]

**Figure S6.** (A) Change in modeled mean annual surface velocities (m/yr) and (B) change in model ice thicknesses (m) during the 10-year ISSM simulation period (2003-2012). Model output is presented as the mean of three different ISSM simulation runs (ISSM-GrIS BOX, ISSM-GrIS MAR, and ISSM-GrIS RACMO).

[Figure]

**Figure S7.** (A) Total dynamic thickness change (difference between the cumulative mass contribution from the SMB forcing anomalies and the total thickness change) simulated by ISSM Greenland (2003-2012); (B) change in surface slope during the simulation; and (C) change in the magnitude of the driving stress over the same period. Model output is presented as the mean of three different ISSM simulation runs (ISSM-GrIS MAR, ISSM-GrIS RACMO, and ISSM-GrIS BOX).

[Figure]

**Figure S8.** Spatial representation of trend in surface mass for the Greenland periphery as estimated from (A) RACMO, (B) BOX, and (C) MAR.

[Figure]

**Figure S9.** Interior mascons, total cumulative mass timeseries for GRACE-JPL, SMB-GrIS, and ISSM-GrIS (including the mean and results from the individual simulations of ISSM-GrIS MAR, ISSM-GrIS RACMO, and ISSM-GrIS BOX). Also included is the residual between GRACE-JPL and ISSM-GrIS (green).

[Figure]

**Figure S10.** Southwest mascons, total cumulative mass timeseries for GRACE-JPL, SMB-GrIS, ISSM-GrIS (including the mean and results from the individual simulations of ISSM-GrIS MAR, ISSM-GrIS RACMO, and ISSM-GrIS BOX), and ISSM-GrIS+P. Also included is the residual between GRACE-JPL and ISSM-GrIS+P (green).

[Figure]

**Figure S11.** Same as Fig. S10 but for Northeast mascons.

[Figure]

**Figure S12.** Same as Fig. S10 but for Southeast mascons.

[Figure]

**Figure S13.** Same as Fig. S10 but for Northwest mascons.

[Figure]

**Figure S14.** Spatial representation of differences in mean annual amplitude from 2003-2012 between various combinations of model spinup and the ISSM-GrIS MAR3.5.2 presented in the manuscript (i.e. MAR3.5.2 forced by ERA-I reanalysis and BOX SMB used as the reference relaxation climatology, $\overline{SMB}$). Comparison runs include: (A) ISSM-GrIS MAR3.5.2, where MAR3.5.2 is forced with NCEP1 reanalysis; (B) ISSM-GrIS MAR3.5.2, where MAR3.5.2 SMB is used for $\overline{SMB}$; and (C) ISSM-GrIS MAR2.0, where MAR2.0 SMB (forced with ERA-I reanalysis) is used for $\overline{SMB}$. Results are less sensitive to variations in RCM forcing (A) and choice of spinup product (B) than to RCM version (C).

[Figure]

**Figure S15.** Spatial representation of differences in mass trend from 2003-2012 between various combinations of model spinup and the ISSM-GrIS MAR3.5.2 presented in the manuscript (i.e. MAR3.5.2 forced by ERA-I reanalysis and BOX SMB used as the reference relaxation climatology, $\overline{SMB}$). Comparison runs include: (A) ISSM-GrIS MAR3.5.2, where MAR3.5.2 is forced with NCEP1 reanalysis; (B) ISSM-GrIS MAR3.5.2, where MAR3.5.2 SMB is used for $\overline{SMB}$; and (C) ISSM-GrIS MAR2.0, where MAR2.0 SMB (forced with ERA-I reanalysis) is used for $\overline{SMB}$. Results are less sensitive to choice of spinup product (B) than to variations in RCM forcing (A) or to RCM version (C).

---

## Referee Report (RR1)

Summary and comments on the revised manuscript entitled

**Application of GRACE to the assessment of model-based estimates of monthly Greenland Ice Sheet mass balance (2003-2012)**

presented on 09.06.2016
by

N.-J. Schlegel et al.

**Summary**

The authors combine three model records of surface mass balance (SMB) over the Greenland ice sheet (GrIS) with mass change observations from the Gravity Recovery and Climate Experiment (GRACE). The aim is to exploit these records for validating the initialisation of a state-of-the-art ice-flow model. Major effort is therefore put in discerning regional differences in the GRACE mass loss signal by relying on improved processing of the gravimetry data. In addition, the initialisation strategy for the ice-flow model is two-fold. First, inverse techniques are used to infer ice properties as ice temperature, the viscosity parameter and the sliding coefficient. Second, the ice-flow model is run forward over ~50 kyr under constant climate to guarantee consistency between ice flow and the prescribed SMB. Prognostic output over the period 2003-2012 is compared to the regional mass change pattern observed by GRACE. Many aspects of the presented initialisation are state-of-the-art which supports that ice-flow models are able to reproduce and explain the dynamic state of the ice sheet. The results show a strong underestimation of the observed mass change (Fig. 3A), which is explained by not accounting for fast processes in the ice-flow model. This might sound excusable but I fear that this puts the entire application of the flow model into question, as the major results do not significantly differ from a pure RCM SMB and GRACE data comparison.

**Main comment**

While reading the manuscript, the above mentioned concern gradually became more and more manifest. Most findings are actually comparisons between cumulated SMB and GRACE data. Any conclusion/results which include the ice-flow model are not very significant (see detailed comments below). The authors even state themselves that relevant dynamic processes are not included in ISSM. Therefore, the remaining slow dynamic effect, they see, is small on the overall mass budget (Fig. 3). It is no surprise that many conclusion basically stem from a pure comparison between SMB modelling and GRACE mass change signals. The ice-flow model, as you present it, is not able to explain the recent dynamic response of the GrIS as it is not solely driven by SMB changes and resultant geometric adjustment. This is in fact what you calculate. In consequence, your ISSM mass budget calculation only explains 64% of the GRACE signal (Fig. 3). Ice dynamics is key here, but you need to include a process that explains the discharge increase through the marine terminated glaciers. Otherwise the flow model only serves to redistribute mass. The fundamental problem is that the extensive introduction of the flow model initialisation insinuates that you actually gain in reproducing the GRACE observations. But as you state yourself this is not really the case.

If the authors intend to revise or re-submit their manuscript, I urge them to strongly reformulate their central objectives and/or re-structure the article accordingly. I see two options, either exclude the entire flow model application or find a defensible parameterisation to simulate the recent ice discharge increase.

**Specific comments**

P1L4 '[...] scarcity of highly resolved [...] improvements in spatial resolution and noise reduction of monthly global gravity fields.' You speak of high-resolution validation data which is missing and you put it into context with the improved GRACE data. GRACE processing clearly improved but I'm not sure if you should insinuate by your wording that the 300-km GRACE posting is a high-resolution, in the light of almost automatically processed DEM and velocity data which is now available.

P1L9-P1L10 As you say that no fast dynamic processes are included in your model, you actually see difference between the GRACE signal and cumulated SMB values from regional climate models (RCM).

P1L11 You did not force your model with the SMB from three RCMs. You used one RCM during spin-up and then you added anomalies in the more recent period. This is a fundamental difference.

P1L18-L19 Differences to GRACE are the full ice-dynamic signal, which is not modelled, and uncertainties in the RCMs.

P2L6-L7 If you speak of uncertainties in SMB projections, you should also mention the relevance of melt-water retention and refreezing.

P2L7-L9 Uncertainties in projecting future ice discharge: if you are speaking of short-term variations in ice discharge, I'm not sure if the link to SMB changes is most relevant. This link evokes the notion of slow geometric adjustment and gradual flow adjustment (over millennia). This process is well understood. Fast changes in ice discharge, however, are triggered at the margin (within some tens of kilometres). SMB variations can be one factor but other processes might be much more relevant (you mention them all in the next paragraph).

P2L10 'An alternative [...]' There you raise a big point. Extrapolating trends is in fact highly speculative and I prefer a well-initialised model setup for such projections. So I would not use the word 'alternative'. I would state that flow-model projections are also a sort of extrapolations but well-informed by first physical principles of ice volume evolution.

P2L18-L21 This criticism on missing processes in current ice-flow models invokes the notion that your approach will be able to address these issues. I therefore suggest reformulation because most of the criticism applies to the presented approach.

P2L25 I do not agree with you here. Most state-of-the-art ice flow model, used for the ice2sea projections, account for longitudinal stress coupling (CISM, ISSM, PISM, Pollard, Elmer/Ice, etc.). The somewhat simpler SIA models are mostly applied in paleo applications, where long time-scales justify to some degree their applicability.

P3L8 You introduce this paragraph with: 'Another pressing issue is the lack of observational data for model evaluation.' The previous paragraph is already about the sparse observational record and unresolved issues linked to it. Please reformulate.

P4L28 What are the GAE and GAF products?

P6L11 This is a nice fit between observed and modelled velocities. For quantitative comparison, could you please give some measure like a root-mean-square error. Then one can directly compare to similar assimilation strategies (the ice sheet-wide RMS is for example given in Arthern et al., 2015).

P8L17 What is this CRI filter?

P9L10-L11 Your setup does not describe all components of the MB. You miss a process that can explain fast dynamic changes along the marine terminated outlet glaciers. Otherwise, the only dynamic response comes from the slow response to the geometric adjustment resulting from the SMB changes.

P9L11-L12 It is nice to see that the seasonality of the mass change signal is reproduced, reflecting the applicability of the RCMs.

P12L31-P13L1 Here, you formulate my main concern yourself. In other words, ISSM-GrIS mass changes do not differ much from the integrated SMB. The relevant dynamic process is not captured. It is highly controversial how this acceleration is triggered so I suggest to rely on parameterisations. Often basal sliding was increased by a constant factor or linked to changes in ocean temperature.

P21L7 I repeat myself but could you please provide a RMS deviation between modelled and observed surface velocity magnitudes.

Figure 3A The figure shows the total mass budget of the Greenland ice sheet between 2003 and 2013. I understand that you want to put the GRACE signal next to the modelled mass changes. First you have the SMB-GrIS curve (dashed line), which shows a somewhat reduced signal. If I understand it correctly, SMB-GrIS is the cumulated SMB anomaly. I did not quite get if this is for any specific RCM or a mean evolution. Anyhow, my question points at the difference between the SMB-GrIS and ISSM-GrIS. Again if I understood the setup correctly, ISSM-GrIS accounts for ice discharge (D) changes. At moments in the recent decade, observed changes in D were forwarded to have explained a large portion of the mass loss from GrIS. Yet the difference between ISSM-GrIS and SMB-GrIS are almost negligible. The reason for this is that you do not account for processes which explains fast dynamic response. Your model simply redistributed mass, except at the marine ice fronts. The slight dynamic reduction of the SMB mass loss signal, which you explain by the slow geometric adjustment (slopes, driving stress, etc.), has already been analysed in depth in the literature. As the title and abstract of the manuscript do not suggest that this small dynamic (long-term) effect is the target of your study, I fear that your model setup is not chosen appropriately.

---

## Author Response (AR2)

Again, we would like to thank the referees for giving their time to raise important discussion points. Most importantly, we recognize that there is a misunderstanding that we are attempting to validate the ice flow model (specifically, an ice flow model that does not include representation of key dynamic processes believed to be responsible for a significant portion of mass loss during our study period), and we hope the new version of the manuscript alleviates this issue.

Below, we address the specific comments and suggested modifications raised by the referees. For those comments that required updates to text, we enclose a paper draft with corresponding changes in red.

**1 Anonymous Referee 1**

Re-Review of TC manuscript 2015-224 now titled "Application of GRACE to the assessment of model-based estimates of monthly Greenland Ice Sheet mass balance (2003-2012)" by Schlegel et al.

The authors have addressed my comments from an earlier revision and the manuscript has clearly improved from its last version. It is, however, still not ready for publication. I have found many issues of imprecision in the language and in the descriptions that need to be addressed before publication. While the manuscript has gained in clarity from the restructuring, it needs another round of careful revision by the authors to clean up remaining problems. I have listed some detailed comments below, but my estimate is that it needs a bit more work than checking off that list.

We have reviewed the manuscript and have revised it according to the comments raised below, including changes to almost every figure. In addition, we have attempted to expand the introduction to better encompass details about model and GRACE uncertainties. We have also read through the manuscript and have edited any vague or confusing language. We hope that these changes address your concerns. Thank you for your suggestions.

General comments:

The use of minus signs in product names (e.g. ISSM-GrIS+P) is confusing, because they also appear in difference calculations. You could use underscores instead, or find another solution.

Thank you for this suggestion. The dashes in the ISSM and SMB product names have been changed to underscores for clarity.

Figures: For me Figure 8-11 are clearly supplementary material. A move would help to make the main paper a bit lighter.

While we do understand the reason for this suggestion, it is important to point out that Figures 8-11 form the basis for the majority of our discussion topics. In fact, we feel that these figures, in particular, most strongly support our argument that the temporal variability of the dynamic mass signal (representative of mass change forced by ocean and hydrologically driven processes) is significant enough to be captured by GRACE on a monthly timescale. For this reason, we disagree that Figures 8-11 are supplementary material, and we have chosen to keep them included as figures in the main paper.

While not crucial for the interpretation, some attempts to make plots more comparable seem in place. Projections and aspect ratios e.g. differ considerably between figures. The GrIS appears in 5-6 different "projections". S2, S3, S6 and S7 appear compressed, same for S9-S13. For projected grids 1:1 aspect ratio should be the standard.

All these figures have been updated with new color bars, and we now ensured that they all have the same aspect ratio. We hope that this will help for comparisons between figures and improve the overall clarity of the manuscript.

The choice of colour schemes for 2D plots in general seems poor to me and should be improved. The standard red-blue gives a good idea what is positive and what is negative but not much more. The jet colour scheme does not work well when the full colour range is not used. See also detailed comments per figure below.

We have updated most of the 2D figures to use a new red/blue colorbar, and in many cases (where needed) we have made the scales non-linear. The new colorbar extends deeper into the red and into the blue so that the colors are richer, and the coloring is more descriptive of the values being plotted. Thank you for your suggestions.

Other comments:

P1,L16 add "and other" before related.

"and other" has been added to this line of the abstract.

P1,L17 Maybe "predicted by the SMB models". SMB doesn't predict.

The abstract has been been updated to read "predicted by the models".

P2,L8-9 "accurate understanding of ice flow sensitivity to future changes in surface mass balance" is only one aspect. Sensitivity to ocean forcing and bed geometry needs to be mentioned here as well.

We have added a statement to include the physical processes believed to be driving rapid ice flow, in addition to ice flow response to surface mass balance.

P2,l15 Shannon et al. (2013) is not the right reference to allude to "model-based estimates [...] plagued with significant spread and large uncertainties". What may be true for the SEARISE experiments (own spinup procedure, different treatment of boundary conditions/model forcing) does not apply here. Instead of removing the reference I would suggest to attempt a more balanced description of the state of large-scale ice-sheet modelling. It is also misleading that the first missing process listed just below is basal lubrication, which was explicitly addressed by Shannon et al. (2013).

We have reworked the introduction to include more extensive descriptions of the uncertainties in ice sheet modeling experiments. We now discuss regional models of basal lubrication and ice/ocean interface and uncertainties associated with parameterization of these processes at a continental scale. In addition, we have a description of model spinups and discuss uncertainties associated with the choice of spinup procedure and model setup. We hope these new paragraphs help the reader better grasp the difficulties in quantifying uncertainties in ice sheet modeling.

P2,L21 Remove one closing bracket.

The parentheses at the end of this sentence have been removed.

P2,L26 "ice sheet surface deformation" sounds unusal. Please reformulate.

This sentence has been removed from the introduction in the reworked manuscript.

P2,l27 Move (Enderlin et al., 2014) to the end of the sentence to avoid consecutive brackets.

This citation has been moved to the end of the sentence for clarity.

P2,l32-33 Reformulate "onto high-resolution grids". Maybe "with high spatial resolution".

This sentence has been updated to read "at considerably high spatial resolutions".

P3,L1 There are many examples of references that show a good agreement between SMB models. A recent example is Vernon et al. (2013). I reiterate my call for a balanced view on the achievements of the modelling community.

Thank you for this suggestion. *Vernon et al.* [2013] is indeed a good reference for this. We have reworked this paragraph to discuss how RCMs agree on larger

regional scales, but have been found to disagree on local scales. We think this presents a more balanced view of the recent advancements that have been made in quantifying Greenland SMB.

This sentence has been split into two, as suggested.

P3,l20 "offer an opportunity *to improve* ... ", as estimates of GrIS MB have existed before.

"to improve" has been added to this sentence.

P3,l26 "conclusions" seems too strong of a wording for the amount of uncertainty. Same applies to P3,L34. Please reformulate.

We have replaced conclude with "hypothesize" in these two sentences, and on L34, have added "With consideration to the calculated error bars, ...".

We have removed this text as it is unnecessarily confusing. Instead, we have included a reference to the paper that describes the correction which we applied.

"formulation" has been changed to "solution".

P5,l24-28 BMB is not included in your model! Remove it from equation in l25, it's confusing.

BMB has been removed from this equation.

Here, we have inserted "ice" before thickness, and have also made this change throughout the manuscript.

"The" has been added before SMB.

P6,l2 There is an inconsistency in the argumentation here. You write between 1971-1988 the ice sheet had on average zero mass balance, but the reference period is a different one (1979-1988). Clearly, the ice sheet did not have a MB of zero each year between 1971-1988 but on average. Please clarify.

We have reworded this section to clarify that the ice sheet was believed to be near equilibrium during the 1970s and 1980s, as concluded by *Rignot et al.* [2008]. In addition, this paragraph has been reworked as to explain the logic behind choosing this period. Because we don't actually know the exact SMB of Greenland before 1840, we have attempted to choose this period based on observational evidence, that suggests the ice sheet was near steady state during a time period in the satellite era. The goal was to simulate this relatively constant period of mass balance, without introducing unrealistic model responses to inconsistencies in SMB. Because of this, we found it necessary to use only ERA-I forced RCM output, to avoid inconsistencies/jumps in the timeseries between 1978 and 1979. We are satisfied that the simulation results in a near steady-state condition during the 1970s and 1980s, as this was the goal our spinup procedure.

P6,l3 insert "net" before mass balance. This change may be needed in other places as well.

Instead of "net", we have inserted "total" into this sentence for consistency with the the terminology used in the rest of the manuscript. We have also added "total" where appropriate throughout the manuscript.

P6,l6 I miss a clear motivation why the ice sheet relaxed to 1979-1988 SMB may serve as a staring point for 1840. My understanding is that there may not be many good physical reasons (like assumed similarity between 1840 and 1979-1988) but better technical reasons. This should be clarified.

We have rewritten this paragraph to describe our criteria for the period that needs to be chosen for our model runs. These include 1) a period when the ice sheet was believed to have a mass balance close to zero and 2) a period during which all three SMB forcing products are defined. The second criterion, as pointed out, is technical. 1979-1988 is the period which satisfies our criteria [*Rignot et al.*, 2008]. We also point out that our results are insensitive to this choice. We believe that our reasoning should now be more clear to the reader.

P6,l6 The causality implied by the use of "Therefore" escapes me here. Is the relaxation you talk about the steady-state relaxation to 1979-1988 SMB or further relaxation between 1840 and present? You should quantify the effect of both the 140 year simulation and of the original relaxation.

This paragraph is now rewritten, and it should be clear that in this section, we are only discussing the 56k year relaxation to our reference climatology. This section is only meant to describe the relaxation methodology and not forward historical simulations. The results of the 140 year simulation are discussed in the following section and total mass balance for that simulation is quantified in Fig. 2.

P6,l9 Not clear now what "For this" relates to. Reformulate.

This statement has been updated to read "in this study".

P6,l9-10 "topography" is also not a "surface property". With surface property I would refer to the albedo or the type of snow cover but not a geometrical quantity like topography.

This sentence has been updated, and now states that we are trying to match the Greenland ice sheet present-day "state", instead of surface properties.

P6,l9-10 It is not correct to say that you "best capture [...] surface velocities at the beginning of our simulation". You may best capture velocities for the period 1979-1988 but not for 1840. For 1840 you make some kind of ad-hoc decision to use the present state. See also comment P6,l6 above.

This sentence has been updated to specify that we best capture the "present-day" velocities through our initialization techniques. The following sentences describe the limitations of this assumption, including the "lack of validation due to general uncertainty about the state of the ice sheet in 1840".

P6,l24 What is a "relative steady state"? Reformulate.

We have updated this sentence to read "a mass balance near zero" instead of "relative steady state".

P6,l30 It should be "we restart from the 1978 model state and force with SMB anomalies ..."

This statement has been updated as suggested here.

P6,l32 Confusing that the SMB product time ranges are given as [1979-2014], when you only use 1979-2012 as in the title and all the plots.

We have added a sentence at the end of the paragraph that reads "In this study, we will focus on comparison of historical simulation results from 2003-2012, which is the time overlap between the GRACE-JPL solution and the three SMB forcing products considered here." Note that we do show the results through 2014 for each mascon for reference, in the supplement.

P7,l1 Include "lateral" after "their"

The manuscript has been updated as suggested.

P7,l3 Include "change" after "thickness"

"change" has been added to this sentence.

P7,l11 For me, "mascon elements" should be used throughout the manuscript, but since you have argued against it, for consistency, it should be be called "mascons" here instead.

We agree and have made the suggested change. Thank you for catching this inconsistency in our language.

We have clarified this statement to read "To determine the modeled mass balance within the Greenland ice sheet boundary ...".

P7,l19 Since you say projected, mention which projection is used.

The text has been updated to clarify that the mascon boundaries are "projected into the ISSM Greenland coordinate projection (polar stereographic projection with standard parallel at 71°N and a central meridian of 39°W)".

p7,l32-33 Sentence "Note ..." should be mentioned earlier when masking is discussed (around p7,l18)

This sentence has been moved to the suggested location.

The text has been updated as suggested.

We have updated this statement as suggested.

"of" has been inserted as suggested.

p9,l19 Clarify over which area the thickening has been averaged.

We have added a statement of clarification that states that this number is averaged "within the area defined by the interior mascons".

This sentence has been split into two sentences and rewritten for clarity.

 This whole paragraph needs to be substantiated and reworked. Main problem: The ice sheet model is treated as a grey box that does something largely unknown to the SMB forcing.

This paragraph has been rewritten as suggested. We focus on discussing how the model is responding to SMB, and how this response contributes to our results (i.e. the model physics responds to the SMB forcing with a reduction of ice discharge).

Some suggestions are given here.

P10,l8 I believe the section title should be "Dynamic contribution" or similar.

We have updated the name of this section to "Contribution from SMB-driven Ice Dynamics"

P10,l10 "decreasing the rate of ice discharge" has to be much better explained. It is a reason for the difference between SMB and ISSM not the result. References have to be included that have described this effect before (e.g. Huybrechts and de Wolde, 1999; Gillet-Chaulet et al., 2012; Goelzer et al., 2013)

As suggested, this section has been rewritten. In the new version of this section, we have tried to better explain that marginal thinning and decrease in marginal velocity, both contribute a decrease in marginal mass flux (and ice discharge) in the simulations. Thank you also for pointing out the missing references. They were removed during the last revision, and have now been reinserted into this section.

Please describe what is actually happening physically and remove the following statements. P10,l11 "effects of the ISSM model"; l12 "use of the ice sheet model"; l13 "the model mutes"

In the new version of this section, we describe that marginal ice thins and also slows down (due to a decrease in driving stress). The result is less ice discharge into the oceans. In addition, we have have removed the offending phrases specifically mentioned above.

P10,L13 "suggesting"; l15 "likely related to" are vague statements given that a process is physically modelled and can be studied looking at the output.

We have removed these statements and have added more specific description of the process that is occurring along the margins in response to increased local runoff.

P10,l22 I find this interpretation confusing. The better agreement you refer to is likely referring to the reduced error range. However, the error range is only

compressed relative to the SMB results because of the dynamic response, which is non-linear. There may be a problem with the error propagation.

Because we only have three SMB models and three ISSM runs, the uncertainty bars around ISSM_GrIS and SMB_GrIS in Fig. 3 are indicative of the actual spread between these products. The error range is smaller in ISSM_GrIS than in SMB_GrIS because the spread between the three different products is smaller for the ISSM_GrIS runs than between the SMB products themselves. We have tried to clarify this in this section by explaining that this results from the feedback described above. A run with a SMB forcing that has a more negative SMB will thin more on the margins and have a larger feedback (i.e. decrease in ice discharge). The opposite is true for a run with a less negative SMB. We hope the explanation in the text now makes this clearer.

P10,l24 "close match" suggests that ISSM-GrIS and SMB-GrIS should match, which is not the case. Better to reformulate to "close similarity" or similar.

We have adopted the suggested terminology here of "close similarity". The text has been updated accordingly.

P13,l9 Why are problems with the initial velocity raised in this context? Not clear how the background state is expected to change the results here.

Point taken. These sentences have been removed from the discussion here, as they do not strengthen this paragraph.

P13,l11 Not clear what the continuation refers to, maybe "the Northwest loses mass".

The sentence has been updated as suggested.

p17,l25 "differences in" instead of "differences between"

"between" has been changed to "in" in this sentence.

p19,l13-16 Details about what numbers are used to encode the masks are irrelevant. Please remove.

These lines have been removed from the manuscript.

p19,l28-31 Details about what numbers are used to encode the masks are irrelevant. Please remove.

These lines have been also been removed from the manuscript.

Figures and Tables:

 The colours still appear muted in the figure compared to the legend and are not easily distinguished.

For this figure, we have removed the top color layer, so that the velocity colors are more distinct. The legend and the plot should now match exactly.

Figure 2 Include a black dashed line for reference total SMB of the BOX model

We have added a dashed black line to represent $\overline{SMB}$.

Figure 3 Include "cf." or "see" before "Sect. 5" Add a note for the different vertical scale in A compared to B in the caption.

These changes have been made to the caption of Fig. 3.

The figures are not of good enough quality. Zooming in to distinguish overlapping lines doesn't help. –¿ Panel A has too many lines to be legible. Maybe you could omit outlines of the error ranges to improve this. Otherwise, consider creating a new figure or split content with S5.

Thank you for pointing this out. Fig. 3A/B from the original submission were of much higher quality than those of the resubmission. This was a mistake on our part when recreating the figures for the L1L2 solution. We have corrected this in the new submission, and it is now helpful to zoom in to see the overlapping lines. Because this was not a concern noted by the referee on the first submission (and only the resolution was changed between the subsequence versions of the manuscript), we are hopeful that the improved resolution is an adequate response to this concern.

Shouldn't ISSM-GrIS have the same error bars as SMB-GrIS? The error range is shown to be smaller, which can not be the case. Please clarify.

As described above, we find that the feedback between marginal thinning is indeed responsible for a decrease in the "error bars" (or in this case "uncertainties", since it is more closely related to the spread in the three distinct model simulations than to actual errors - which we do not attempt to quantify in this study). We have tried to make this clearer in the section now titled "Contribution from SMB-driven Ice Dynamics". In addition, we have gone through the manuscript, and made sure that we are not using "uncertainty" and "errors" interchangeably in this context. We expect this will avoid further confusion.

Figure 4 The standard jet colour scale makes it difficult to distinguish positive and negative trends, especially as the upper 30 % of the scale are not used. Using the same scale for the absolute and the difference, while elegant for the

display may be too much of a limitation. Please consider revising the colour scales.

Thank you for the comment and suggestion. We agree that the jet color scale prohibited the distinction between positive and negative trends. Based on your suggestion, we have decided to remove the jet colorbar from all figures in this manuscript and have remade all figures with a red/blue color scale. We hope that this change alleviates many of your concerns with the figures. We have made a deliberate decision, however, to keep the colorscale of the difference plots the same as the absolute plots. This decision is made to put the magnitude of the difference in context with the magnitude of the signal. This contrast would be lost if we used different color scales. While shrinking the colorscale would, indeed, allow for better interpretation of the differences between GRACE_JPL and ISSM_GrIS+P, we feel that this is not the purpose of having these figures. Figures 8, 9, 10, and 11 serve the purpose of looking at specific differences within mascons along with their magnitudes. The more informative red/blue color scale should hopefully allay some of your concerns regarding this point.

Figure 5 Same comment as for Figure 4. Here the colour scale saturates at the top end, while the lower half is not used for A and B (all positive values). I suppose this was chosen to have a zero-centred bar for the differences. Please consider using two different scales for absolute values and differences and revising the colour scales.

This figure has been changed to a red/blue color scale. We still prefer to have a zero-centered color bar, so half of the colorscale is not used in Figures A and B. While we understand the concerns with this approach that the reviewer has raised, we feel that it is more important to be consistent in our presentation of figures throughout the manuscript (i.e. red is always positive, blue is always negative), such that the reader can easily interpret the results. Changing the colorscale of each figure to conform precisely to the scale in the figure would introduce unnecessary difficulties in quickly interpreting the results. Again, we additionally desire to keep the colorscale of the absolute and difference plots the same such that they can be placed in context with each other. Figures 8, 9, 10, and 11 should allow the reader to better understand the magnitude of the differences within individual mascons. The more informative red/blue color scale should hopefully allay some of your concerns regarding these two points.

Figure 6 Same comment as in Figure 5. Trend difference is mainly positive, and amplitude difference mainly negative. Half of the colour information is wasted here.

Please see our comments to Figures 4 and 5. The same applies here.

Figure 7 Similar to colour problems in Figure 6.

Please see our comments to Figures 4 and 5. The same applies here.

Table 1 Hard to read this table. Start caption with "Cumulative mass trends (Gt/yr)". Consider exchanging right and left side. Why not include a first row of informative categories in the table? Why refer to Fig 3A for the left entries, aren't they corresponding averages of the right entries?

The table has been updated as suggested. The Mean values have been placed on the right side of the table, and a first row has been added to describe the categories in the columns. In addition, the caption has been updated to "see" Fig3A as reference, and it now begins with the words "Cumulative mass trends (Gt/yr)".

Figure S1. Indicate the inset area of B in A.

The inset has been indicated on panel A, with a red rectangle.

Figure S2. Consider non-linear colour scale in B.

We have updated this plot to use a non-linear colorscale.

Figure S2. Consider more informative colour scale in B.

This plot now uses the new, deeper red/blue colorbar.

Figure S4. See similar problem with colour scale and map as in figure 4 and 5.

Please see our response to Figure 4 and 5, above. The same applies here. For this particular figure, we especially feel compelled to keep the colorscale the same between the absolute and difference plot. We wish to convey to the reader that GRACE and ICESat agree particularly well at the spatial scale of an individual mascon.

Figure S5. Too many lines. This figure is largely similar to Figure 3. The added value is the different SMB products. Maybe remove GRACE, ISSM-GrIS and SMB-GrIS for better legibility. Otherwise rename GRACE to GRACE-JPL to make legend in line with text and figure 3.

We agree - the added value of this figure is to have the results of the ISSM run for the three different SMB products. To highlight these three results in particular, and to make them more visible, we have reduced the width of the other lines (since they are already found in Fig.3). We have also updated the legend to read GRACE_JPL so that it matches Fig. 3.

It should be pointed out in the text that BOX is very similar to the three-model average except for the last year.

In the results section "Greenland Cumulative Mass", we have added a statement that highlights the fact that ISSM_GrIS+P MAR has the largest negative trend, ISSM_GrIS+P RACMO has the smallest negative trend, and ISSM_GrIS+P BOX is similar to mean during the majority of the study period.

Figure S6. Consider more informative colour scheme for A. Delta velocity should probably be displayed on a non-linear scale. I suppose the Northern outlets exceed -10 m/yr difference. How was the limit motivated, to fit with scale in B?

Both plots now use the new, deeper red/blue colorbar, and both scales are now non-linear. The limits have also been updated to reflect the numerical extent of the parameters being plotted.

Consider more informative colour scheme for B.

The red/blue color scale has been enhance to include a deeper red and a deeper blue. We believed that this improves the figures substantially, as more contrast is visible.

Figure S7. Consider more informative colour schemes and harmonise (A is different from B and C for no apparent reason)

We have updated this figure with a new red/blue colorbar, so that the contrast in more visible. The new colorbar extends deeper into the red and into the blue so that the colors are richer, and the coloring is more descriptive of the values being plotted. All plots now have the same colorscale.

Figure S8. If I understand correctly, the trends should not be zero, but undefined for interior mascons. If so, this should be represented in the figure (maybe grey or white instead of green).

Yes, you are correct. Thank you for this suggestion. For plots that refer to the periphery, all interior mascons now appear white.

**2 Anonymous Referee 3**

Summary

The authors combine three model records of surface mass balance (SMB) over the Greenland ice sheet (GrIS) with mass change observations from the Gravity Recovery and Climate Experiment (GRACE). The aim is to exploit these records for validating the initialisation of a state-of-the-art ice-flow model. Major effort is therefore put in discerning regional differences in the GRACE mass loss signal by relying on improved processing of the gravimetry data. In addition, the initialisation strategy for the ice-flow model is two-fold. First, inverse techniques are used to infer ice properties as ice temperature, the viscosity parameter and the sliding coecient. Second, the ice-flow model is run forward over 50 kyr under constant climate to guarantee consistency between ice flow and the prescribed SMB. Prognostic output over the period 2003-2012 is compared to the regional mass change pattern observed by GRACE. Many aspects of the presented initialisation are state-of- the-art which supports that ice-flow models are able to reproduce and explain the dynamic state of the ice sheet. The results show a strong underestimation of the observed mass change (Fig. 3A), which is explained by not accounting for fast processes in the ice-flow model. This might sound excusable but I fear that this puts the entire application of the flow model into question, as the major results do not significantly differ from a pure RCM SMB and GRACE data comparison.

While a good match between our ice sheet model results and GRACE observations would (to a certain extent) validate the use of our initialization procedure, this is not the main goal of our manuscript. This misunderstanding is likely due to a vagueness in our introduction. Indeed, we did not explicitly define our goals upfront for the reader. While the inclusion of an ice sheet model in a regional GRACE comparison is a new element, we are not attempting to use the analysis to validate the model. Instead, we are using it as a tool to quantify a portion of the ice flow response (response to historic SMB) and to quantify the uncertainties associated with this ice flow response and with the

SMB forcing. Admittedly, we do use results of our comparison to make statements about our initialization assumptions and to hypothesize about the what processes might be responsible for the differences between the modeled MB and GRACE. In addition, in many areas, we believe that physically-based models should be incorporated at the continental scale. We have reworked the introduction to describe our goals upfront, and we hope that it is clearer why we have chosen to take the presented approach.

**Main comment**

While reading the manuscript, the above mentioned concern gradually became more and more manifest. Most findings are actually comparisons between cumulated SMB and GRACE data. Any conclusion/results which include the ice-flow model are not very significant (see detailed comments below). The authors even state themselves that relevant dynamic processes are not included in ISSM. Therefore, the remaining slow dynamic effect, they see, is small on the overall mass budget (Fig. 3). It is no surprise that many conclusion basically stem from a pure comparison between SMB modelling and GRACE mass change signals. The iceflow model, as you present it, is not able to explain the recent dynamic response of the GrIS as it is not solely driven by SMB changes and resultant geometric adjustment. This is in fact what you calculate. In consequence, your ISSM mass budget calculation only explains 64% of the GRACE signal (Fig. 3). Ice dynamics is key here, but you need to include a process that explains the discharge increase through the marine terminated glaciers. Otherwise the flow model only serves to redistribute mass. The fundamental problem is that the extensive introduction of the flow model initialisation insinuates that you actually gain in reproducing the GRACE observations. But as you state yourself this is not really the case.

We have reworked the introduction to better describe our goals for this manuscript, which include quantifying uncertainties associated with model-based estimates of mass balance (i.e, SMB forcing and ice flow response to SMB forcing). We represent these specific components of the mass balance with models, because they have minor uncertainties compared to other dynamic processes. We have decided to not represent the other ice dynamic processes through modeling, because in many cases, the uncertainties are large (or even unquantifiable). We hope that the new introduction does not insinuate validation of an ice sheet model, but instead, clearly describes our intentions to use this tool to reduce uncertainties in the partitioning of mass balance between SMB and different processes responsible for driving changes in discharge. Results of this comparison allow us assess where/when the SMB products may have errors and where/when it is important to have physical representations of the processes not included in our ice sheet model simulation. Because of this, we believe that this statistically-based assessment, at higher spatial and temporal

resolution that has been previously attempted, will be helpful to the scientific community.

If the authors intend to revise or re-submit their manuscript, I urge them to strongly reformulate their central objectives and/or re-structure the article accordingly. I see two options, either exclude the entire flow model application or find a defensible parameterisation to simulate the recent ice discharge increase.

Per your suggestion to reformulate our objectives, we have attempted to rework the introduction in order to better describe the goals of this manuscript. We are reluctant to remove the ice sheet model from our analysis, or to introduce a parameterization for ice-ocean interaction/hydrological processes, as doing so would increase uncertainty in our model results. In light of the goals of this investigation, we do not see an advantage to doing either. We hope the new version of the manuscript makes this clearer.

Specific comments

P1L4 "[...] scarcity of highly resolved [...] improvements in spatial resolution and noise reduction of monthly global gravity fields." You speak of highresolution validation data which is missing and you put it into context with the improved GRACE data. GRACE processing clearly improved but I'm not sure if you should insinuate by your wording that the 300-km GRACE posting is a high-resolution, in the light of almost automatically processed DEM and velocity data which is now available.

We have removed the reference to high resolution validation data, as this statement was clearly misleading. We did not originally intend to insinuate that the 300 km GRACE data is "high resolution" with respect to other measurement types (altimetry, DEM, velocity), because, as you point out, it has quite coarse resolution when compared to these other measurement types. The advantage to using GRACE is that it is continuous in space and time, and, more importantly, provides the only direct estimate of mass, which is the variable we are interested in for Greenland.

P1L9-P1L10 As you say that no fast dynamic processes are included in your model, you actually see difference between the GRACE signal and cumulated SMB values from regional climate models (RCM).

Yes, we agree, SMB dictates much of the difference. But here, we investigate multiple RCM's and beyond, with the goal of quantifying uncertainties - therefore, we claim to make the comparison between GRACE and "state-of-the-art, high-resolution models", which includes both RCM and ISSM. Note that ISSM SMB-driven dynamics contributes up to $+21$ Gt/yr to the trend and our RCM-based calculation of peripheral mass balance contributes up to -63 Gt/yr. These

contributions are not trivial, and contribute a fair amount to the uncertainty (which we attempt to quantify). We have now specified in the abstract that we compare models and GRACE-JPL within error and uncertainty bounds, to make the extent of our effort clearer.

P1L11 You did not force your model with the SMB from three RCMs. You used one RCM during spin-up and then you added anomalies in the more recent period. This is a fundamental difference.

Here, we now specify that we force our model with anomalies from three different regional climate models, as is customary in GRACE/SMB comparisons. If we used the total SMB forcing instead, we would be introducing even more uncertainty into the ice sheet model that is non-linear and difficult to quantify.

P1L18-L19 Differences to GRACE are the full ice-dynamic signal, which is not modelled, and uncertainties in the RCMs.

As discussed in the paper, as well as in The Cryosphere Discussions with Referee 1 and Referee 2, while ISSM does not succeed in modeling the full ice-dynamic signal due to missing physical processes in the model, it does in fact model the SMB-driven ice-dynamic signal.

P2L6-L7 If you speak of uncertainties in SMB projections, you should also mention the relevance of melt-water retention and refreezing.

Here, we now specify that surface runoff includes the processes of "melt-water retention and refreezing".

P2L7-L9 Uncertainties in projecting future ice discharge: if you are speaking of shortterm variations in ice discharge, I'm not sure if the link to SMB changes is most relevant. This link evokes the notion of slow geometric adjustment and gradual flow adjustment (over millennia). This process is well understood. Fast changes in ice discharge, however, are triggered at the margin (within some tens of kilometres). SMB variations can be one factor but other processes might be much more relevant (you mention them all in the next paragraph).

This is a good point. We now mention that understanding the physical processes behind these transient dynamic processes are important in making future predictions about Greenland mass balance.

P2L10 "An alternative [...]" There you raise a big point. Extrapolating trends is in fact highly speculative and I prefer a well-initialised model setup for such projections. So I would not use the word "alternative". I would state that flow-model projections are also a sort of extrapolations but well-informed by first physical principles of ice volume evolution.

We have removed the wording that insinuated that ice modeling is an alternative to extrapolations, and have changed the wording to contrast that numerical ice sheet models are the only method guided by first physical principles, unlike extrapolations.

P2L18-L21 This criticism on missing processes in current ice-flow models invokes the notion that your approach will be able to address these issues. I therefore suggest reformulation because most of the criticism applies to the presented approach.

This section of the introduction has been rewritten, in order to better explain the state of ice sheet modeling and the uncertainties associated with modeling a continental ice sheet. The goal here is not to improve upon models, but to use them to better quantify the uncertainties associated with partitioning Greenland MB into discharge and SMB, spatially and temporally. We hope the reworked introduction makes this goal more clear.

P2L25 I do not agree with you here. Most state-of-the-art ice flow model, used for the ice2sea projections, account for longitudinal stress coupling (CISM, ISSM, PISM, Pollard, Elmer/Ice, etc.). The somewhat simpler SIA models are mostly applied in paleo applications, where long time-scales justify to some degree their applicability.

This sentence has been removed from the introduction, and we have instead included two new paragraphs that describe uncertainties associated with ice dynamic processes and with choice of spinup procedure.

P3L8 You introduce this paragraph with: "Another pressing issue is the lack of observational data for model evaluation." The previous paragraph is already about the sparse observational record and unresolved issues linked to it. Please reformulate.

We have updated this sentence to specify that here we are referring to the lack of observational data for ice sheet model evaluation. In the previous paragraph, we also add clarification that we are referring to the observational data required to validate SMB surface models.

P4L28 What are the GAE and GAF products?

We have removed this text as it is unnecessarily confusing. Instead, we have included a reference to the paper that describes the correction which we applied.

P6L11 This is a nice fit between observed and modelled velocities. For quantitative comparison, could you please give some measure like a root-mean-square error. Then one can directly compare to similar assimilation strategies (the ice sheet-wide RMS is for example given in Arthern et al., 2015).

As suggested in your comment below, we have added a measurement of RMSE (90 m/yr) in the "Details of ISSM spinup and forward simulation" section. This should aid in a comparison of velocities with other studies on a continental scale.

P8L17 What is this CRI filter?

The CRI filter is first described in Section 2 in the text. It is a filter that separates between "land" and "ocean" mass within mascons that span coastlines.

P9L10-L11 Your setup does not describe all components of the MB. You miss a process that can explain fast dynamic changes along the marine terminated outlet glaciers. Otherwise, the only dynamic response comes from the slow response to the geometric adjustment resulting from the SMB changes.

Yes, we agree with this statement. The experiment was constructed in this way so that we could regionally quantify the magnitude and temporal variability of the mass balance directly related to SMB, and the uncertainty that may be associated with each.

P9L11-L12 It is nice to see that the seasonality of the mass change signal is reproduced, reflecting the applicability of the RCMs.

We agree, and a main objective of this study is to highlight when and where this is not the case.

P12L31-P13L1 Here, you formulate my main concern yourself. In other words, ISSM-GrIS mass changes do not differ much from the integrated SMB. The relevant dynamic process is not captured. It is highly controversial how this acceleration is triggered so I suggest to rely on parameterisations. Often basal sliding was increased by a constant factor or linked to changes in ocean temperature.

We also agree with this. The acceleration is highly controversial and largely uncertain. By including a parameterization, we would introduce a significant amount of unknown uncertainty, and we do not believe that it would improve our analysis or change the conclusions presented here. Instead, we argue that our results suggest that the physical processes associated with these dynamics, including the seasonal-to-monthly variability, need to be understood locally before continental ice sheet models will match observations.

P21L7 I repeat myself but could you please provide a RMS deviation between modelled and observed surface velocity magnitudes.

As commented above, we have now added an estimate of RMSE (90 m/yr) in the "Details of ISSM spinup and forward simulation" section.

Figure 3A The figure shows the total mass budget of the Greenland ice sheet between 2003 and 2013. I understand that you want to put the GRACE signal next to the modelled mass changes. First you have the SMB-GrIS curve (dashed line), which shows a somewhat reduced signal. If I understand it correctly, SMB-GrIS is the cumulated SMB anomaly. I did not quite get if this is for any specific RCM or a mean evolution.

SMB_GrIS is defined as "the mean total SMB anomaly RCM-derived forcing over the ice sheet". We have added clarification in the caption of Fig. 3 to this effect.

Anyhow, my question points at the difference between the SMB-GrIS and ISSM-GrIS. Again if I understood the setup correctly, ISSM-GrIS accounts for ice discharge (D) changes. At moments in the recent decade, observed changes in D were forwarded to have explained a large portion of the mass loss from GrIS. Yet the difference between ISSM-GrIS and SMB-GrIS are almost negligible. The reason for this is that you do not account for processes which explains fast dynamic response. Your model simply redistributed mass, except at the marine ice fronts. The slight dynamic reduction of the SMB mass loss signal, which you explain by the slow geometric adjustment (slopes, driving stress, etc.), has already been analysed in depth in the literature. As the title and abstract of the manuscript do not suggest that this small dynamic (long-term) effect is the target of your study, I fear that your model setup is not chosen appropriately.

While SMB dictates much of the difference, it is important to note that ISSM SMB-driven is not negligible - as dynamics contributes up to +21 Gt/yr and contributes to the uncertainty of the model-based mass balance estimate. Indeed, use of the model actually decreases the uncertainty that would be suggested by the spread in the RCM SMB forcing. A main goal of this study is to statistically quantify the differences between the model and GRACE estimates at a relatively high spatial and temporal resolution. Because ice sheet models are able to model SMB-driven dynamics with relatively high certainty in the physical model, we find the inclusion of this tool to be an important consideration in the accurate quantification of historic estimates of ice sheet MB evolution. In order to make our goal clearer, we have added a paragraph at the end of the introduction, which describes our intentions in terms of quantifying uncertainty in model output and observations in order to assess regional changes in ice discharge. From this viewpoint, we believe that our model setup is defensible.

[revised manuscript text omitted]

**Supplement Figures**

Model Mesh

[Figure]

A

B

**Figure S1.** The ISSM Greenland mesh for (A) the entire ice sheet and (B) the northwest margin, which, for reference, is outlined in red on (A).

[Figure]

**Figure S2.** Spatial representation of differences in mean annual amplitude from 2003-2012 between various combinations of model spinup and the ISSM_GrIS MAR3.5.2 presented in the manuscript (i.e. MAR3.5.2 forced by ERA-I reanalysis and BOX SMB used as the reference relaxation climatology, $\overline{SMB}$). Comparison runs include: (A) ISSM_GrIS MAR3.5.2, where MAR3.5.2 is forced with NCEP1 reanalysis; (B) ISSM_GrIS MAR3.5.2, where MAR3.5.2 SMB is used for $\overline{SMB}$; and (C) ISSM_GrIS MAR2.0, where MAR2.0 SMB (forced with ERA-I reanalysis) is used for $\overline{SMB}$. Results are less sensitive to variations in RCM forcing (A) and choice of spinup product (B) than to RCM version (C).

[Figure]

**Figure S3.** Spatial representation of differences in mass trend from 2003-2012 between various combinations of model spinup and the ISSM_GrIS MAR3.5.2 presented in the manuscript (i.e. MAR3.5.2 forced by ERA-I reanalysis and BOX SMB used as the reference relaxation climatology, $\overline{SMB}$). Comparison runs include: (A) ISSM_GrIS MAR3.5.2, where MAR3.5.2 is forced with NCEP1 reanalysis; (B) ISSM_GrIS MAR3.5.2, where MAR3.5.2 SMB is used for $\overline{SMB}$; and (C) ISSM_GrIS MAR2.0, where MAR2.0 SMB (forced with ERA-I reanalysis) is used for $\overline{SMB}$. Results are less sensitive to choice of spinup product (B) than to variations in RCM forcing (A) or to RCM version (C).

[Figure]

**Figure S4.** (A) Modeled surface velocities (m/yr), from the relaxed ISSM Greenland and (B) the departure of modeled surface velocities from observations. (Rignot and Mouginot, 2012). Note the non-linear color scales.

[Figure]

**Figure S5.** (A) Modeled ice thickness (m), from the relaxed ISSM Greenland and (B) the departure of modeled ice thicknesses from observationally-based data (Morlighem et al., 2014b).

[Figure]

**Figure S6.** Spatial representation of trend in surface mass from 2003-2009 as estimated from (A) GRACE_JPL and (B) ICESat altimetry (Csatho et al., 2014), and (C) the difference: GRACE_JPL - ICESat.

[Figure]

**Figure S7.** Cumulative mass from 2003-2012 for (A) all of Greenland and (B) the Greenland Interior, comparing observations from GRACE (GRACE_JPL), with model outputs: ISSM over the Greenland Ice Sheet (ISSM_GrIS), SMB anomalies over the Greenland Ice Sheet (SMB_GrIS), ISSM_GrIS with mass from the periphery (ISSM_GrIS+P), and ISSM_GrIS+P for each individual SMB forcing (ISSM_GrIS+P MAR, ISSM_GrIS+P RACMO, ISSM_GrIS+P BOX).

[Figure]

**Figure S8.** (A) Change in modeled mean annual surface velocities (m/yr) and (B) change in model ice thicknesses (m) during the 10-year ISSM simulation period (2003-2012). Model output is presented as the mean of three different ISSM simulation runs (ISSM_GrIS BOX, ISSM_GrIS MAR, and ISSM_GrIS RACMO). Note the non-linear color scales.

[Figure]

**Figure S9.** (A) Total dynamic ice thickness change (difference between the cumulative mass contribution from the SMB forcing anomalies and the total ice thickness change) simulated by ISSM Greenland (2003-2012); (B) change in surface slope during the simulation; and (C) change in the magnitude of the driving stress over the same period. Model output is presented as the mean of three different ISSM simulation runs (ISSM_GrIS MAR, ISSM_GrIS RACMO, and ISSM_GrIS BOX).

[Figure]

**Figure S10.** Spatial representation of trend in surface mass for the Greenland periphery as estimated from (A) RACMO, (B) BOX, and (C) MAR.

[Figure]

**Figure S11.** Interior mascons, total cumulative mass timeseries for GRACE_JPL, SMB_GrIS, and ISSM_GrIS (including the mean and results from the individual simulations of ISSM_GrIS MAR, ISSM_GrIS RACMO, and ISSM_GrIS BOX). Also included is the residual between GRACE_JPL and ISSM_GrIS (green).

[Figure]

**Figure S12.** Southwest mascons, total cumulative mass timeseries for GRACE_JPL, SMB_GrIS, ISSM_GrIS (including the mean and results from the individual simulations of ISSM_GrIS MAR, ISSM_GrIS RACMO, and ISSM_GrIS BOX), and ISSM_GrIS+P. Also included is the residual between GRACE_JPL and ISSM_GrIS+P (green).

[Figure]

**Figure S13.** Same as Fig. S12 but for Northeast mascons.

[Figure]

**Figure S14.** Same as Fig. S12 but for Southeast mascons.

[Figure]

**Figure S15.** Same as Fig. S12 but for Northwest mascons.

---

## Author Response (AR3)

Thank you very much for your helpful noting of typos in the manuscript. Below, please see our responses to your comments and suggested modifications. For those comments that required updates to text, we enclose a paper draft with corresponding changes noted in red.

**1 Editor**

Dear Nicole-Jeanne,

Thanks for this revised version which has greatly been improved thanks to the reviewers suggestions. I think it can now be accepted for publication in The Cryosphere. I have nevertheless found few typos that should be corrected before going to the editing process.

Thanks for having submitting your work to The Cryosphere!

Regards, Olivier Gagliardini

- page 6, lines 12-20: it is not clear from this paragraph if the post-processing of the GRACE data are part of this work or has been already published in Watkins et al. (2015)? This should be re-written to make it clearer.

We have added a sentence to explain that all post-processing performed is consistent with the Watkins et al. paper, as well as the publicly available dataset, with the exception of two corrections: the mean pole correction and the correction for the jumps in the background dealiasing product. These two corrections are not made in the publicly available dataset, but we have improved the analysis by including them.

- page 7, line 5: 91,490 **triangular** elements

We have made the correction.

- page 7, line 27: it was **close to** steady-state

We have made the correction.

- page 11, line 26: should be mentioned that the thickening of 2cm is in ice equivalent.

We have added this clarification.

- page 11, lines 27-28: this sentence regarding the Northeast of GrIS should be moved as it cuts a discussion regarding the interior of GrIS.

We have reformulated this text to explain that the interior thickening is dominant in mascons 58 and 88, rather than calling this the Northeast. We realize now that this text was confusing.

- Figures with moscons: would be nice to add the number of moscon on all these figures (not only Fig. 1) to help following the text which is highly referencing to these numbers.

Thank you for this suggestion - we agree it will add value. We have remade all figures that display data on Greenland to include 1) the outline of the mascons, and 2) the mascon number. We chose linewidths and font colors so as to try to increase legibility of this new addition. Overall, we feel that the new figures will provide the audience increased access to the interpretation of the results in the discussion.

- page 19, line 23: since the regions that

We have made the correction.

- Acknowledgement: you have had more than one anonymous referee.

Thank you! We have made the correction.

[revised manuscript text omitted]

**Supplement Figures**

Model Mesh

A

[Figure]

B

200 400 km

**Figure S1.** The ISSM Greenland mesh for (A) the entire ice sheet and (B) the northwest margin, which, for reference, is outlined in red on (A).

[Figure]

**Figure S2.** Spatial representation of differences in mean annual amplitude from 2003-2012 between various combinations of model spinup and the ISSM_GrIS MAR3.5.2 presented in the manuscript (i.e. MAR3.5.2 forced by ERA-I reanalysis and BOX SMB used as the reference relaxation climatology, $\overline{SMB}$). Comparison runs include: (A) ISSM_GrIS MAR3.5.2, where MAR3.5.2 is forced with NCEP1 reanalysis; (B) ISSM_GrIS MAR3.5.2, where MAR3.5.2 SMB is used for $\overline{SMB}$; and (C) ISSM_GrIS MAR2.0, where MAR2.0 SMB (forced with ERA-I reanalysis) is used for $\overline{SMB}$. Results are less sensitive to variations in RCM forcing (A) and choice of spinup product (B) than to RCM version (C).

[Figure]

**Figure S3.** Spatial representation of differences in mass trend from 2003-2012 between various combinations of model spinup and the ISSM_GrIS MAR3.5.2 presented in the manuscript (i.e. MAR3.5.2 forced by ERA-I reanalysis and BOX SMB used as the reference relaxation climatology, $\overline{SMB}$). Comparison runs include: (A) ISSM_GrIS MAR3.5.2, where MAR3.5.2 is forced with NCEP1 reanalysis; (B) ISSM_GrIS MAR3.5.2, where MAR3.5.2 SMB is used for $\overline{SMB}$; and (C) ISSM_GrIS MAR2.0, where MAR2.0 SMB (forced with ERA-I reanalysis) is used for $\overline{SMB}$. Results are less sensitive to choice of spinup product (B) than to variations in RCM forcing (A) or to RCM version (C).

[Figure]

**Figure S4.** (A) Modeled surface velocities (m/yr), from the relaxed ISSM Greenland and (B) the departure of modeled surface velocities from observations. (Rignot and Mouginot, 2012). Note the non-linear color scales.

[Figure]

**Figure S5.** (A) Modeled ice thickness (m), from the relaxed ISSM Greenland and (B) the departure of modeled ice thicknesses from observationally-based data (Morlighem et al., 2014b).

[Figure]

**Figure S6.** Spatial representation of trend in surface mass from 2003-2009 as estimated from (A) GRACE_JPL and (B) ICESat altimetry (Csatho et al., 2014), and (C) the difference: GRACE_JPL - ICESat.

[Figure]

**Figure S7.** Cumulative mass from 2003-2012 for (A) all of Greenland and (B) the Greenland Interior, comparing observations from GRACE (GRACE_JPL), with model outputs: ISSM over the Greenland Ice Sheet (ISSM_GrIS), SMB anomalies over the Greenland Ice Sheet (SMB_GrIS), ISSM_GrIS with mass from the periphery (ISSM_GrIS+P), and ISSM_GrIS+P for each individual SMB forcing (ISSM_GrIS+P MAR, ISSM_GrIS+P RACMO, ISSM_GrIS+P BOX).

[Figure]

**Figure S8.** (A) Change in modeled mean annual surface velocities (m/yr) and (B) change in model ice thicknesses (m) during the 10-year ISSM simulation period (2003-2012). Model output is presented as the mean of three different ISSM simulation runs (ISSM_GrIS BOX, ISSM_GrIS MAR, and ISSM_GrIS RACMO). Note the non-linear color scales.

[Figure]

**Figure S9.** (A) Total dynamic ice thickness change (difference between the cumulative mass contribution from the SMB forcing anomalies and the total ice thickness change) simulated by ISSM Greenland (2003-2012); (B) change in surface slope during the simulation; and (C) change in the magnitude of the driving stress over the same period. Model output is presented as the mean of three different ISSM simulation runs (ISSM_GrIS MAR, ISSM_GrIS RACMO, and ISSM_GrIS BOX).

[Figure]

**Figure S10.** Spatial representation of trend in surface mass for the Greenland periphery as estimated from (A) RACMO, (B) BOX, and (C) MAR.

[Figure]

**Figure S11.** Interior mascons, total cumulative mass timeseries for GRACE_JPL, SMB_GrIS, and ISSM_GrIS (including the mean and results from the individual simulations of ISSM_GrIS MAR, ISSM_GrIS RACMO, and ISSM_GrIS BOX). Also included is the residual between GRACE_JPL and ISSM_GrIS (green).

[Figure]

**Figure S12.** Southwest mascons, total cumulative mass timeseries for GRACE_JPL, SMB_GrIS, ISSM_GrIS (including the mean and results from the individual simulations of ISSM_GrIS MAR, ISSM_GrIS RACMO, and ISSM_GrIS BOX), and ISSM_GrIS+P. Also included is the residual between GRACE_JPL and ISSM_GrIS+P (green).

[Figure]

**Figure S13.** Same as Fig. S12 but for Northeast mascons.

[Figure]

**Figure S14.** Same as Fig. S12 but for Southeast mascons.

[Figure]

**Figure S15.** Same as Fig. S12 but for Northwest mascons.